# Modelling river-sea continuum: the case of the Danube Delta

Christian Ferrarin[1], Debora Bellafiore[1], Alejandro Paladio Hernandez[1], Irina Dinu[2], and Adrian Stanica[2]

[1]CNR - National Research Council of Italy, ISMAR - Marine Sciences Institute, Venice, Italy
[2]National Institute of Marine Geology and Geoecology - GeoEcoMar, Bucharest, Romania

**Correspondence:** Christian Ferrarin (christian.ferrarin@cnr.it)

**Abstract.** Understanding water transport and circulation in coastal seas and transitional environments is a key focus of oceanographic and climate research, particularly in recognizing the role of the land-sea interface. The Danube Delta serves as a natural laboratory for river-sea hydrodynamic modelling due to its complex morphology, composed of multiple river branches, channels, and lagoons. Moreover, this coastal environment is subjected to various natural and anthropogenic stressors, and numerical modelling can provide a scientific basis for assessing the impact of human activities. In this work, the SHYFEM finite element hydrodynamic model was applied to the entire river-sea continuum of the Danube Delta region to describe the transport and mixing processes within and between the interconnected water bodies forming the delta. The model was run for the period 2015-2019 and enabled the characterization of: (1) water discharge distribution among the river branches; (2) general hydrodynamic characteristics of the coastal region of freshwater influence; (3) transport time scale of the Razelm Sinoie Lagoon System. Finally, the Danube Delta modelling tool was used to evaluate the potential effects of hydrological reconnection (restoration) measures in the Razelm Sinoie Lagoon System aimed at improving connectivity and water renewal.

## 1 Introduction

Coastal environments at the river-sea interface, like estuaries and deltas, are critical components of coastal ecosystems due to their importance in supporting biodiversity and providing ecosystem services such as nutrient cycling, carbon sequestration, and habitat for marine life (Newton et al., 2023, and references therein). However, these regions are highly sensitive to both natural events (e.g., storms, sea-level rise, droughts) and anthropogenic pressures (e.g., damming, land reclamation, pollution). As such, modelling these environments is essential for understanding their dynamics, predicting their responses to environmental change, and guiding sustainable management practices.

Modelling estuaries and deltas is challenging due to their complex morphology made of different components such as river network, coastal lakes, lagoons, creeks, marshes. These different water compartments are generally interconnected and thus influencing each other. In this context, unstructured ocean models, like SCHISM (Zhang et al., 2016), SHYFEM (Umgiesser et al., 2022), FESOM-C (Androsov et al., 2019), Delft3D (D-Flow, 2023), TELEMAC (Telemac-Mascaret, 2022), ADCIRC (Zhang and Yu, 2025), FVCOM (Chen et al., 2003), SLIM (Vallaeys et al., 2018) have proven to be powerful tools for simulating complex hydrodynamics in shallow estuarine and deltaic environments. These models help assess the effects of climate change (Ferrarin et al., 2014; Pein et al., 2023), human interventions (Ferrarin et al., 2013; Thanh et al., 2020), and natural variability on the structure and functioning of coastal systems (Maicu et al., 2018; Zhu et al., 2020; Feizabadi et al., 2024).

Models used in estuarine and delta research typically integrate hydrodynamics, sediment transport, water quality, and ecological processes. However, the higher the complexity of the processes to be investigated, the higher is the amount of data needed for the model implementation and testing.

This paper investigates the hydrodynamic processes, water exchange, and connectivity among the various interconnected water compartments - river branches, channels, lagoons, and the coastal sea - that together form the Danube Delta river–sea continuum. To achieve this, we implemented the SHYFEM model (System of HydrodYnamic Finite Element Modules; Umgiesser et al., 2022) across the entire Danube Delta, encompassing approximately 500 km of the river network, the Razelm–Sinoie Lagoon System, and part of the prodelta coastal sea (Fig. 1). The model outputs are used to quantify the distribution of water

discharge among river branches, assess the influence of multiple river plumes on coastal dynamics, and analyze the water exchange and renewal capacity of the Razelm–Sinoie Lagoon System. Moreover, the numerical tool is used to assess the potential impacts of different hypothetical lagoon-sea reconnection solutions (*what-if* scenarios) on the processes regulating the exchanges between the river, lagoon, and sea.

## 1.1   The Danube Delta

The Danube Delta is the final part of the Danube River's journey of almost 2,900 km, crossing 10 countries and draining a hydrographic basin of over 800,000 $km^2$ from 19 states towards its connection with the Black Sea (e.g., Panin, 1998). The Danube Delta plain begins at the first bifurcation of the Danube, called Ceatal Izmail (Fig. 1). Here, the Danube River divides into two distributaries: the northern one, Chilia, and the southern one, Tulcea. The Chilia distributary creates a natural border between Romania and Ukraine. The Tulcea distributary divides again at Ceatal, 17 km farther downstream, into two

other branches, Sulina and Sf. Gheorghe. The fluvial delta plain covers 4,000 $km^2$ and the marine one covers 1,800 $km^2$. The Romanian section of the Danube Delta, including the Razelm-Sinoie Lagoon System, was designated a Biosphere Reserve in 1991 under UNESCO's "Man and the Biosphere" Programme and has remained a Nature Reserve ever since.

    The Razelm Sinoie Lagoon System (hereinafter RSLS) extends for about 1,000 $km^2$ and is located in the southern part of the Danube Delta Biosphere Reserve. RSLS is a semiclosed shallow water body (the average depth is 1.8 m)receiving Danube

freshwater from the Dunavăţ and Dranov canals and exchanging waters with the Black Sea via the Edighiol and Periboina inlets. The lagoon system is suffering from poor water renewal and stagnation (Dinu et al., 2015) and has been significantly affected by human interventions since the end of the 19th century (Panin, 1996, 1998, 1999; Giosan et al., 2006; Vespremeanu-Stroe et al., 2013; Constantinescu et al., 2023). Dredging works to connect the Razelm lagoon to the Sf. Gheorghe arm of the Danube via the Dunavăţ and Dranov canals were finalized at the beginning of the 20th century. As a result, more fresh water

is discharged into the lagoon system. During the 1950s, management plans were made to decrease the salinity in the lagoon system, with the purpose to increase the freshwater fish culture productivity. Between 1960 and 1990, the lagoon has been used mainly for irrigations and, secondly, for fish breeding. In 1973, the Portiţa Inlet of the Razelm Lagoon was completely closed by a system of breakwaters and groins. Following the closure of the Portiţa inlet, the Razelm Lagoon was transformed into a a freshwater lake, receiving Danube water via Dranov and Dunavăţ Canals. The inlet of Gura Buhazului in the southern part

of the Sinoie Lagoon was clogged more than 3 decades ago (late 1980s - early 1990s). The permanent circulation between

the Sinoie Lagoon and the Black Sea has been restored by the beginning of the years 2000, being controlled by the Periboina and Edighiol inlets. The Periboina inlet has become clogged around 2017, with an intermittent connection with the sea during spring and autumn months up to the year 2021. As part of the master plan for the protection of the Romanian Littoral against erosion, a major hydraulic engineering project is currently implemented, to ensure a permanent water exchange through the Periboina Inlet.

The Danube River before the delta has an average water discharge of 6,500 m$^3$ s$^{-1}$, with values ranging from 1,300 to 16,000 m$^3$ s$^{-1}$ (Pekárová et al., 2021). The two major wind regimes characterizing the study area are from north-east, being the most intense, and south-south-west, that can drive alongshore water and sediment transport (Dan et al., 2009).

## 2 Methods

### 2.1 The modelling system

In this study, we used the System of HydrodYnamic Finite Element Modules (SHYFEM, Umgiesser et al., 2022) model to simulate the three-dimensional (3D) hydrodynamics in the river-sea continuum of the whole Danube Delta region. SHYFEM is an open-source unstructured ocean model for simulating hydrodynamics and transport processes at very high resolution. The model solves the primitive equations written in z-coordinate system and using a finite element numerical method and semi-implicit time stepping. The model has been already applied to simulate the hydrodynamics in the Mediterranean Sea (Ferrarin et al., 2018), in the Adriatic Sea (Bellafiore et al., 2018; Ferrarin et al., 2019), in the Black Sea (Bajo et al., 2014) and in several coastal systems (Ferrarin et al., 2021; Umgiesser et al., 2022).

The SHYFEM model was applied to a domain that comprises the river network of the Danube Delta from Isaccea (100 km upstream from the river mouths, in the vicinity of the delta apex at Ceatal Izmail) to the sea (from north to south the Chilia, Sulina and Sf. Gheorghe branches), the Razelm Sinoie Lagoon System, the nearby prodelta and shelf area (290x100 km) and the narrow canals and inlets connecting the different water compartments (Dunavăţ, Dranov, Canal 2, Canal 5, Edighiol and Periboina) (Fig. 1). The application of a triangular unstructured grid in the hydrodynamic model has the advantage of describing accurately complicated bathymetry and irregular boundaries in the river and shallow water areas. It can also solve the combined offshore-coastal interactions and small-scale river-sea dynamics in the same discrete domain by subdividing the basin into triangles varying in form and size. The unstructured mesh is generated using the mesh-generation software GMSH (Geuzaine and Remacle, 2009). The numerical domain is composed of about 48,000 triangular elements having horizontal resolution varying from about 4 km in the open sea to a few metres in the river branches and the connecting channels. Vertically, the model runs in z layer configuration, with 68 layers of increasing thickness, from 1 m in the topmost layers, to 50 m in the deepest ones (below 500 m).

The model bathymetry is obtained by a bilinear interpolation on the numerical grid of the following available datasets (all referred to the Marea Neagra Sulina vertical datum):

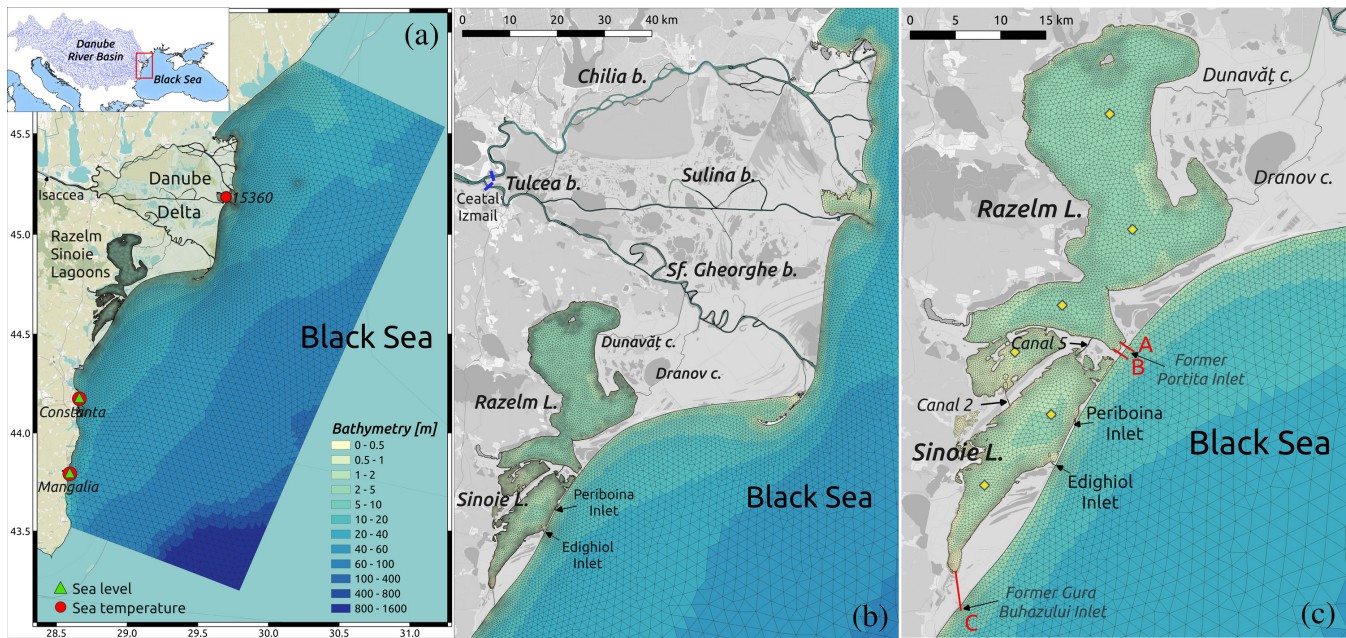

**Figure 1.** (a) Unstructured numerical grid and bathymetry of the hydrodynamic model of the Danube Delta and Black Sea shelf with the red dots and the green triangles marking the sea temperature and sea level monitoring stations, respectively; (b) zoom of the grid over the Danube Delta with the blue bars near Ceatal Izmail indicating the river discharge monitoring stations; (c) zoom of the grid over the Razelm Sinoie Lagoon Systems with the red bars illustrating the considered reconnection solutions and the yellow diamonds marking the satellite SST control points. Background: ©OpenStreetMap contributors 2024; distributed under the Open Data Commons Open Database License (ODbL) v1.0.

- – the 2022 European Marine Observation and Data Network dataset (EMODnet Bathymetry Consortium, 2022) for the shelf sea on a regular grid of 1/16*1/16 arc minutes, ca. 115 metre grid;

- – the 2024 dataset for the Razelm Sinoie Lagoon System acquired on (mostly West-East-oriented) transects spaced 450 m apart on average and covering the whole system. The distance between two points within each transect is ∼1 m.

- – three separate multibeam datasets (provided at a ∼1 m resolution) for the main river branches: the 2023 dataset for Chilia; the 2019 dataset for Sulina; the 2016-2017 dataset for Sf. Gheorghe. Available sparse data was used for some secondary branches and small channels.

The maximum allowable time step in the simulation was set to 60 s, and the model adopts automatic sub-stepping over time to enforce numerical stability with respect to advection and diffusion. The wind drag coefficient was set to $2.5 \ 10^{-3}$. Bottom friction is computed following the Strickler formulation (Umgiesser et al., 2004) with the friction parameter considered homogeneous over the whole domain and set equal to $32 \ \mathrm{m}^{1/3} \ \mathrm{s}^{-1}$. Vertical diffusivities are calculated by the $k - e$ turbulence closure model. Model outputs are saved at a daily frequency. This choice was made to limit the volume of model outputs and is justified by the fact that the boundary conditions for the open sea and the river have a daily frequency.

To investigate how the difference forcing and processes influence the water mixing and renewal in the semiclosed Razelm Sinoie Lagoon System, the numerical model has been used to estimate two transport time scales: the water flushing time (WFT) and the water renewal time (WRT) (Umgiesser et al., 2014). The basin-wide WFT is defined as the theoretical time necessary to replace the complete volume of the water body with new water and assuming a hypothetical fully mixed basin and is computed diving the basin volume by the volumetric water flux flowing out of the system. WRT is computed by simulating the transport and diffusion of a Eulerian conservative tracer released uniformly throughout the entire lagoon system with a concentration corresponding to 1, while a concentration of zero was imposed on the seaward and freshwater boundaries. The local WRT is considered as the time required for each cell of the RSLS to replace the mass of the conservative tracer, originally released, with new water. The ratio between the basin wide WFT and WRT can be interpreted as an index of the mixing behaviour of the basin. The reader may refer to Cucco et al. (2009) and Umgiesser et al. (2022) for a more comprehensive description of the transport time scales.

## 2.2 Numerical experiments

SHYFEM is here used to reproduce the 3D hydrodynamics of the Danube Delta under the influence of river discharge, heat and momentum fluxes at the water surface, salinity and sea temperature gradients and open sea forcing (sea level oscillations and currents). Moreover, the concurrence of intense atmospheric forcing, direct morphological interventions within the delta territory and freshwater inflows lead the Danube Delta to be characterized by a wide range of different transport phenomena.

To reproduce the past sea conditions over the period 2014/01/01 - 2019/12/31, the simulations were forced by 3-hourly wind, mean sea level pressure, air temperature, relative humidity, incident solar radiation, total precipitation and cloud cover fields from Copernicus European Regional ReAnalysis (CERRA; Schimanke et al., 2021) made available via the Copernicus Climate Change Service (https://doi.org/10.24381/cds.622a565a). CERRA has a horizontal resolution of 5.5 km and is forced by the global ERA5 reanalysis (Hersbach et al., 2020). The SHYFEM hydrodynamic model was forced at the lateral boundary of the Black Sea with daily sea level, current velocity, sea temperature and salinity fields from the Black Sea Physics Reanalysis made available via the E.U. Copernicus Marine Service Information (Lima et al., 2020, https://doi.org/10.25423/CMCC/BLKSEA_MULTIYEAR_PHY_007_004, accessed on 14-Jun-2024). Daily observed water river discharges at Isaccea were provided by the National Institute of Hydrology and Water Management of Romania and imposed as a boundary condition for the Danube River. Water temperature at the Danube River boundary was taken from the daily results of the *wflow* catchment model implemented over the Danube River basin (van Gils et al., 2025).

In the past, the lagoons were connected to the sea via several inlets, while nowadays only the Periboina and Edighiol connections remain open. Additional numerical experiments were conducted to investigate the potential effects on the lagoons' water renewal and salinisation of different reconnection solutions designed in collaboration with local stakeholders to enhance the river-lagoon-sea exchange. The dredging of a new inlet is under consideration by local communities and authorities, as part of the activities developed under the framework of the Horizon Europe Project DANUBE4all (https://www.danube4allproject. eu/). The three *what-if* scenarios considered in this study consisted of opening one 1.5 m depth and 70 m wide channel to connect the either the Razelm Lagoon (solutions A and B in Fig. 1c) or the Sinoie Lagoon (solution C in Fig. 1c) with the

Black Sea. These reconnection solutions are located in the vicinity of previous inlets, now either closed by humans (Portiţa)
or clogged (Gura Buhazului inlet, active till the beginning of the 1990s). The period, parametrization, forcing and boundary
conditions considered in these *what-if* numerical experiments are the same as those adopted in the reference run (hereinafter
called REF).

In all simulations, the entire year 2014 is considered as the model's spin-up time, and the results are analyzed over the period
2015-2019.

## 2.3 Model validation

The application of the SHYFEM model to the Danube Delta was validated by comparing various parameters. The validation
aims at assessing the skill of SHYFEM model in reproducing the hydrodynamics in the different water compartment of the
delta. The available datasets used in the validation procedures are grouped into the following four categories:

– In-situ river discharge: daily values are provided by the National Institute of Hydrology and Water Management of
  Romania for two river sections near Ceatal Izmail where the Danube River splits into the Chilia and Tulcea branches
  (blue bars in Fig. 1b). The Tulcea branch downstream splits into the Sulina and Sf. Gheorghe arms, but no observations
  were available for these branches.

– In-situ sea level: hourly values were retrieved from the in-situ ocean thematic centre of the Copernicus Marine Service
  (https://marineinsitu.eu/dashboard/) for stations Constanta and Mangalia (green triangles in Fig. 1a).

– In-situ sea temperature: daily values were retrieved from the in-situ ocean thematic centre of the Copernicus Marine
  Service (https://marineinsitu.eu/dashboard/) for BS_TS_MO stations 15360, 15480 (Constanta) and 15499 (Mangalia)
  (red dots in Fig. 1a).

– Satellite Sea Surface Temperature (SST): Level 2 data derived from the Landsat-8 Thermal Infrared Sensor (TIRS)
  extracted over six locations within the Razelm Sinoie Lagoon to cover the spatial variability in the lagoons (yellow
  diamonds in Fig. 1c). The SST data are generated through the application of an atmospheric correction algorithm to
  the Top-Of-Atmosphere thermal radiance values from the TIRS bands (B10 and B11). This algorithm accounts for
  atmospheric effects such as water vapor and aerosol interference and applies a split-window technique to estimate surface
  temperatures with high accuracy (Barsi et al., 2003). The data, provided by the United States Geological Survey (USGS),
  were accessed via the Google Earth Engine platform using the LANDSAT/LC08/C02/T1_L2 dataset. To ensure data
  quality and reliability, scenes with cloud cover below 1% were selected, effectively minimizing the impact of atmospheric
  interference. For each location, SST values were extracted at midnight using a 1x1 pixel window, corresponding to the
  nearest pixel to the specified coordinates. A total of 31 time frames providing 135 valid SST observations were selected
  in the period from 2015 to 2019.

In this work, we consider the root mean square error (RMSE), the difference between the mean of simulation results and
observations (BIAS), the Pearson correlation coefficient between model results and observations (CC) and the slope of the

**Table 1.** Statistical analysis of simulated river discharge, sea level, sea temperature and sea surface temperature.

| Variable | Station | N data | RMSE | BIAS | CC | SLOPE |
|---|---|---|---|---|---|---|
| River discharge ($m^3 \ s^{-1}$) | Chilia | 120 | 158 | -46 | 1.00 | 0.90 |
| | Tulcea | 120 | 158 | 43 | 1.00 | 1.10 |
| Sea level (cm) | Constanta | 624 | 6.5 | - | 0.66 | 0.70 |
| | Mangalia | 722 | 7.8 | - | 0.55 | 0.62 |
| Sea Temperature (°C) | 15360 | 972 | 1.7 | 0.2 | 0.98 | 1.08 |
| | Constanta | 966 | 1.6 | -0.4 | 0.97 | 0.98 |
| | Mangalia | 908 | 1.5 | -0.2 | 0.97 | 0.96 |
| SST (°C) | Razelm Sinoie | 135 | 1.0 | 0.3 | 0.97 | 0.99 |

linear regression best-fit line (SLOPE) as the metrics for measuring the accuracy of the numerical results of the reference simulation. The results of the statistical analysis of river discharge, sea level and sea temperature are reported in Table 1.

As shown in Fig. 2a, the Danube River discharge at Isaccea (the upstream river boundary in the model domain) in the period 2015-2019 varies from 2,000 to 14,000 $m^3 \ s^{-1}$ with an average value of about 6,000 $m^3 \ s^{-1}$. Figures 2b and 2c represent the scatter plots of observed and simulated river discharge through the Chilia and Tulcea river branches at Ceatal Izmail. The model well represents the total water discharge distribution in the Chilia and Tulcea branches, with a RMSE of 158 $m^3 \ s^{-1}$ and CC of 1.00 in both distributaries (Table 1). It is worth noting that the model tends to underestimate (overestimate) the peak discharge values in the Chilia (Tulcea) arm, as revealed also by a slope of the linear regression best-fit line of 0.90 (1.10). The underestimation at Chilia coincides with situations of overestimation at Tulcea. The inconsistency during flood events is likely attributable to the model's inability to correctly reproduce the river overflow into the surrounding floodplains.

We report in Figure 3, the time series of the modelled daily sea levels compared with the observations for 2017 at the coastal stations Constanta and Mangalia located along the Romanian coast (green triangles in Fig. 1a). The model can reproduce the long-term sea level variability as well as the major fluctuations associated with intense meteorological events (storm surges) with typical time scales of 1 to 10 days. The model is slightly underestimating sea levels in some periods (e.g., Constanta in July-August and Mangalia in March). The statistical analysis revealed that RMSE and CC in Constanta and Mangalia are 6.5 cm and 0.66, and 7.8 cm and 0.55, respectively (Table 1). It must be noted that the performance of the coastal model in reproducing sea levels is strongly determined by the open boundary conditions, therefore any discrepancy in the sea level simulated by the Black Sea Physics Reanalysis is propagated to the coast.

As presented in the right panels of Fig. 3 (illustrated for the year 2019 at stations 15360 and Mangalia), the numerical model captures correctly the seasonal variability as well as the short-term fluctuations of the sea temperature in the investigated area. The statistical analysis of the simulated sea temperatures revealed that RMSE is between 1.5 and 1.7 °C and the CC is always above 0.97 (Table 1).

Satellite derived data demonstrate that sea surface temperature in RSLS varies strongly over the 2015-2019 period with values ranging from 0 °C in winter to 28 °C in summer. Spatially SST in the lagoons has a small spatial variability with

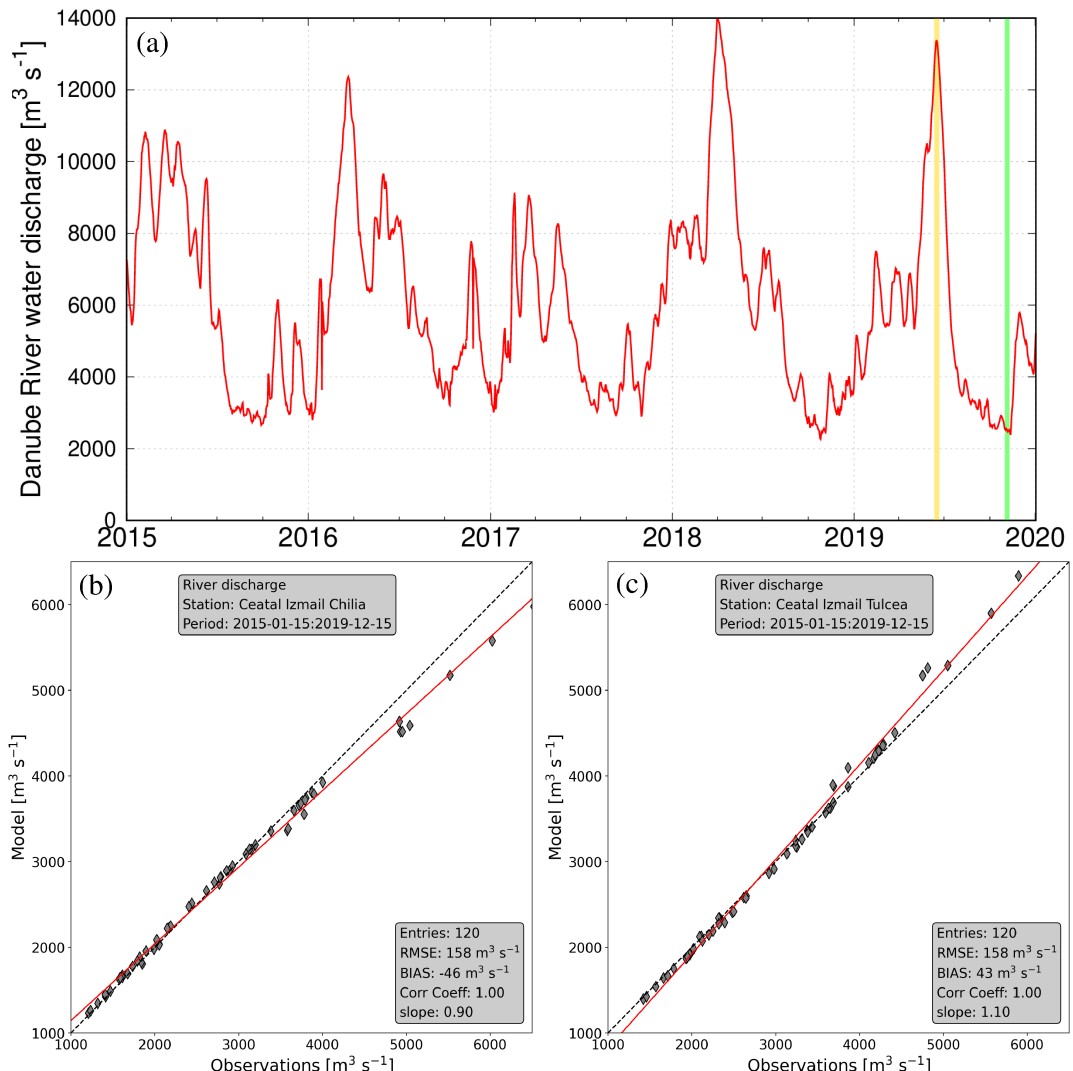

**Figure 2.** Danube River water discharge timeseries (a) and scatter plots of observed and modelled river discharge in the Chilia (b) and Tulcea (c) river branches. The gold and green bars in panel (a) indicate the flood and drought conditions considered in Figs. 5b and 5c. The gray diamonds and the red lines in panels b and c represent the scatter data and the line of best fit, respectively.

difference among stations lower than 1.5 °C. To assess the model's performance, we extracted water temperature values from the simulation results in the surface layer at the location and time corresponding to the satellite SST data. The statistical comparison between model results and observations reported in Table 1 (RMSE = 1°C, BIAS = 0.3°C, CC = 0.97, slope = 0.99) demonstrated that SHYFEM can represent well both the spatial and temporal variability revealed by satellite data.

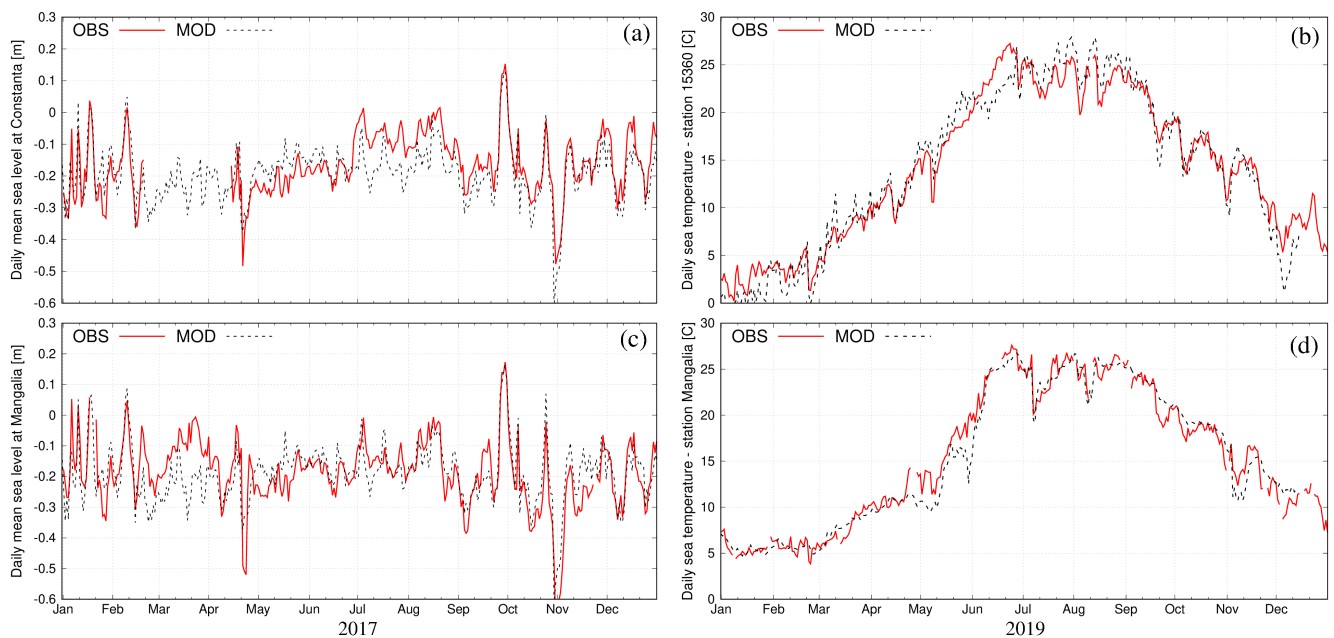

**Figure 3.** Observed (red line) and simulated (black dashed line) sea levels at Constanta (a) and Mangalia (c), and sea temperature at station 15360 (b) and Mangalia (d).

## 3 Results

The modelling results are presented and analyzed separately for the main water compartments forming the delta: the river network (section 3.1), the coastal sea (section 3.2) and the lagoons (section 3.3). The assessment of the impact of the different lagoon-sea reconnection interventions is included in section 3.3.1.

### 3.1 Water division in the river network of the delta

The Danube Delta's river network comprises a highly complex system of hundreds of natural and artificial channels, streams, marshes, and lakes, whose morphological complexity exceeds the resolution capabilities of the current model implementation. The model was configured to represent only the most hydraulically significant watercourses, enabling the estimation of water discharge distribution among the principal river branches. Here, the water fluxes were extracted for several river sections and averaged over the simulation period (2015-2019) to estimate the relative runoff (in %) of each branch with respect to the average Danube River discharge imposed at the open boundary of Isaccea (6000 m$^3$ s$^{-1}$). The average water division values are reported in Fig. 4. Unless otherwise specified, the reported values refer to averages over the whole 2015-2019 simulation period.

The Danube discharge firstly subdivides into the Chilia and Tulcea branches, which have an average fraction of 46.3 and 53.7 %, respectively, this well reproducing the observed distribution. The Chilia branch in the northern part of the delta is

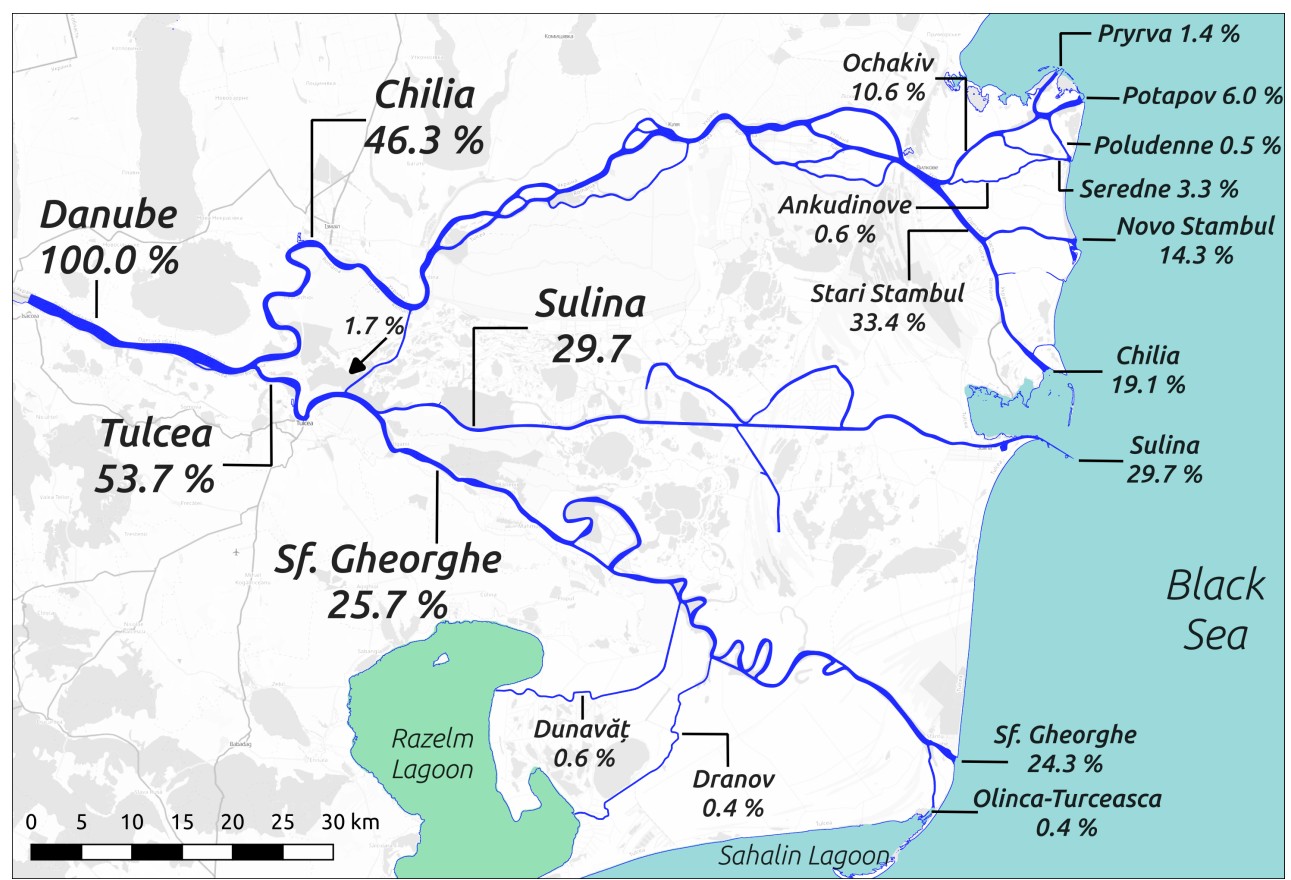

**Figure 4.** Modelled average water diversion within the Lower Danube River downstream of Isaccea. Background: ©OpenStreetMap contributors 2024; distributed under the Open Data Commons Open Database License (ODbL) v1.0.

characterized by a complex network of secondary arms that depart and rejoin to the main channel. 20 km from the sea, Chilia bifurcates into the Ochakiv (also known as Ochakov) channel (10.6 %), that downstream originates four mouths, and the Stari Stambul Vechi (33.4 %) channel, which splits into Novo Stambul (14.3 %) and Chilia (19.1 %) outflows. The 18 km long Tulcea distributary bifurcates into the Sulina branch, carrying almost 29.7 % of the Danube waters to the sea, and the Sf. Gheorghe (25.7 %) branch flowing through the southern part of the delta. A small fraction of the Sf. Gheorghe discharge is captured by the Dunavăț (0.6 %) and Dranov (0.4 %) canals that flow down into the Razelm Lagoon. Just before flowing into the Black Sea, the Sf. Gheorghe branch separates in the Olinca-Turceasca channel (0.4 %), flowing into the Sahalin Lagoon (0.4 %), and the main mouth (24.3 %).

We point out that the model estimate of the water division into the multiple branches of the delta is very sensitive to the accuracy of the morphological and bathymetric datasets used to create the numerical grid, which - due to lack of observations as mentioned in section 2.3 - has been validated only for the upper part of the delta river network. However, the simulated distribution of the Danube's mean discharge between the Chilia, Sulina and Sf. Gheorghe (46.3 %, 29.7 % and 25.7 %, respec-

tively) is similar to the results reported by Nichersu et al. (2025) (45 %, 34 % and 21 %, respectively). It is worth noting that the river discharge division among the main branches has been altered by human interventions (Constantinescu et al., 2023; Bloesch et al., 2025).

## 3.2 Spatial and temporal variability of coastal dynamics

The dynamics in coastal areas at the river-sea interface is generally determined by the mixing processes induced by the interaction of river outflow and coastal currents, mainly driven by the open sea circulation and wind (Garvine, 1995; Fong and Geyer, 2002; Bellafiore et al., 2019). Along the coast, these processes create peculiar hydrodynamic patterns, the so-called river plumes, having thermohaline characteristics and buoyancy that allow to distinguish them from seawater. The river plume extension delineate the coastal Region Of Freshwater Influence (ROFI; Simpson et al., 1993).

In front of the Danube Delta, the general coastal circulation (determined averaging the values over the whole simulated period) reflects these processes with the several branches of the multiple-mouth delta forming separated freshwater plumes having shape and dimension defined by amount of water carried out by the different river branches and the coastline characteristics (Fig. 5a). Indeed, the largest plume is found south of the Sulina mouth, where the 8 km long artificial jetty enhances the offshore spread of riverine waters and creates a well-defined recirculation structure. It has to be noted that this plume is
reinforced by the freshwater discharged by the nearby Chilia mouth. Well-defined plumes can be also recognized out of the Sf. Gheorghe, Novo Stambul and Potapov mouths. On average, the ROFI associated with the Danube River extends for about 15 km offshore the river mouths. As illustrated in Fig. 5d, the freshwater inputs determine a stratified water column along the coast, with Black Sea waters (defined here as having salinity higher than 16 g $L^{-1}$) located on average below 5 to 10 m from the surface. A low-intensity (< 0.1 m $s^{-1}$) southward current characterize the shelf area which is also influenced by the
long-shore dynamics induced by the rivers flowing along the northwestern Black Sea coast (Southern Bug, Dniestr and Dniepr; Miladinova et al., 2020). The lagoon system is on average characterized by very low currents (in the order of a few cm $s^{-1}$) and salinity ranging from 1 to 4 g $L^{-1}$, with the Sinoie Lagoon being saltier than the Razelm lagoon due to the inflow of marine waters through the Edighiol and Periboina inlets.

   The hydrodynamics of the whole area is strongly variable in time and space depending on Danube River discharge and other
forcing (e.g., wind, heat fluxes, long-shore currents). As examples of a such a high variability, we analysed the dynamics in the investigated area during two events having different hydro-meteo-marine conditions: 1) a summer event (15 June 2019) with peak river discharge (13,000 m$^3$ $s^{-1}$) and northerly wind (Figs. 5b and Figs. 5e), and 2) an autumn event (11 November 2019) with low river discharge (2,400 m$^3$ $s^{-1}$) and southerly wind (Figs. 5c and 5f). The high freshwater input during peak Danube River flow extends ROFI far offshore and to the south, determining a reduction in salinity over a large portion of the coastal
area and the enhancement of the southward surface coastal currents up to 0.6 m $s^{-1}$ (Fig. 5b). During peak river flow and northerly wind conditions, vertical mixing processes near the coast occupy the whole water column (Fig. 5e). On the contrary, during low river discharge, the surface coastal dynamic is mainly driven by the wind. The autumn event presented in Fig. 5c is characterized by a general northward surface transport of saline waters with the ROFI limited to river plumes extending

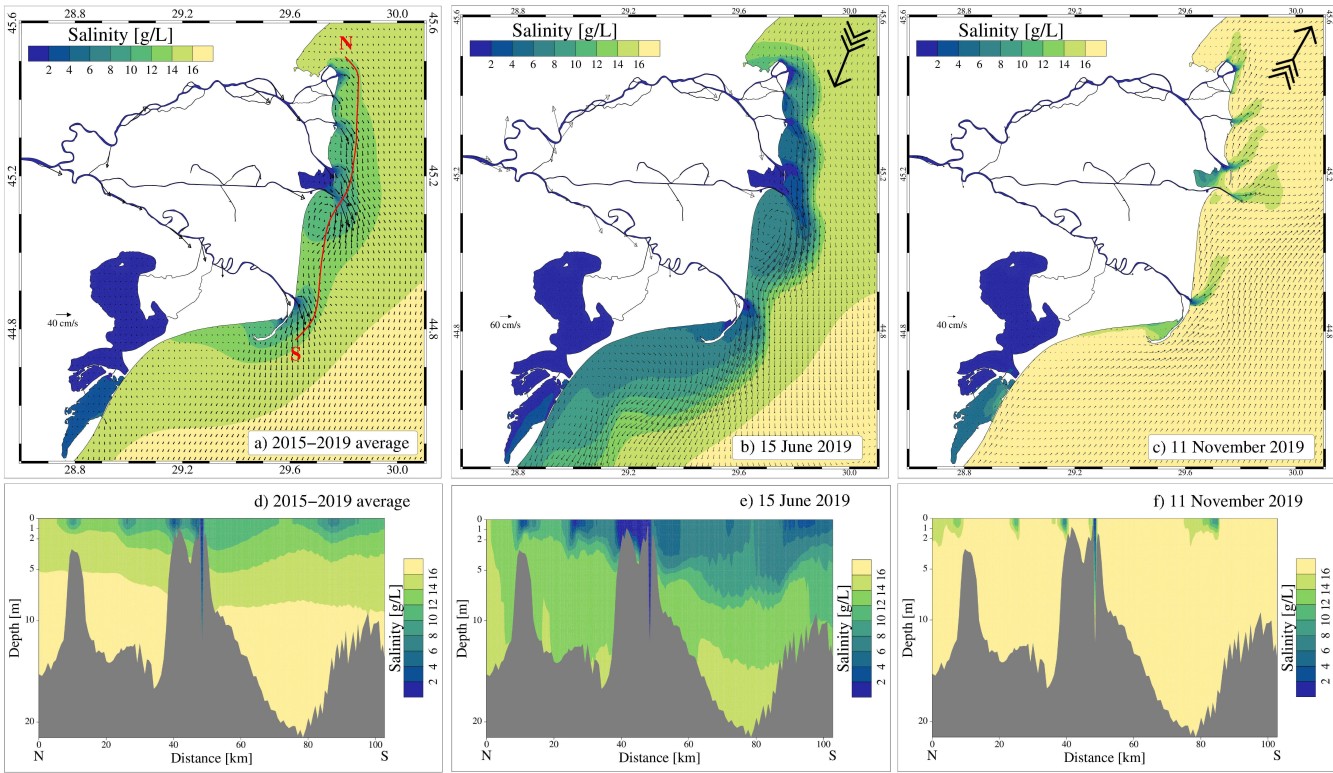

**Figure 5.** Surface salinity and current velocity maps, and N-S salinity transects: a) and d) average values over the 2015-2019 period; b) and e) instant values on 15 June 2019; c) and f) instant values on 11 November 2019. The arrows in the top right corner of panels b and c indicate the wind direction.

north-eastward for a few km from the river mouths. During such an event, the water column in front of the delta is well mixed
except for a surficial 2 m thick layer in front of the main river branches (Fig. 5f).

The analysis of the sea temperature results during events of southerly wind revealed the presence of small scale near-shore patterns located between the river mouths and having thermo-haline characteristics different from the surrounding areas (Fig. 6). The vertical alongshore sea temperature transect presented in Figs. 6c and Figs. 6d indicate an upwelling-driven transport of marine waters from deeper layers to the coastal zone, enhancing mixing between open sea and riverine waters.
The presented analysis indicates that these peculiar features are generated by upwelling processes induced by the action of southerly winds blowing along the coastline and interacting with the river outflow.

To analyse the temporal variability of the coastal dynamics, the model results were processed to computed the standard deviation (hereinafter STD) of the month of all years belonging to the four seasons (winter=DJF, spring=MAM, summer=JJA, fall=SON). The surface current variability is higher in winter (Fig. 7a) and spring (Fig. 7b) with STD values above 0.5 m s$^{-1}$ in
a coastal strip extending from the Sulina mouth down to the end of the Sahalin spit. During summer (Fig. 7c) and fall (Fig. 7d) the highest current velocity variability is found south of the Sf. Gheorghe mouth. The highest variability in the surface salinity

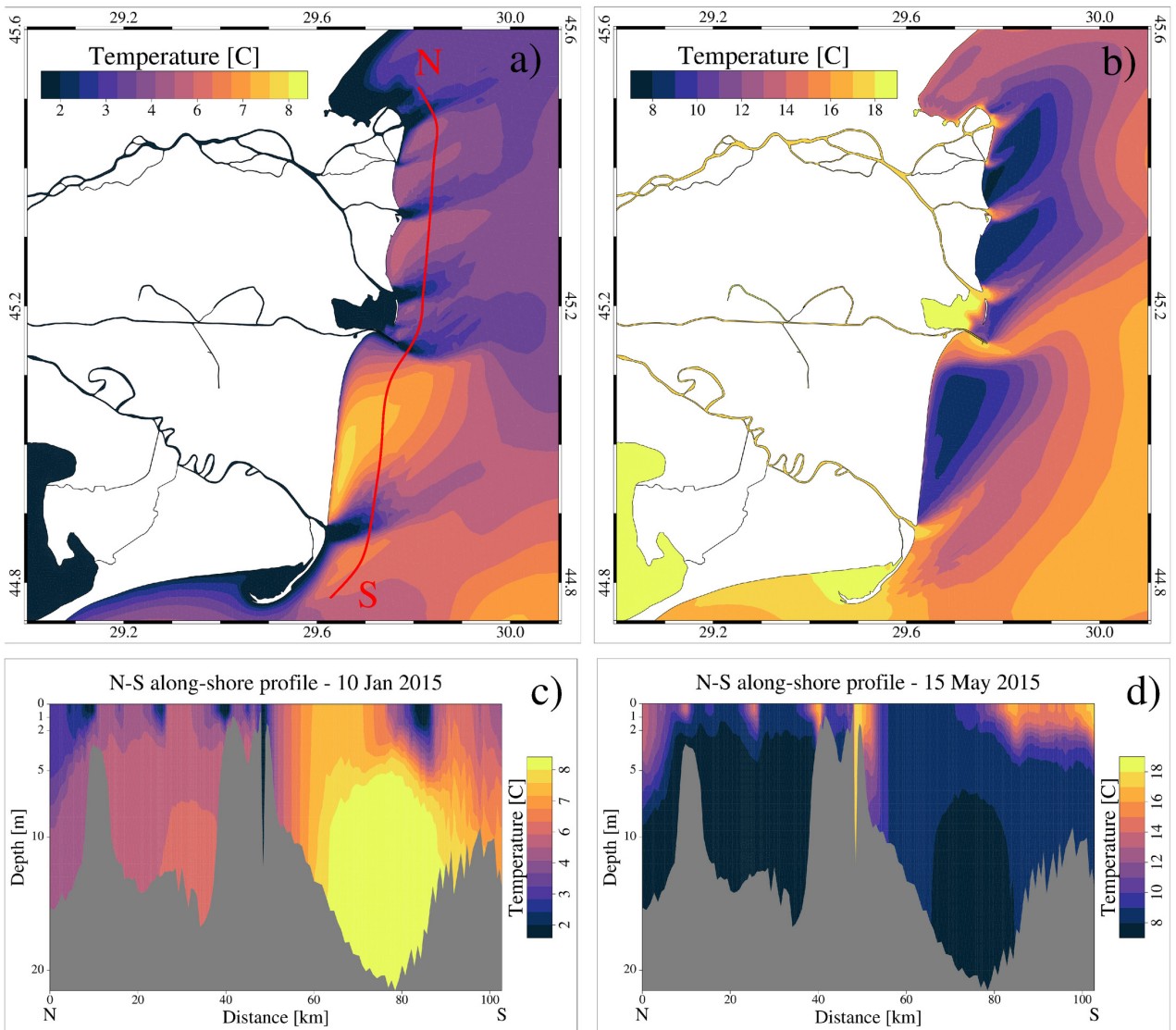

**Figure 6.** Map of sea surface temperature and north-to-south alongshore transect of sea temperature for the 10 January 2015 (a and c) and 15 May 2015 (b and d). The transect location is indicated with a red line in panel a.

is found during spring (Fig. 7f) and summer (Fig. 7g) months with STD values above $3\,\mathrm{g\,L^{-1}}$ characterizing large areas in front of each river mouth and a large coastal band south of the delta. The freshwater discharged by the different branches determine a similar coastal salinity pattern in winter (Fig. 7e) and fall (Fig. 7h). These findings can be explained by the variability of the

Danube River discharge, that usually peaks in spring or early summer, while drought conditions are generally found in autumn (Fig. 2a), and the winds (either northerly and southerly), that are generally stronger in winter and autumn (Bajo et al., 2014).

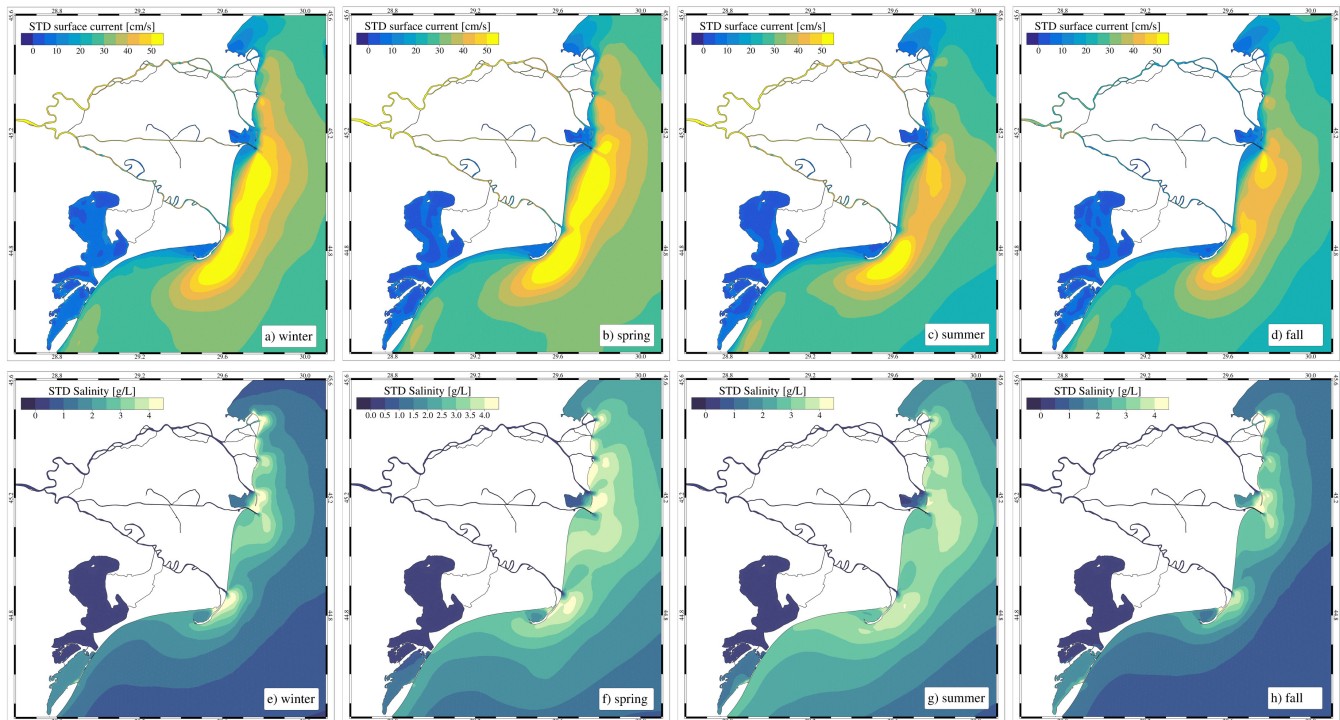

**Figure 7.** Seasonal standard deviation of surface currents (panels a, b, c and d) and surface salinity (panels e, f, g, and h).

## 3.3 Lagoons' water exchange, mixing and renewal capacity

The Razelm Sinoie Lagoon System is a transitional coastal environment connected to the Danube River and the Black Sea. The lagoon's circulation is influenced by the freshwater inflow, the coastal sea level and the wind action over the system. Such dynamics are well illustrated in Figure 8, which presents daily values from 2018 for Danube River discharge, wind speed and direction in the RSLS, sea-lagoon (Sinoie) water level differences, and total water and salt fluxes through the Edighiol and Periboina inlets.

The water level in the Sinoie Lagoon is generally higher than in the coastal area particularly during flood river conditions (e.g., from March to May 2018). However, the model results show high temporal variability induced by the wind action over the lagoons and the coastal sea. It must be noted that the water flux is not linearly dependent on the water level gradients confirming that the flow between the lagoon and the sea is limited by the transport capacity of the Edighiol and Periboina inlets (ad example at the beginning of March). A two-layers flow in the Edighiol inlet may occur when the lagoon-sea water level gradient is small (in the order of a few of cm).

The freshwater inflow from the Dunavăţ and Dranov canals creates, on average, a persistent water level gradient from Razelm Lagoon to Sinoie Lagoon and the adjacent coastal sea (black line in Fig. 9). The water level jumps between the two lagoons and between the Sinoie Lagoon and the open sea indicate that the water exchange between the different water bodies

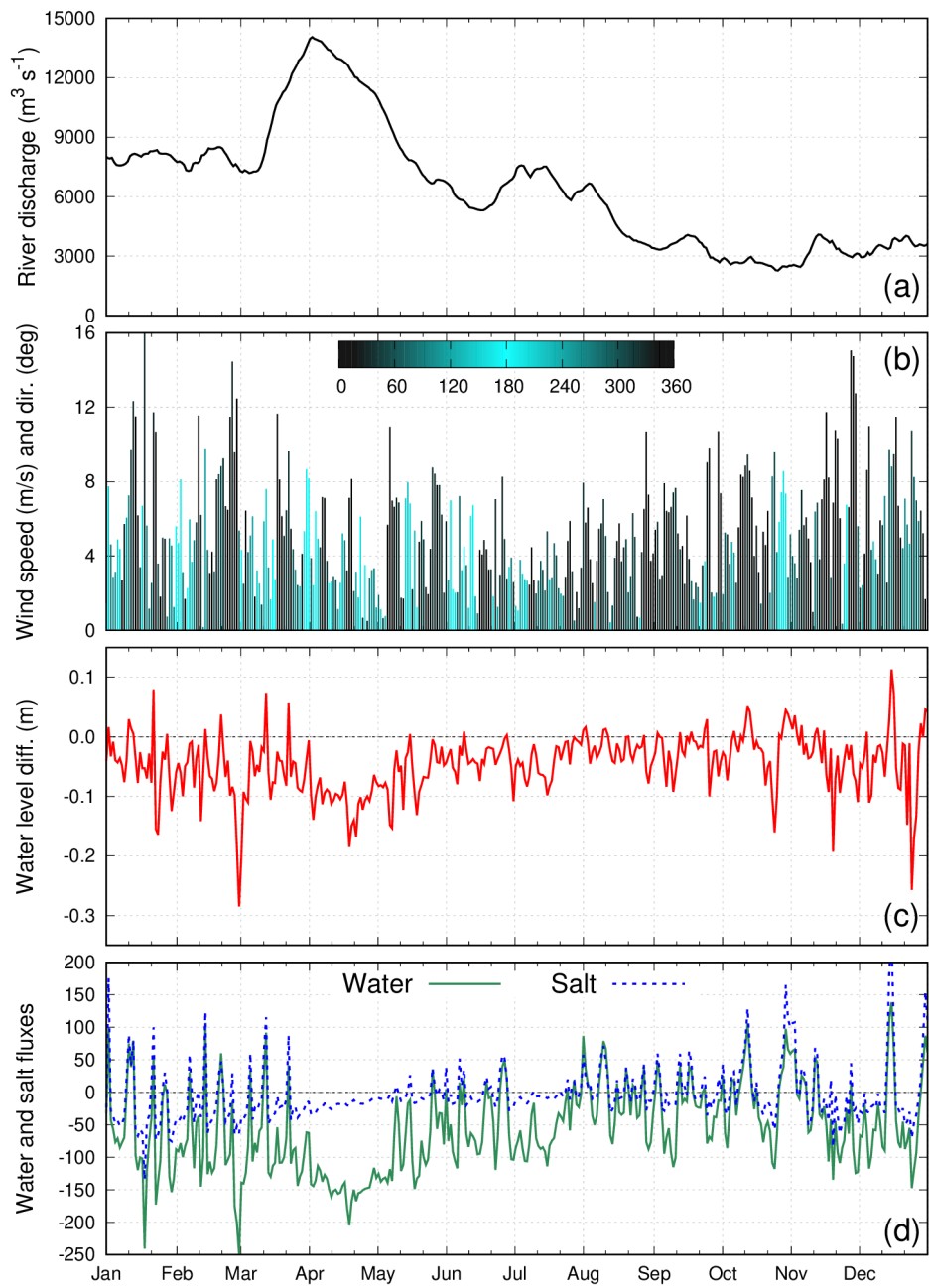

**Figure 8.** (a) Daily values for the year 2018 of Danube River discharge, (b) wind speed and direction ($0°$ means a northerly wind and $180°$ indicates a southerly wind) in the RSLS, (c) simulated sea-lagoon (Sinoie) water level difference, (d) simulated sea-lagoon water (in $m^3\ s^{-1}$) and salt (in $10^{-1}\ kg\ s^{-1}$) fluxes. Positive values of water and salt fluxes indicate inflow into the lagoons, while negative values indicate outflow from the lagoon to the sea. Model results are from the reference simulation.

is limited by the transport capacity of the narrow and shallow connecting canals (Canal 2, Canal 5, Edighiol and Periboina inlets). The internal average north-to south sea level gradient found into both lagoons is determined by the dominant winds from north-easterly direction.

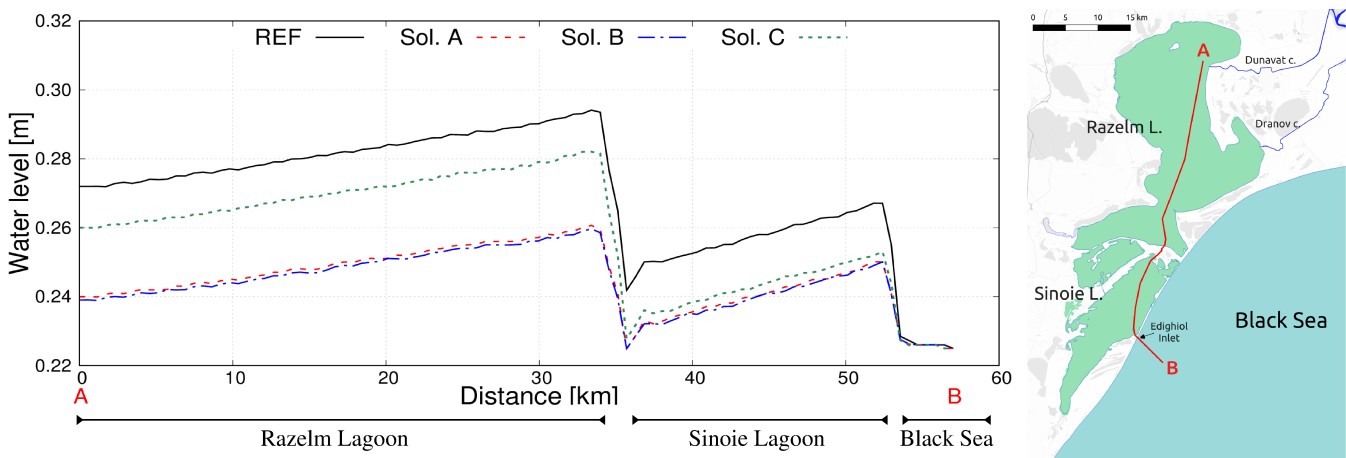

**Figure 9.** Average (over 2015-2019 period) water levels along the AB transect crossing the Razelm Sinoie Lagoon Systems indicated with a red line in the right panel. Background: ©OpenStreetMap contributors 2024; distributed under the Open Data Commons Open Database License (ODbL) v1.0.

The RSLS has an average water volume of about 1,300 millions m$^3$ and receives 40 and 22 m$^3$ s$^{-1}$ of freshwater from the Dunavăț and Dranov canals, respectively. This excess water entering the lagoons is primarily discharged into the Black Sea via the Edighiol and Periboina inlets, resulting in a average seaward flow of 58 m$^3$ s$^{-1}$. The average inflow of marine water into the RSLS amounts to 16 m$^3$ s$^{-1}$. Evaporation over the lagoon system overpasses precipitation resulting in a net loss of 20 m$^3$ s$^{-1}$. The lagoons receive a total water flux of 78 m$^3$ s$^{-1}$ from the sea and the river. The average fluxes are reported in Table

2.

**Table 2.** Average water fluxes (in m$^3$ s$^{-1}$) between the lagoons and the sea via the Edighiol and Periboina inlets and the new inlets (positive values indicate inflow into the lagoons, while negative values indicate outflow from the lagoon to the sea).

| Scenario | Edighiol and Periboina inlets | | | New inlet | | | Total flow | | |
|---|---|---|---|---|---|---|---|---|---|
| | Net | Outflow | Inflow | Net | Outflow | Inflow | Net | Outflow | Inflow |
| REF | -42 | -58 | 16 | - | - | - | -42 | -58 | 16 |
| A | -21 | -43 | 22 | -23 | -49 | 26 | -44 | -92 | 48 |
| B | -20 | -42 | 22 | -23 | -51 | 28 | -43 | -93 | 50 |
| C | -24 | -45 | 21 | -19 | -28 | 9 | -43 | -73 | 30 |

     The basin-wide water flushing time, computed dividing the volume by the incoming fluxes, is 193 days. The flushing time estimate was used to determine the duration of the water renewal time simulations. In this work, we performed five one-year-

long replicas of WRT starting the simulations at the beginning of each year. The dispersion and dilution of the tracer initially released into the lagoons (see section 2.1) are determined by the inflow of new water and internal mixing processes that in the shallow lagoon system are mainly induced by the wind. The average (over the five replicas) basin-wide WRT is 241 days (minimum = 181 days in 2018; maximum = 333 days in 2017; standard deviation = 63 days) for the RSLS, thus revealing a mixing efficiency (determined as the ratio between WFT and WRT) of 0.8.

The spatial and temporal variability of river, ocean and meteorological conditions affects the river-lagoon-sea fluxes as well as the internal mixing in the lagoons, and consequently the WRT computation. Indeed, the difference in WRT across the different years primarily reflects the freshwater input into the lagoons that mainly drives the river-lagoon-sea fluxes and therefore the flushing of the lagoon waters. Indeed, the minimum (181 days) and maximum (333 days) basin-wide WRT values are found in the flood (2018) and drought (2017) years, respectively (Fig. 2). A secondary, but not negligible, role is played by the wind which, in 2018 was characterized by frequent and intense Northerlies that enhanced internal mixing and favored the outflow from the lagoon towards the sea. Spatially, a marked east-to-west WRT gradient (from 50 to more than 300 days) is evident in the RSLS, with the Razelm Lagoon having lower values than the Sinoie Lagoon (Fig. 10a). This is because the new (fresh) waters enter the Razelm Lagoon from the Dunavăţ and Dranov canals and are subsequently mixed and transported to the Sinoie Lagoon. The input of marine waters through the Edighiol and Periboina inlets has a limited effect on the local WRT, which resulted mostly influenced by the outflow of tracer. Salinity has a limited variability over the RSLS, with values ranging from 1 to 8 g L$^{-1}$ and where the higher values are found in area of the Sinoie Lagoon near the Edighiol and Periboina inlets (Fig. 10e).

### 3.3.1 Assessment of the impact of lagoon-sea reconnection solutions

As illustrated in previous section, the RSLS is a large and shallow water body separated from the sea by narrow sandy barriers and with limited renewal capacity. In this study, we use the modelling setup for the period 2015-2019 to evaluate the effects of several reconnection solutions (Fig. 1c) on the river-lagoon-sea exchange, water renewal time and salinization. The numerical model results of the different simulations were processed to estimate the lagoon-sea water exchange through the inlets (the two existing and the newly designed) and the values are reported in Table 2. The reference run presented in previous sections is used as a basis for comparison for the *what-if* scenarios.

Opening a new inlet has a significant effect on the lagoon hydrodynamics altering the water budget of the basin and the fluxes through the existing Edighiol and Periboina inlets, which generally resulted to be enhanced inflow into the lagoon and reduced outflow out of the lagoon (Table 2). The net flow between the lagoons and the sea is not significantly altered (about 40 m$^3$ s$^{-1}$), being mostly determined by the river inflow into the lagoon. However, opening a new inlet increases up to four times the total inflow of marine waters into the lagoon with the respect of the reference simulation, with the solutions planned for the Razelm Lagoon (A and B) having a higher effect on the fluxes than the ones designed in the Sinoie Lagoon (C).

Connecting the Razelm Lagoon with the Black Sea with solutions A and B do not only allow the inflow of marine waters but also changes the water level of the two basins (red and blue lines in Fig. 9) decreasing the water exchange between the two

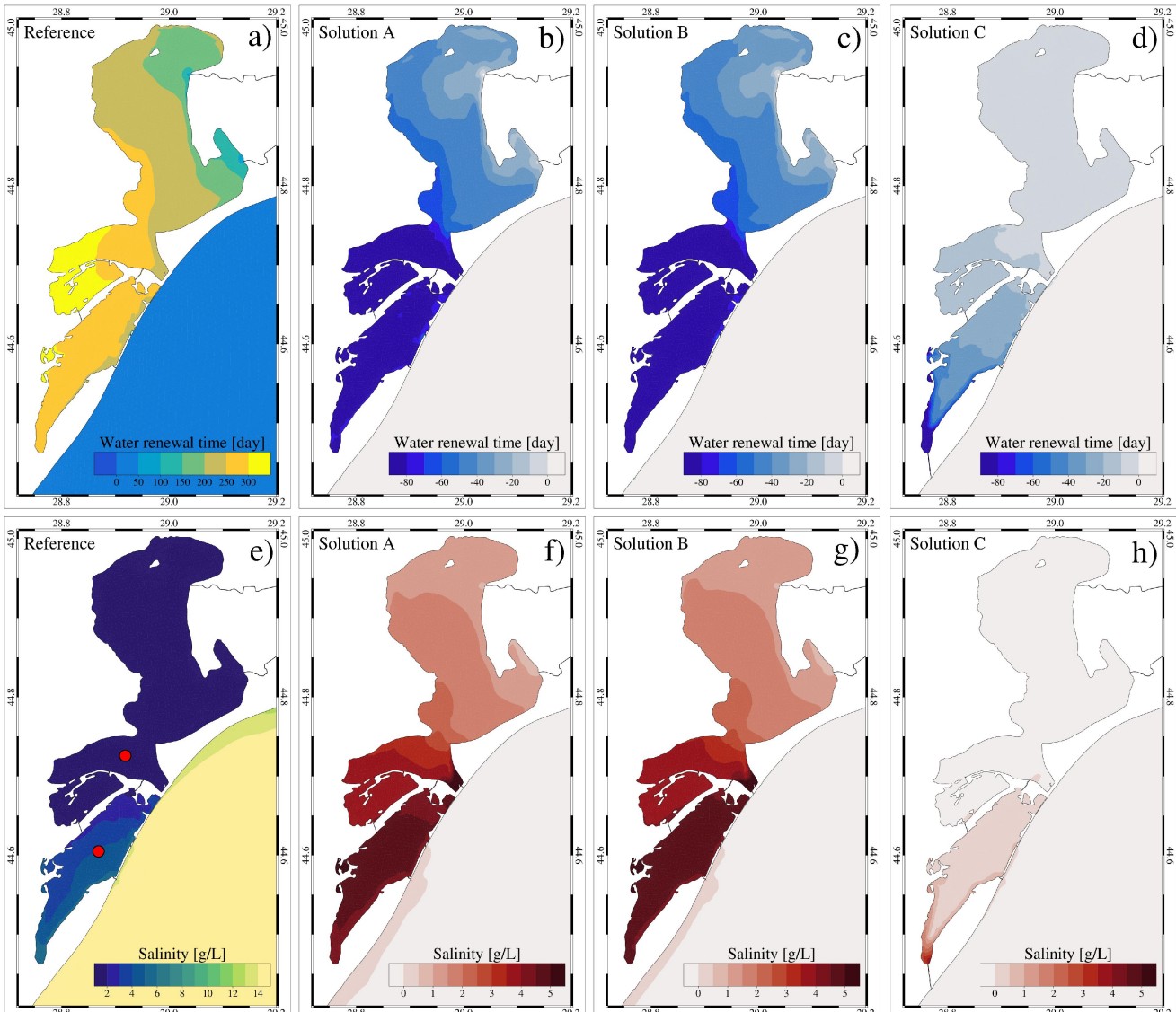

**Figure 10.** Average water renewal time and salinity over the Razelm Sinoie Lagoon System for the reference simulation (as absolute values) and the reconnection scenarios A, B and C (as difference with respect to the reference run). The red dots in panel *e* indicate the location of the two control points where the salinity timeseries were extracted (Fig. 11).

lagoons and the outflow via the existing inlets (Table 2). On the contrary, solution C (green line in Fig. 9) affects water levels mostly in the Sinoie Lagoon and the fluxes with the Black Sea but has a limited impact on the Razelm basin.

Because of these changes in the water level and fluxes, the water renewal capacity and the salinity increase (lower water renewal time values). The average WRTs decrease to 191, 190 and 230 days in scenarios A, B and C, respectively. The spatial
distributions of WRT illustrated in Fig. 10 clearly reflect the changes in the lagoon-sea fluxes. Therefore, solutions A (Fig. 10b)

and B (Fig. 10c), are the ones determining a more significant decrease in the water renewal times, especially in the southern part of the Razelm Lagoon and in the Sinoie Lagoon where the WRT values decrease by up to 80 days with respect to the REF simulation. Solution C (Fig. 10d) has a moderate effect on WRTs, which is anyway limited to the southern past of the Sinoie Lagoon.

The augmented inflow of marine waters through the existing and the new inlets determines a general increase in salinity in the southern part of the lagoons. The highest salinity changes with respect to the REF simulation (Fig. 10e) are found in scenarios A (Fig. 10f) and B (Fig. 10g) in the Sinoie Lagoon where the average salinity increases to more than 5 g $L^{-1}$. As for the water renewal times, solution C (Fig. 10h) has a limited effect on salinity.

To investigate more into details the opening effects on the salinity, we extracted from the simulation results the timeseries

in two control stations in the Razelm and Sinoie lagoons identified with red dots in Fig. 10e (Fig. 11). Solutions A and B have very similar effects on salinity which fluctuates between 2 and 8 g $L^{-1}$, and between 6 and 16 g $L^{-1}$ in the Razelm and Sinoie lagoons, respectively. Lastly, solution C has an almost negligible (< 1 g $L^{-1}$) effect on salinity in both lagoons.

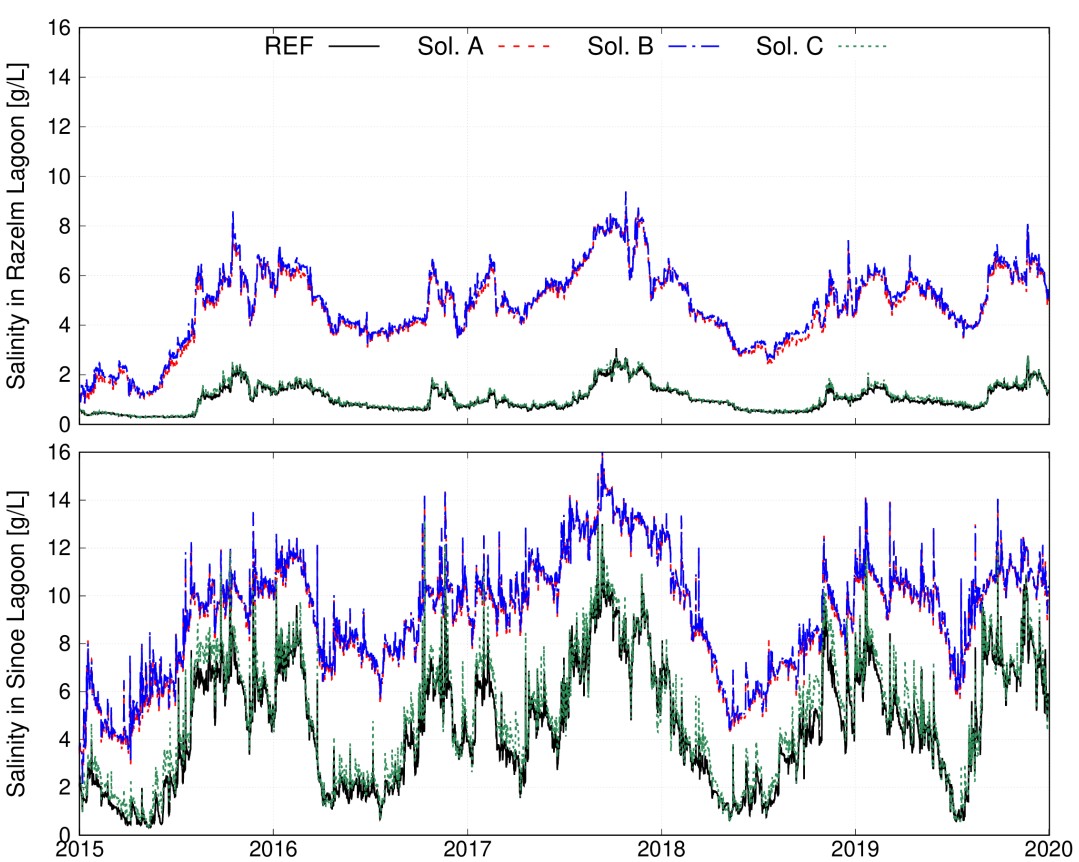

**Figure 11.** Timeseries of modelled salinity extracted in the control points in the Razelm (top panel) and Sinoie (bottom panel).

## 4    Discussion

The Danube Delta, like many other coastal systems at the river-sea interface, is composed of several interconnected water bodies (river branches, coastal lakes, lagoons, shelf sea) having different physicochemical characteristics and influencing each other (Passalacqua, 2017). Here we interpret how cross-scale hydrodynamics and water mass exchanges between river branches, lagoons, and the coastal sea shape renewal and mixing in the Danube Delta, and we assess how these findings generalize to other river-sea systems. Finally we review the implication of the reconnection solutions on the connectivity between the different water bodies.

### 4.1    River-sea modelling characteristics, requirements and limitations

Numerical modelling may play a crucial role in guiding evidence-based efforts to protect and sustainably manage coastal transitional systems, such as estuaries, deltas and lagoons. However, to represent the dynamics inside each water basin, as well as the water exchange among them, the numerical tools need to have specific characteristics:

- the representation of the whole land-ocean area through a numerical domain that extends upstream into the river network and offshore into the open sea. Such an approach has been widely adopted to model processes in several deltas and estuaries. Some model applications are saltwater intrusion and compound floods in the Pearl River Delta (China) (Shen et al., 2018; Zhang and Yu, 2025), hydrodynamics and saltwater intrusion in the Po Delta and adjacent coastal area (Italy) (Maicu et al., 2018; Bellafiore et al., 2021), river plume dynamics in the Columbia River (USA) (Vallaeys et al., 2018), flooding in the Mekong Delta (Vietnam) (Thanh et al., 2020), hydrological connectivity in the Wax Lake Delta (USA) (Feizabadi et al., 2024), tidal intrusion in the Ganges-Brahmaputra-Meghna mega delta (Bangladesh) (Bricheno et al., 2016), among others. Similarly, our cross-scale SHYFEM hydrodynamic model implementation allowed the full representation of the Danube river-sea continuum by comprising the delta river network, the Razelm Sinoie Lagoon System and part of the Western Black Sea shelf area. Moreover, it is crucial, at least for the case of the Danube Delta, to represent also the narrow (about 50 m wide) channels connecting the different water compartments (river to river, river to lagoon, lagoon to lagoon, lagoon to sea).

- an adequate horizontal resolution (at least in the order of a few tens of meters) to correctly reproduce the aforementioned river-sea continuum and the complex morphology that often characterizes deltas and estuaries. Particular attention must be paid to describing narrow channels, secondary river branches, and areas subject to periodic wetting and drying, such as tidal marshes and river floodplains. To meet this constraint and limit computation time, unstructured or flexible mesh models with variable element resolution, such as the SHYFEM model adopted in this study, are preferable to structured mesh models (rectangular grids) (Teng et al., 2017; Umgiesser et al., 2022). The variable model resolution is of fundamental importance for reproducing the complex morphology of the Danube Delta realizing a seamless transition between different spatial scales, from river branches to the coastal sea. As presented in section 3.2, a relatively high horizontal resolution (in the order of a few hundreds of meters) is required in the coastal sea in front of the river mouths to properly capture the plume dynamics. In contrast, most Black Sea regional models (e.g., Lima et al., 2020; Miladinova

et al., 2020) have a coarse resolution (> 2 km) and a simplified representation of the river inputs that does not allow the description of the complex coastal circulation patterns highlighted in Figs. 5 and 6.

- a three-dimensional baroclinic approach for representing vertical stratification induced by temperature and salinity gradients. While a vertically integrated barotropic model is generally sufficient for reproducing the circulation along the river course and in shallow coastal environments, a 3D approach must be adopted in coastal environments influenced by freshwater (Horner-Devine et al., 2015). This is certainly the case of the coastal area in front of the Danube Delta, where, as shown in the transects included in Figs. 5 and 6, the freshwater outflow determines a stratified water column. A 3D modelling approach, coupled with a turbulence closure model, is also essential for accurately capturing the mixing of different water masses and the coastal upwelling that occurs during strong wind events (Figs. 5 and 6).

- a robust validation of the model results. Even if Danube Delta is poorly monitored and we had limited (in space and time) observational datasets, the analysis presented in section 2.3 demonstrates that our SHYFEM model application correctly reproduces hydrodynamics in the different water compartments of the Danube Delta. The model validation could be further enhanced with the availability of additional future observations, particularly river discharge and salinity data. In light of these limitations, there is a clear need for an transnational integrated observation system that combines all available monitoring networks managed by academic and research institutions, national and regional environmental protection agencies and local communities. In this context, the pan-European Research Infrastructure DANUBIUS-RI (the International Centre for Advanced Studies on River-Sea Systems, http://www.danubius-ri.eu/) is building integrated infrastructures - made of observational networks, modelling and forecasting systems, and living laboratories - on ten of the major European river-sea systems, one of which is the Danube Delta (De Pascalis et al., 2025).

## 4.2 Processes driving the exchange between water bodies

The exchanges of water between the different interconnected water compartments of the delta are regulated by barotropic and baroclinic processes driven by the forcing acting on the area: upstream river discharge, wind stress, heat fluxes, open sea conditions.

The water flow in the river is predominantly governed by advection, and the redistribution of water over the distributaries is mainly determined by the branch geometry and hydraulic roughness. Nevertheless, the distribution of water discharge can vary considerably under different flow regimes (Maicu et al., 2018; Constantinescu et al., 2023). Our simulation results indicate that the standard deviation of the relative discharge among the Danube Delta branches remains below 0.7 %. Such a low temporal variability can be partially attributed to the omission of the delta floodplain system - comprising channels, wetlands, lakes, and marshes - from the computational domain. Floodplains play a critical role in the Danube Delta by providing flood control, water purification, groundwater replenishment, and habitat for diverse species like fish and birds (Frank et al., 2025). The representation of flood dynamics over floodplains (Ciobotaru et al., 2025) in our modelling system would necessitate further technical development (e.g., a more accurate wetting and drying algorithm) and is beyond the scope of this study.

In the coastal area in front of the Danube Delta, the presence of multiple freshwater outlets generates multiple buoyant fluxes that interact laterally modulating coastal mixing (Fig. 5). A similar configuration is relevant to many of the world's major river deltas (e.g., the Mississippi and the Nile; Horner-Devine et al., 2015) and coastal settings with multiple river mouths in close proximity (Warrick and Farnsworth, 2017). The interaction between stratified waters belonging to river plumes and the wind force determines complex coastal dynamics. During northerly wind conditions, the freshwater plumes are constrained close to the coast and advected to the south. Southerly winds that promote coastal upwelling cause the plumes to thin and be advected offshore due to Ekman transport (Fong and Geyer, 2001). During these events, upwelling - rather than horizontal advection - generates small-scale nearshore patterns between the river mouths, characterized by surface temperatures that are either warmer or colder than offshore waters, depending on the season. Similar patterns were found by Bellafiore et al. (2019) in front of the Po River Delta.

In addition to the river influence on the coastal dynamics, during drought periods marine waters can intrude the lower part of the river course, altering the water's physicochemical properties. Such a phenomenon, known as saltwater intrusion, is driven by a three-dimensional estuarine dynamics that determines freshwater flowing on the surface layers and the salt wedge intruding along the riverbed (Valle-Levinson, 2010). According to the reference simulation results, marine waters enter up to 20 km upstream from the mouth in the Chilia and Sulina branches, and up to 7 km upstream from the Sf. Gheorghe mouth. While saltwater intrusion poses a serious threat to several coastal areas compromising freshwater supplies for agriculture and human use (Li et al., 2025), it has not yet been reported as a major issue in the Danube Delta. The situation is predicted to worsen in the near future due to sea level rise (van de Wal et al., 2024) and decreasing summer runoff (Probst and Mauser, 2023; Stolz et al., 2025).

The water masses of the Razelm Sinoie lagoons, as many shallow coastal water systems (Umgiesser et al., 2014), are generally vertically well mixed by the action of the wind. Consequently, the long-term net water transport in the RSLS is mainly barotropic and directed from the Danube River to the Razelm Lagoon (via the Dunavăţ and Dranov canals), then to the Sinoie Lagoon (via canals 2 and 5), and finally to the open sea (via the Edighiol and Periboina inlets). While the water flow from the river to the lagoons is unidirectional, a bidirectional flow characterizes the water exchange between the lagoons and the Black Sea. Marine waters flow into the lagoon when sea level is higher than the lagoon water level. The barotropic flow of marine waters into the lagoons occurs mostly during autumn and winter drought and windy conditions. As a result, salinization events of the lagoon environments are sporadic and have a general duration of a few days. The salt content entering the Edighiol and Periboina inlets is advected and diluted in the Sinoie Lagoon and sporadically reaches the Razelm basin (Fig. 8a). The barotropic water exchanges with the river and the sea regulate the renewal capacity of the lagoons, while wind stress modulates internal mixing. Due to the limited degree of water exchange, the Razelm-Sinoie lagoons can be classified as a choked water body, according to Kjerfve (1986). Wind stress actively promotes water circulation within the lagoons, making the RSLS a well-mixed water body, according to Umgiesser et al. (2014).

## 4.3 Anthropogenic influence on river-sea systems

The Danube Delta, as many other coastal transitional systems (Maselli and Trincardi, 2013), has been heavily affected by human interventions (Nichersu et al., 2025, and references therein). Among river-sea systems, lagoons are highly productive areas that support numerous industrial, commercial, and recreational activities (Pérez-Ruzafa et al., 2019). Similarly, the RSLS is subject to economic interests related to fisheries, agriculture, and water-based tourism.

Numerical models have been widely used to evaluate the impact of anthropogenic interventions on coastal environments throughout the simulation of *what-if* scenarios (Ferrarin et al., 2013; Umgiesser, 2020; Hariharan et al., 2023; Kolb et al., 2022, among others). In the RSLS, efforts to enhance ecological status and improve water circulation have prompted exploration into the potential impacts of creating a new inlet to strengthen the lagoon's connection with the sea. The findings discussed in section 3.3.1 suggest that even a localized morphological modification can significantly influence the overall hydrodynamics of the lagoon system. Introducing a new inlet leads to a reduction in water renewal time, which helps mitigate stagnation and enhances ecological conditions. However, it also results in elevated salinity levels. While fisheries and tourist activities would benefit from this intervention, the increased salinization of the lagoon's waters poses a considerable risk to agricultural freshwater resources. To help local authorities and communities manage these issues, the model will next be used to explore several alternative lagoon-sea reconnection solutions.

## 5 Conclusions

This work presents the first cross-scale hydrodynamic model implementation covering the entire Danube Delta to investigate the river-sea continuum. To study the hydrodynamic processes driving water exchange and connectivity among the various interconnected water compartments of the delta, the 3D unstructured hydrodynamic SHYFEM model was applied to a domain representing the river network, coastal lagoons, and part of the shelf sea. A multi-year (2015-2019) hindcast simulation was performed using observational data and reanalysis fields as forcings. The developed model was validated by comparing various parameters with the available observations.

The simulation results enabled the quantification of riverine discharge distribution among the major branches (Chilia: 46.3 %, Sulina: 29.7 %, and Sf. Gheorghe: 25.7 %) and the distributaries of the delta river network, thereby characterizing the relative significance of the nine river mouths. Such a detailed description of the freshwater outflow into the Black Sea enabled the investigation of spatial and temporal variability in the main oceanographic processes occurring offshore of the Danube Delta. River inputs create a stratified water column along the coast, with Black Sea waters typically found at depths greater than 5 to 10 meters below the surface. The region of freshwater influence is shaped by the multiple buoyant fluxes forming coalescing river plumes, and alongshore winds. The predominant northerly wind regime sustains a southward coastal current that enhances the southward propagation of the freshriver plumes. Southerly wind conditions favour coastal upwelling, which enhances the offshore propagation of river plumes and creates entrapped marine water regions between them.

On average, approximately 2,000 million m$^3$ of water per year flows from the Danube River into the Razelm Lagoon via the Dunavăț Dranov canals. This input of freshwater creates a north-to-south water level gradient in the Razelm Sinoie Lagoon

System, which determines a barotropic flow from the Razelm Lagoon to the Sinoie Lagoon and ultimately to the Black Sea. The inflow of marine waters into the Sinoie Lagoon occurs sporadically during southerly wind events that induce a positive sea-to-lagoon water level gradient. Water exchanges with the Danube River and the Black Sea determine the flushing capacity of the RSLS, while wind is the primary driver of horizontal and vertical mixing of the lagoon waters. These processes are accounted for in the simulation of a passive Eulerian tracer, which allowed the estimation of the water renewal time in this interconnected coastal environment. The average water renewal time of the RSLS is 241 days, with marked interannual variability mainly driven by freshwater input into the lagoons and the wind conditions.

Lastly, the numerical model was used to assess the potential impacts of opening a new inlet (with three different configurations) designed to improve the river-sea-lagoon connectivity. Opening a third connection to the Black Sea would enhance the water exchange between the RSLS and the sea, thereby increasing water flushing and reducing the water renewal time to as slow as 190 days (solutions B). At the same time, the augmented inflow of marine waters significantly increased salinity in the southern and central parts of the RSLS (up to $5 \, \text{g} \, \text{L}^{-1}$). We demonstrated that this modelling system is a powerful tool for efficiently evaluating the effects of human interventions in the coastal environment.

The applied numerical model and implementation approach are easily exportable to other river-sea environments and can be further developed to support decision-making. An operational version of the Danube Delta model is currently under implementation to provide forecasts that enhance awareness and preparedness for weather-related risks. The simulated *what-if* scenarios and forecasting system will form the core of the first digital twin of the Danube Delta.

*Code and data availability.* The community SHYFEM hydrodynamic model is open source (GNU General Public License as published by the Free Software Foundation) and freely available through GitHub at https://github.com/georgu/shyfemcm-ismar.

This study has been conducted using the following public available datasets: the Black Sea Physics Reanalysis (https://doi.org/10.25423/CMCC/BLKSEA_MULTIYEAR_PHY_007_004); the Copernicus European Regional ReAnalysis (https://doi.org/10.24381/cds.622a565a); the 2022 European Marine Observation and Data Network bathymetry (https://doi.org/10.12770/ff3aff8a-cff1-44a3-a2c8-1910bf109f85); in situ sea level and sea temperature data (https://marineinsitu.eu/dashboard/); the Danube River temperature data generated by the Black Sea Catchment model developed by Deltares in the EU project DOORS (https://doi.org/10.5281/zenodo.15675190); The following datasets are not public available and can be requested to the mentioned authorities: the National Institute of Hydrology and Water Management of Romania for the Danube River discharge data; the University of Stirling (UK) for satellite sea surface temperature data; GeoEcoMar (RO) for the 2024 Razelm Sinoie Lagoons, the 2019 Sulina branch and the 2016-2017 Sf. Gheorghe branch bathymetric datasets.

*Author contributions.* CF conceived the idea of the study with the support of AS. ID and CF collected the bathymetric and validation data sets. AS designed the reconnection solution to improve river-lagoon-sea hydrological connectivity. CF and APH performed the numerical simulations and analysed the results. All authors discussed, reviewed and edited the manuscript.

*Competing interests.* The authors declare that they have no conflict of interest.

*Acknowledgements.* The authors wish to thank the National Institute of Hydrology and Water Management of Romania for providing river
discharge data at Ceatal Izmail; the European Union's Horizon 2020 DOORS project, grant agreement number 101000518, in particular
Jos Van Gils, Hélène Boisgontier and Sibren Loos (Deltares, NL) for providing Danube River temperature data (https://doi.org/10.5281/
zenodo.15675190); Nagendra Jaiganesh Sankara Narayanan, Andrew Tyler and Evangelos Spyrakos from the University of Stirling (UK)
for providing satellite sea surface temperature data; the Lower Danube River Administration for providing 2023 bathymetric data for Chilia
Branch. The bathymetry of the Razelm Sinoie was updated during 2024 activities in the Romanian UESFISCDI "DANUBE4all Support"
project (77PHE/ 2024). This study was conducted as part of the DANUBE4all (Restoration of the Danube River Basin for ecosystems and
people from mountains to coast; ID 101093985; https://www.danube4allproject.eu/) and iNNO SED (iNNOvative SEDiment management
in the Danube River Basin; ID 101157360; https://innosed.eu/) projects funded by the European Union under EU Mission "Restore our
Ocean and Waters". This work is part of the activities of the scientific community that is building the pan-European Research Infrastructure
DANUBIUS-RI - The International Centre for Advanced Studies on River-Sea Systems (http://www.danubius-ri.eu/).

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
