# Peer review of "Modelling river-sea continuum: the case of the Danube Delta"

_EGUsphere, 2025_

## Referee Comment (RC2)

Review of the paper Modelling river-sea continuum: the case of the Danube Delta, by Christian Ferrarin, Debora Bellafiore , Alejandro Paladio Hernandez, Irina Dinu and Adrian Stanica.

The paper introduces a rather unique SHYFEM configuration that is integrating the Danube delta and the RSLS lagoon system into a coastal model for the western Black Sea. This unified approach to modelling the land-estuary-sea continuum is demanding in terms of model numeric and physics. The authors have carried out a thorough study and achieved good and relevant results that were used to estimate the water transport and hydrographic conditions in the Danube delta system.

The following includes some general comments, followed by a more detailed review of the paper.

General points:

The authors carried out a thorough research study, but could have done more to analyse the data in a more comprehensive way targeting well-defined research questions. I am missing a consistent storyline leading through the paper. This is the case for both the entire paper and individual chapters. As it is, the authors have basically written a general study of the Danube river and Razelm Sinoie Lagoon System (RSLS), which provides a lot of information, but is not embedding it into a consistent story-line. The manuscript is actually two papers in one, each with its own research question. The first paper is (1.) demonstrating that the river-sea continuum can be modelled successfully using unstructured grid models and (2.) the second paper is studying the impact of openings of the RSLS towards the Black Sea and its impact on local circulation pattern and hydrographic conditions. I would suggest the authors to define one or two research questions and then to develop a story line addressing this questions. With this, they could streamline the whole paper and make it more concise.

Comments: Below, I list some ideas for the two paper ideas mentioned above. Of course, this is all hypothetical and intended only for providing some ideas.

In the first paper (1.), I would expect to find an analysis of the impact of coastal high-resolution configurations on river plume modelling: comparison with and without the Danube river model and RSLS lagoon model. The paper could demonstrate the effect of improved river-plume-modelling on the freshwater distribution, pollutant distribution, etc., with impact on the local hydrographic conditions. To be true, this has been done, but the analysis could have been further expanded and could have been presented with a view to answer the research question rather than to present model results. The analysis could also include a comparison of modelled river data with either observations or hydrological model results (E-hype, SWAT, etc.). I think E-hype climatology is freely available.

The subject of the second paper idea (2.) mentioned above, the study of the effects of openings in the RSLS towards the Black Sea on the hydrographic conditions is only shortly covered in the paper. The analysis of impacts could be extended. The model results could be analyzed in the light of the objectives of the openings, which are not very clear to me. Here, too, I would prefer if the model results could be used to answer the research question rather than being presented.

The quality of the writing varies. I would strongly recommend to improve the orthography and grammar. Sometimes the construction of the sentences is not correct. Furthermore, the style is often rather direct, and focused on presenting facts. This is often done in loosely connected paragraphs, which could be better integrated.

The different measures of water transport and mixing: ROFI, WFT, WRT could be introduced in a combined and more consistent way in the method section. Currently there is only WRT defined, in the part of the

paper that is dealing with the SHYFEM model (Line 77-87). WFT is later on defined when using it. A clear definition for ROFI has not been provided. It would be good to define these quantities and how they are used in a consistent way. It is for example not clear until later, that the ratio of WFT and WRT is used. This should be done in a separate part of the method section, not in the model description.

In the following, I go through the paper from start to end and provide some more detailed comments. I will always provide the line number for reference.

Chapter 0. Abstract

In its current form, the abstract outlines the scope of the study. It could provide more motivation as to why this study has been carried out. The key findings should be listed as well. It is good to think of the abstract as a mini-IMRAD scheme, including all the parts of the paper.

Line 3: The sentence should end after "morphology". Then a new sentence should start.

Line 5: "The model was run for several years …" How many? "Several years" is a bit vague. ¨

Chapter 1. Introduction

Line 19: Think of a better begin of the sentence than "Modelling these coastal transitional water systems". Are these not "estuaries and coastal seas".

Line 31: I would say "… Danube delta, covering …" Please refer to Figure 1.

Line 32 "The manuscript …", maybe better: "The paper". You can also use the active voice and write "We focus …".

Note: This part of the introduction gives an overview of the scope of the paper. The different points could be used to identify the research question. The advantage of a good research question is that it describes the problem and the motivation for solving the problem.

Line 36: When we talk about what-if scenarios here, then they must at least described in general terms. It must also be explained what a what-if scenario is. It is not a commonly used term.

Chapter 1.1 The Danube Delta

Line 44: Please rewrite the sentence starting with "The Romanian part …"

Line 46: "extends on about", use "extends for about"

Line 50: correct: "connect Razelm lagoon with the …"

Line 51: "were finalized … " not "ended up"

Line 51: "As a result, more fresh water is discharged into the lagoon system"

Line 54: Please mark Portit or use another way to show it on the map. It is not good to say: "near reconnection option A", before these have been introduced. The same is true for "reconnection option C" in Line 58

Line 55: It's not intuitive that a coastal defense structure could enhance coastal erosion. This sentence could be reformulated or explained in more detail.

Line 62: The sentence starting with "Anyway" should be rewritten, like: As part of the master plan for the protection of the Romanian Littoral against erosion, a major hydraulic engineering project is currently implemented, to ensure a permanent water exchange through the Periboina Inlet. This is an example. The structure of many sentences in the document could be improved.

Line 65: What does "lower part" mean? Please re-write: "… average water discharge of [so much], with values ranging from …"

Chapter 2. Methods

Line 77-88: I would suggest to move this paragraph to a different part of the method section and to enhance it. I don't think it makes sense here in the SHYFEM related part of the document. The WRT parameter should be introduced after the model. The other variables ROFI, WFT should be introduced as well. The motivation for using these variables to study water transport through the lagoon system should be clear from the beginning.

Line 86: Does WRT really measure the time until the concentration fall to zero or is the time until they fall below a small value enough? I could imagine that it takes a long time until absolute zero is reached.

Line 86: I assume the water parcels are grid cells inside the investigated water body, i.e. the lagoon.

Figure 1a: The sea level and sea temperature stations could be marked a bit more clearly

Line 99-129: I would suggest to restructure this part of the document and to combine the information on model configuration: bathymetry data, boundary data, initial conditions, forcing data (atmosphere, river), etc. into one part. The first paragraph of 2.2 Numerical experiments is actually presenting the model configuration. It should be included here. The second paragraph of 2.2 Numerical experiments is actually belonging to the model settings that should be part of the SHYFEM related part of the method section. The information on the numerical experiments: (a.) the model validation and water transport assessment and (b.) for the what-if scenarios should be presented in a separate section. Here you could provide more background information on the choosen simulations.

Line 101-105: Please add the resolution of the input data. I assume that you used gridded data products. Who provided the data for the Razelm Sinoie Lagoon and the river branches?

Line 123-129: Could you provide a bit more information on why these what-if scenarios were chosen? Are these realistic scenarios?

Line 130-151: In my opinion, it would be better to move the list of validation data sets to the validation chapter. The model validation chapter could be a separate part of the paper, because it is not so much presenting new results, but is demonstrating the quality and usefulness of the model.

Line 132-135: Please refer to figure 2 here. The validation could also include a comparison with hydrological model results: (E-Hype, SWAT). I think the E-Hype climatology is freely available. Annual mean discharges could be calculated and compared with hydrological model results.

Line 136-137: Here you say that hourly values were available. Then why did you perform the model validation using daily averaged data sets? You say that the model can represent the sea level fluctuations (anomaly) associated with intense meteorological events (Line 168). But these have much shorter time scales. I assume that at least hourly data would be needed. We use 10 minutes data for sea level warnings.

Line 141-151: I'm a bit puzzled that you did not use CMEMS satellite SST product for Black Sea. Is the quality of the CMEMS product not good enough? Why did you use the level 2 product and not the gridded data set?

Chapter 3. Results

Chapter 3.1 Model Validation

Line 153-183: The chapter jumps right away into model validation statistical methods and results. It would be good if you could provide a bit more background information and motivate the validation exercise and the specific choice of parameters and methods.

Line 154-157: Please split this sentence. It is much too long.

Line 155: I assume it is "Pearson correlation" rather than "Pearson cross-correlation"

Line 156: "slope of the linear regression best-fit line". I would suggest to write the "best fit calculated by linear regression".

Line 160: "The model well represents … ", change to "The model represents ….. well."

Line 162-163: I would suggest avoiding this type of reduced writing using brackets. You can form sentences like: While it is underestimating here, it is overestimating there.

Line 158-165: This is a less comprehensive validation study of the quality of hydrological predictions than I would have expected from a paper focusing on river-to-sea continuum modelling. Only 2 stations close to the Danube source point have been chosen. There is a straight river section running from the source point to the Danube river branching point at Ceatal Izmail where the model is validated. I can only assume that errors accumulate further down the river network. The validation exercise could be extended with a comparison of modelled discharge values using SHYFEM and modelled discharge values using hydrological model (E-Hype, SWAT,…). Maybe a literature study would also provide runoff data that could be used for comparison.

Line 162: Are the situations with underestimation at Chilia coinciding with situations of overestimation at Tulcea?

Figure2: Fit2 (a): Is this the river runoff time series at Isaccea? Could this location be marked in Fig 1. What do the different colors of the symbols in Fig2 (b) and (c) represent? There are periods of systematic undeprediction (Constanta: July-September 2027). Could these be linked to meteorological conditions?

Line 168: As mentioned before, I doubt that the model quality with regards to predicting storm events can be validated using daily mean sea level data. It is not possible to do a peak error validation. At least hourly data (which is available, Line 136) should be used.

Line 169: It is mentioned here that storm events up to 10 days lead time can be predicted well. This would require a forecast validation (assessing the quality of the forecast according to lead time), which has not been presented. The validation exercise uses daily hindcast data sets.

Line 170-171: Could the validation results be presented in a table, maybe table 1.

Line 172: You mention that sea level prediction errors originate from the reanalysis product that you use at the boundaries. You could validate the CMEMS reanalysis product at the two stations Mangalia and

Constanta to demonstrate this. The CMEMS Black Sea MYP QUID unfortunately does not use tide-gauge data for validation.

Line 178: "varies strongly" rather than "strongly varies"

Line 178: Could you write which year

Line 178-183: Why did you use daily mean values for satellite SST validation? Aren't midnight values usually used to reduce the errors related to the impact of diurnal warming on the skin-to-bulk temperature conversion?

Line 178-183: could you present the validation results in a figure, maybe even a spatial distribution of model errors.

After Line 183: The model validation chapter provides a lot of data, but only little analysis. Could you write a paragraph evaluating the model performance and in the light of your model application, i.e. the adequate representation of the river-to-sea continuum for detailed model studies of the Danube delta. I think this is needed.

Chapter 3.2 Water division in the river network of the delta:

Line 185: First 3 sentences: I think you should rewrite these sentences to make them more clear. I think you want to say that you can only use the model system to estimate the water discharge distribution among the "main" river branches.

Line 188: "estimate the relative load". I think you mean "relative runoff". The term load refers to substances carried with the river, liked pollutants.

Figure 5 and text: Can you provide a number for the total runoff in the considered period 2015-2019. Then readers can calculate the absolute values of river runoff from the percentages presented in the figure.

Line 194-203: Would it be possible to compare your river runoff data with literature values or values from hydrological models, to get a feeling for the quality of the prediction. This could be done in the validation section.

Chapter 3.3 Spatial and temporal variability of coastal dynamic

Line 206: "river plumes"

Line 209: Could you briefly introduce the variable ROFI.

Line 215: and following: Why do you use the unit g/L and not the more widespread unit psu?

Line 218: Is this the 15$^{th}$ of Jun 2019?

Line 222: The currents in figure 6 are not very easy to see. Do you know what drives these coastal currents? Are they influenced by steeric effects? In other words, does the amount of discharged freshwater and its distribution affect the coastal currents.

Line 222 and following: If you want to use standard unit for currents, then you should use m/s.

Line 226: "seasonal analysis"

Line 226-236: What exactly is a seasonal standard deviation? Is it the standard deviation of the month of all years belonging to the season? Could you motivate, why you are doing a seasonal analysis? You say that you are using a seasonal analysis to calculate the standard deviation, but that can not be the motivation for the analysis.

Line 233: "multiple mouth". I would use another term.

Line 205-236: This is just an idea: To show the advantage of using SHYFEM for modelling the Danube delta and RSLS, a comparison of 2 simulations – one including and one excluding the Danube delta and RSLS domain could be presented. In the second type of simulations without Danube delta and RSLS domain, the freshwater discharge could be added to the coarse Black Sea grid. I think the simulation would show that the second type of simulations are less able to produce realistic river plumes and discharge patterns. This is the advantage of resolving the lagoons and estuaries using dynamical models.

Chapter 3.4 River-Lagoon-sea connectivity

Line 238-264: Can you provide a motivation for why you want to study River-Lagoon-Sea connectivity? The main purpose seems to be, to calculate the ratio of WFT and WRT. Later, however, only WRT is used for assessing the what-if scenarios.

Line 238: What is a "choked water body". Can you please explain this term or use another one.

Line 242-249: Could you rewrite this part and make the different contributions more clear? The following questions may help you.

Of the 62 m3/s water discharge from the Dunavat and Dranov channel, are 42 m3/s discharged into the Black Sea?

Should the sum of the inflow from the Black Sea (+16 m3/s), the outflow to the Black Sea (42 m3/s), the river/channel runoff (+62 m3/s) and the amount of water lost by evaporation (-20 m3/s) cancel out? When I add all the contributions, I get 16 m3/s, which is equal to the inflow from the Black Sea.

Chapter 3.5 Assessment of lagoon-sea reconnection solutions

Line 268: "connections remain open"

Line 266-272: Could you provide a few more words on the different proposals for reconnecting RSLS with the Black Sea? Could you motivate the choice of the proposals.

Line 266 and following: What is the time scale of the assessment? How long are the model simulations? Which years do they cover?

Table 2: It took me a minute to understand the table. I guess, I expected it also to cover the case of multiple openings: single opening, two openings together, three openings together. Maybe this could be explained somewhere.

Line 277-297: It should be explained somewhere that changing the net-flow through the lagoon, changes also the sea level in the lagoon. This is visible in figure 13.

Line 278: "enhanced inflow into the lagoon" and "reduced outflow out of the lagoon"

Line 283-288: I think it would be easier to analyse difference plots (figure 9).

Figure 9: Here in the figure you use the salinity unit psu, whereas in the text you are using g/L.

Chapter 4. Discussion

Line 299: I'm not sure that I understand the structure of the paper. After analyzing the time scales of water transport in the Danube delta system and introducing the what-if scenarios for water transport, now you take one step back and discuss the hydrographic conditions predicted by your model. I would rather suggest move this part to the model study and validation exercise as it is related to the general dynamic in the area. You could focus on the what-if scenarios here.

You could also split the analysis into a Danube and the RSLS part. This way you could have one chapter dealing with the what-if scenarios.

This chapter presents many new results that properly analyzed could form even another paper. I would suggest to take a step back, to define a research question and to restructure the paper accordingly.

Line 299: The first 2-3 sentences are very general. You could just say that you want to investigate the water exchange between the different parts of the Danube delta to study their impact on the local hydrographic conditions. I guess you want to discuss as part of the what-if scenario studies, but this is not clear her.

Line 305: Please rewrite " .. flows into the Black sea via different mouths, having different dimension and discharge"

Line 309: "A deeper analysis … " This sentence sounds more like conclusions than discussions.

Figure 10: Could you please choose some different colors. I can not see the difference between A and B. Solution C is also rather difficult to see.

Line 311: "Water bulges". I can only see the temperature distribution, not the sea level. Is this because of the limited transport capacity from the near-shore to the off-shore?

Line 313: I would suggest to avoid constructions like "warmer (colder)".

Line 314: Upwelling is usually the result of water mass transport, not mixing.

Line 313-317 and figure 11: thanks for the nice plot, but I don't know exactly which point you want to make here. The upwelling event is likely wind driven and would have happened with and without rivers. It is of course clear that the horizontal surface temperature distribution would look different without the implementation of the rivers. But I'm not sure that this is what you want to say, that running a model with an extended Danube estuary results in a better representation of the river plumes.

Figure 12:

Line 318-329: further studies could investigate if the salt water intrusions happen gradually with time or if they are related to certain meteorological conditions and events.

Line 323-324: The locations of maximum salt water intrusion could be shown on the map.

Line 330: Here you present the part related to the what-if scenario. This could be combined with the assessment in chapter 3.4

Line 331: Can you rewrite the sentence starting with "Due to the input …". Results should not be put in brackets. Time periods should b e put to the end of the sentence.

Line 333: The flow is not necessarily barotropic because of a sea level gradient. You can still have a stratified flow. But your model results should show if at least a seasonal halo-or thermocline in the RSLS exist.

Line 338: I assume that meteorological conventions are used and the winds are from north-easterly directions.

Figure 13: The results in this figure could be analyzed for the different what-if scenarios. The differences in sea level could be related to the differences in water transport (table 2, chapter 3.5)

Line 349: There could be an easier formulation for the sentence starting with: "The inflow of marine waters."

Figure 14: It should be made clear that this figure is using results from the reference simulation.

Line 348-363: The discussion here is rather qualitative. Only time series results from the reference simulation are presented. The different what-if scenarios are only discussed in general terms. I would suggest to extend the analysis and to present quantitative results for the different what-if scenarios. Otherwise, the analysis remains a bit unsatisfying. As mentioned before, this study could be combined with chapter 3.5.

Chapter 5: Concluding remarks and perspectives

The concluding remarks focus on the comprehensive modelling tools that have been developed, but they leave the results of the modelling study out. As mentioned before, I would restructure the paper, defining a research question and a story line. The conclusions should summarize the key findings.

---

## Author Comment (AC1)

**Responses to the Referee#1 Comments and Suggestions**

Journal: Ocean Sciences (OS)
Manuscript number: egusphere-2025-606
Manuscript title: Modelling river-sea continuum: the case of the Danube Delta

The original Reviewer's comments and suggestions are shown in regular typeface, while our responses are shown in italics. The line and figures numbers we use refer to the revised document.

**R1.1** The manuscript "Modelling river-sea continuum: the case of the Danube Delta", by Ferrarin et al., describes a new model configuration for the river-lagoon-sea interconnections in the Danube Delta Region. The model is first validated under current conditions and then used to explore the temporal and spatial variability of oceanographic parameters and three alternative reconnection options.

Evaluation: overall the manuscript covers and interesting topic in a region and type of system that could be extrapolated to other similar systems. The manuscript also does a good job (through visuals) at showing the spatial and temporal variability of oceanographic parameters. However, there are questions about the datasets used, the validation of the model results and the interpretation of results could be strengthened. The manuscript would benefit from some reorganisation since the findings and the implications are difficult to follow beyond the focus on the local issue. The scientific questions and relevance are not clearly articulated in the Introduction and some of them appear further down in the Methods. New results are presented in the Discussion and on the other hand the results of the study (both those in the result section and those in the discussion section) are not put into the broader context and remain descriptive. Perhaps the validation should go in supplementary materials and the result figures currently in the discussion (11-14) could be presented in the Results section highlighting how they relate to current understanding of this type of systems. The Conclusion restates the local findings and should be strengthen by highlighting why or how this piece of work is of general interest.

*Response. We thank the referee for the in-depth and useful review. We appreciate the comments and we will improve the manuscript in accordance with all suggestions. In particular, all results will be presented in the "Results" section, while the interpretation of our findings within a broader context will be included in the "Discussion and Conclusions" section.*

**R1.2** The modelling system and Numerical experiments. Significant details on the datasets and the modelling system need to be included without these details it would be difficult to reproduce the results and to evaluate the validity of the numerical modelling: What is the bathymetric resolution of the different datasets? In coastal environments the bathymetry has a non-negligible impact on the obtained results. What is the temporal resolution of the Black Sea lateral boundary condition? Some of the datasets are only available upon request so it will be challenging to reproduce these results. What is the temporal resolution of the model outputs?

*Response: We thank the reviewer for highlighting this issue. In the revised version of the manuscript we will include more details about the modelling application setup. In particular, for the bathymetric dataset we will specify that:*

- *the 2022 European Marine Observation and Data Network dataset (EMODnet Bathymetry Consortium, 2022) for the shelf sea on a regular grid of 1/16*1/16 arc minutes, ca. 115 metre grid;*

- *the 2024 dataset for the Razelm Sinoie Lagoon System acquired on (mostly West-East-oriented) transects spaced 450 m apart on average and covering the whole system. The distance between two points within each transect is ~1 m.*

- *three separate multibeam datasets (provided at a ~1 m resolution) for the main river branches: the 2023 dataset for Chilia; the 2019 dataset for Sulina; the 2016-2017 dataset for Sf. Gheorghe. Available sparse data was used for some secondary branches and small channels.*

*The choice of saving daily model results was made to limit the volume of model output and is justified by the fact that the boundary conditions for the Black Sea and the Danube River river have a daily frequency.*

**R1.3** The numerical experiments: it is not clear for which period are the what-if scenarios run (L126). Please clarify whether it is 2015-2019 or 2018.

*Response. We will add a sentence to specify that the period, parametrization, forcing and boundary conditions considered in the what-if numerical experiments are the same adopted in the reference run.*

**R1.4** L123 highlights why this study matters and yet it only appears here. This should we moved to the Introduction section.

*Response. According to the reviewer's suggestion, the mentioned reference, and the related sentence, will be moved to section 1.1.*

**R1.5** L131 you refer to four datasets not three.

*Response. We will correct the text accordingly.*

**R1.6** The model validation: my main concern is the lack of validation data in terms of the water division in the lower parts of the delta river network and salinity on coastal areas.

*Response. We acknowledge the reviewer's concerns; however, the area of interest is poorly monitored. Limited spatial and temporal coverage of existing monitoring networks, along with restricted (freely) available data, are critical issues in the Danube Delta. To address this, we reached out to various authorities and research centers to gather data on bathymetry, river discharge, water level, temperature, and salinity for use in the development and validation of our model. We utilized all available datasets, and while we agree that validation could be further enhanced, we believe the results demonstrate the robustness of our model. We hope this modelling effort will encourage more effective data sharing among institutions and contribute to advancing research in this ecologically significant transitional environment.*

**R1.7** Fig 2. What are the red circles and the black diamonds in Fig2b and 2c? There is no legend for those. There might be a typo on the x and y axis of Fig 2b and Fig 2c, shouldn't it be "m³/s" instead of "m"?

*Response. The figure's caption will be improved by adding the following sentence "The black diamonds, the red circles and the red lines in panels b and c represent the scatter*

*data, the 5 to 99th percentiles and the line of best fit, respectively.” We will also correct the unit on panels b and c.*

**R1.8** The validation data for sea level and sea temperature is relatively far from the zone of interest where the manuscript focuses. Why is the validation for sea level done for mean sea level and not at the time the measurements are collected? Does the validation improve when done for mean daily values or is it that the model output are daily values? This is not clear. Validation of sea level is quite poor at times (e.g. differences of over 0.1m for 0.7m variation in sea level Fig.3). The reasons (L171) for the poor fitting should be better explained/explored, does the bathymetry play a role as well in this? Why are results not presented as scatter plots?

*Response. See the response to comment R1.6 regarding the validation. The sea level and sea temperature validation has been performed on daily values because this is the model output frequency (see the response to comment R1.1). It is worth noting that the tide in the area of interest is negligible and therefore the sea level oscillations are mostly influenced by the open sea conditions and wind and pressure associated with atmospheric perturbations having a typical time scale of 1-10 days.*

*We decided to present the sea level validation as a timeseries instead of a scatter plot to visualise the amplitude and duration of the sea level variability in the area of interest. The results of the statistical analysis are reported in the text (please note that reperforming the validation, we modified some values).*

*As mentioned in the manuscript, the SHYFEM model's performance is strongly related to the capacity of the Black Sea Physics Reanalysis (hereinafter BLKSEA) in reproducing sea level oscillations. To better explore this dependence, we reported in Fig. 1 both the timeseries of the sea level simulated by our SHYFEM application and by the Black Sea Physics Reanalysis.*

*The figure clearly shows the strong dependency of the SHYFEM results on the imposed boundary conditions. However, the statistical analysis reported in Table 1 demonstrates that SHYFEM is performing slightly better than BLKSEA. This is due to the higher resolution of the SHYFEM model application at the coast, which allows to represent coastal dynamics better.*

Table 1: Statistical analysis (in terms of centered RMSE, BIAS and R) of simulated sea temperature at the monitoring stations.

| Station | RMSE (cm) | | CC | |
|---|---|---|---|---|
| | SHYFEM | BLKSEA | SHYFEM | BLKSEA |
| Constanta | 6.5 | 7.6 | 0.66 | 0.58 |
| Mangalia | 7.8 | 8.3 | 0.55 | 0.51 |

*Unless specifically requested by the reviewer, we do not intend to include a validation of the Black Sea Physics Reanalysis in our manuscript, either in the form of time series figures or statistical metrics in the text.*

**R1.9** Similarly for the sea surface temperature, why is the validation for Constanta not shown in Fig. 4 while data is available? Temperature is usually a minor part in estuarine dynamics with most of the density gradient being driven by salinity differences. However, here the validation is presented in terms of temperature and yet you focus later on salinity results, are there no data available in terms of salinity? There are 6 satellite control SST points,

[Figure]

Figure 1: Observed (red line) and simulated (black dashed line) sea levels at Constanta (top panel) and Mangalia (bottom panel) for year 2017.

however a single set of statistical parameters are given in L182. What are the implications of comparing averaged daily values with specific timings in the satellite data?

*Response: The validation for Constanta was not presented to limit the number of panels in Fig. 4. The results of the statistical analysis reported in Table 1 demonstrate the good performance of the model at all three sea temperature monitoring stations. We can include the timeseries of the sea temperature in Constanza, if required by the reviewer.*

*We are aware that salinity plays a major role in driving estuarine dynamics but, unfortunately, the area of interest is poorly monitored and, to our knowledge, no salinity observing stations exist (see the response to comment R1.6).*

*Satellite sea surface temperature values are provided at midnight to reduce the errors related to the impact of diurnal warming on the skin-to-bulk temperature conversion. Water temperature values were extracted from the simulation results in the surface layer at the location and time corresponding to the satellite SST data. Due to the limited number of available values at each control point, we prefer to apply the statistical analysis to a dataset containing all samples.*

**R1.10** I think that if this validation section could go into supplementary materials to streamline the paper.

*Response: This paper presents the first comprehensive modelling study of the Danube Delta, and we believe the validation must be presented in the main text.*

**R1.11** Table 1 could include the statistical analysis for all the validation datasets (i.e. also include sea level and satellite SST) and not just temperature for monitoring stations.

*Response: Following the reviewer's suggestion, we will included a table reporting the statistical analysis for all the validation datasets (e.g., Table 2) at the beginning of the "Model validation" section.*

Table 2: Statistical analysis of simulated river discharge, sea level, sea temperature and sea surface temperature.

| Variable | Station | N data | RMSE | BIAS | CC | SLOPE |
|---|---|---|---|---|---|---|
| River discharge | Chilia | 120 | 158 | -46 | 1.00 | 0.90 |
| $(\text{m}^3 \text{ s}^{-1}$ | Tulcea | 120 | 158 | 43 | 1.00 | 1.10 |
| Sea level | Constanta | 624 | 6.5 | - | 0.66 | 0.70 |
| (cm) | Mangalia | 722 | 7.8 | - | 0.55 | 0.62 |
| Sea Temperature | 15360 | 972 | 1.7 | 0.2 | 0.98 | 1.08 |
| (°C) | Constanta | 966 | 1.6 | -0.4 | 0.97 | 0.98 |
| | Mangalia | 908 | 1.5 | -0.2 | 0.97 | 0.96 |
| SST (°C) | Razelm Sinoie | 135 | 1.0 | 0.3 | 0.97 | 0.99 |

**R1.12** Water division: water division is only validated in the upper part of the delta, and only temperature values are shown for a point in the vicinity of the outlets. Without discharge measurements in the network or temperature or salinity in the outlets it is difficult to assess how well or bad the model is performing. The water division is likely to influence the coastal dynamics and the plumes observed. These results also don't seem to be further explored in the paper.

*Response: The numerical model application has been designed to resolve the more relevant water courses and can therefore be used to estimate the water discharge distribution in the delta. We believe that, even if only partially validated, the results presented in section 3.2 and Fig. 5 provide significant insights on the river-sea dynamics which are worth publishing. See also the response to comment R1.6 regarding the validation.*

*We will include the following paragraph in section 3.3 to highlight the influence of the water division on the plume dynamics: "In front of the Danube Delta, the general coastal circulation (determined averaging the values over the whole simulated period) reflects these processes with the several branches of the multiple-mouth delta forming separated freshwater plumes having shape and dimension defined by amount of water carried out by the different river branches and the coastline characteristics (Fig. 4a). Indeed, the largest plume is found south of the Sulina mouth, where the 5 km long artificial jetty enhances the offshore spread of riverine waters and creates a well-defined recirculation structure. It has to be noted that this plume is reinforced by the freshwater discharged by the nearby Chilia mouth. Well-defined plumes can be also recognized out of the Sf. Gheorghe, Novo Stambul and Potapov mouths. On average, the region of freshwater influence (ROFI; Simpson et al., 1993) associated with the Danube River extends for about 15 km offshore the river mouths. As illustrated in Fig. 4d, the freshwater inputs determine a stratified water column along the coast with Black Sea waters (defined here as having salinity higher than 16 g $L^{-1}$) located on average below 5 to 10 m from the surface."*

**R1.13** Spatial and temporal variability of coastal dynamics. L216 please define the scales of variability you refer to in the text. Here it is implicit you are considering seasonal scales

as per your figure 6.

*Response: We will modify the sentence to clarify that we are here presenting the results for two specific events having different hydro-meteo-marine conditions: (1) a summer event with peak river discharge (13000 $m^3$ $s^{-1}$) and calm weather (16 June 2019), and (2) an autumn event with low river discharge (2400 $m^3$ $s^{-1}$) and windy (northerly) conditions (11 November 2019). The seasonal scale analysis is reported in Fig. 7.*

**R1.14** L234 there is no correlation calculated here, suggest change "are highly correlated" to "can be explained by".

*Response: We will correct the text following the reviewer's suggestion.*

**R1.15** River lagoon connectivity: Over which period are calculated the averages and estimates in L241 to L249?

*Response: We will specify that unless otherwise specified, the reported values refer to averages over over the whole 2015-2019 simulation period.*

**R1.16** what causes the difference in WRT in the different years in L253 to L256? Are the weather regimes very different? Is it the different river discharge? It would be good to further explore the reason for these differences.

*Response: We found an error in the WRT values reported in the original manuscript: the minimum (181 days) and maximum (333 days) basin-wide WRT values are found in 2018 and 2017, respectively. To further explore the role of forcing on the WRT computation, we analysed the characteristics of river discharge and wind in different years. The main findings are reported in the following sentences, which will be included in the revised manuscript: "The spatial and temporal variability of river, ocean and meteorological conditions affects the river-lagoon-sea fluxes as well as the internal mixing in the lagoons, and consequently the WRT computation. Indeed, the difference in WRT across the different years primarily reflects the freshwater input into the lagoons that mainly drives the river-lagoon-sea fluxes and therefore the flushing of the lagoon waters. Indeed, the minimum (181 days) and maximum (333 days) basin-wide WRT values are found in the flood (2018) and drought (2017) years, respectively (Fig. 2). A secondary, but not negligible, role is played by the wind which, in 2018 was characterized by frequent and intense Northerlies that enhanced internal mixing and favored the outflow from the lagoon towards the sea."*

**R1.17** L259 The sense of the gradient and the number of days seem reversed.

*Response: We will correct the sentence as "... a marked east-to-west WRT gradient (from 50 to more than 300 days) is evident ...".*

**R1.18** Assessment of lagoon-sea reconnection solutions L270 please indicate for which period where the what if scenarios run.

*Response: We will indicate the 2015-2019 period. See the response to comment R1.3.*

**R1.19** You could combine Fig. 8 and Fig. 9 and present current Fig. 9 as differences with respect to the reference run instead of absolute values. That way the differences would be more easily perceived.

*Response: Following the reviewer's suggestion, we created a figure (included in this document as Fig. 2) presenting the results of the reference simulation as absolute values and of the reconnection scenarios as differences with respect to the reference run. In the revised version of the manuscript, we will replace Fig. 8 and remove Fig. 9.*

[Figure]

Figure 2: Average water renewal time and salinity over the Razelm Sinoie Lagoon System for the reference simulation (as absolute values) and the reconnection scenarios A, B and C (as difference with respect to the reference run). The red dots in panel *e* indicate the location of the two control points where the salinity timeseries were extracted (Fig. 10).

**R1.20** L277-282: please point to table 2

*Response: We will add the reference to Table 2.*

**R1.21** Fig 10, please change the colours they're not very colour blind friendly particularly sol.A and sol. B.

*Response: Following the reviewer's suggestion, we changed the lines' colours in the mentioned figure (included in this document as Fig. 3).*

**R1.22** Discussion. The discussion further presents new results (there are 4 results figures in this section) and while they are interesting, they're not put in context or related to similar systems. I believe the discussion would be strengthened if these new 4 figures were moved into the results section and the results were further discussed and contextualised in the discussion. It is difficult to see what is of interest beyond the regional area and what learnings could be taken to other regions.

*Response: Following the reviewer suggestion, all results will be presented in the "Results" section, while the interpretation of our findings within a broader context will be included*

[Figure]

Figure 3: Timeseries of modelled salinity extracted in the control points in the Razelm (top panel) and Sinoie (bottom panel).

*in the "Discussion and Conclusions" section.*

**R1.23** L305-309: I believed this is the first time that stratification is mentioned in the manuscript. So far, the analysis has been limited to the surface. Beside there is no reference to the different wind regimes, could you please elaborate on its influence on the circulation and vertical mixing patterns? Fig 6b and 6c correspond to surface salinity and currents.

*Response: We will move these results and the related figure in section 3.2. Moreover, to investigate the influence of river discharge and wind on the coastal dynamics, we added to Fig. 6 (here Fig. 4) the plots of salinity along transect N-S. Such a figure, as well as the following text will be included in the revised version of the manuscript.*

*"As illustrated in Fig. 4d, the freshwater inputs determine a stratified water column along the coast with Black Sea waters (defined here as having salinity higher than 16 g $L^{-1}$) located on average below 5 to 10 m from the surface. During peak river flow and northerly conditions, vertical mixing processes near the coast occupy the whole water column (Fig. 4e). On the contrary, during low river discharge, the surface coastal dynamic is mainly driven by the wind. The autumn event presented in Fig. 4c is characterized by a general northward surface transport of saline waters with the ROFI limited to river plumes extending north-eastward for a few km from the river mouths. During such an event, the water column in front of the delta is well mixed except for a surficial 2 m thick layer in front of the main river branches (Fig. 4f)."*

**R1.24** L306 This is the first reference to the stratification of the water column and the stratification has not been shown anywhere in the paper, please point the reader to the figures

[Figure]

Figure 4: Surface salinity and current velocity maps, and N-S salinity transects: a) and d) average values over the 2015-2019 period; b) and e) instant values on 15 June 2019; c) and f) instant values on 11 November 2019. The arrows in the top right corner of panels b and c indicate the wind direction.

as needed or rephrase.

*Response: We will rephrase this sentence. See also the response to comment R1.23.*

**R1.25** L309 Fig6b and 6c only show surface, not vertical processes as indicated, please rephrase. The wind regimes are not shown anywhere in the paper so it is difficult to follow the reasoning.

*Response: We will rephrase this sentence. The wind arrows have been included in the figures Fig. 4b and c. See also the response to comment R1.23.*

**R1.26** L315 Upwelling is wind driven although it may interact with river plumes. You may wish to clarify the reasoning in this paragraph.

*Response: We will change the mentioned sentence as "The presented analysis indicate that these peculiar structures are generated by upwelling processes induced by the action of southerly winds blowing along the coastline and interacting with the river outflow."*

**R1.27** L325 is the first reference to salt intrusion, there is no previous information or profile to assess the type of salt intrusion. You could include a profile along the main branches in Fig., 12.

*Response: We will remove the section describing saltwater intrusion since it does not add significant advancements for this specific study area.*

**R1.28** Fig 13, please change the colours. Sol. A and Sol. B are difficult to distinguish. It is not clear how or where the differences are calculated. Is this over the whole lagoon? Is it the differences between two points one in the lagoon and one at sea?

*Response: Following the reviewer's suggestion, we changed the lines' colours (included in this document as Fig. 5). We corrected the figure's caption to clarify that the image represents the average (over 2015-2019 period) water levels along the AB transect crossing the Razelm Sinoie Lagoon Systems and indicated with a red line in the right panel.*

[Figure]

Figure 5: Average (over 2015-2019 period) water levels along the AB transect crossing the Razelm Sinoie Lagoon Systems indicated with a red line in the right panel. Background: ©OpenStreetMap contributors 2024; distributed under the Open Data Commons Open Database License (ODbL) v1.0.

**R1.29** L337 please clarify what you mean by hydraulically limited here. You again refer to the wind here, but you do not present any information about the wind. You could include wind information in the figure in the same way that you include the river discharge.

*Response: We will modify the sentence as "The water level jumps between the two lagoons and between the Sinoie Lagoon and the open sea indicate that the water exchange between the different water bodies is limited by the transport capacity of the narrow and shallow connecting canals (Canal 2, Canal 5, Edighiol and Periboina inlets)."*

*As mentioned in section 1.1, the two major wind regimes characterizing the study area are from north-east, being the most intense, and south-south-west, that can drive alongshore water and sediment transport (Dan et al., 2009). To provide more information about the wind variability, we included the wind speed and direction in Fig. 14 (see the response to comment R1.30).*

**R1.30** L340. Fig14 would benefit from including wind regime.

*Response: Following the reviewer's suggestion, we included the wind speed and direction in Fig. 14 (included in this document as Fig. 6.)*

**R1.31** L345-347 please clarify what you mean by with hydraulically controlled.

*Response: The mentioned sentence will be modified as "It must be noted that the water flux is not linearly dependent on the water level gradients confirming that the flow between the lagoon and the sea is limited by the transport capacity of the Edighiol and Periboina inlets (ad example at the beginning of March)."*

**R1.32** Conclusions. The conclusion could be strengthened by highlighting why or how this work is relevant beyond the study area at present it just restates what the paper does, and it is difficult to see why it would be of interest of the broader community beyond the study area. Could this be further explored?

[Figure]

Figure 6: (a) daily values for the year 2018 of Danube River discharge; (b) water level difference between the lagoon and the sea; (c) wind speed (y-axes) and direction (colour); (d) sea-lagoon water (m³ s⁻¹) and salt (10³ kg s⁻¹) fluxes (positive values indicate inflow into the lagoons while negative values indicate outflow from the lagoon to the sea).

*Response: As mentioned in the response to comment R1.22, the interpretation of our findings within a broader context will be included in the "Discussion and Conclusions" section.*

**R1.33** L382 "four lagoon-sea reconnections" I believe you explore 3 different options.

*Response: We will correct the text.*

**References**

Dan, S., Stive, M., Walstra, D.-J. R., and Panin, N.: Wave climate, coastal sediment budget and shoreline changes for the Danube Delta, Mar. Geol., 262, 39–49, https://doi.org/10.1016/j.margeo.2009.03.003, 2009.

EMODnet Bathymetry Consortium: EMODnet Digital Bathymetry (DTM 2022), https://doi.org/10.12770/ff3aff8a-cff1-44a3-a2c8-1910bf109f85, 2022.

Simpson, J. H., Bos, W., Schirmer, F., Souza, A., Rippeth, T., Jones, S., and Hydes, D.: Periodic stratification in the rhine ROFI in the North Sea, Oceanologica Acta, 16, 23–32, 1993.

---

## Author Comment (AC2)

**Responses to the Referee#2 Comments and Suggestions**

**Journal: Ocean Sciences (OS)**
**Manuscript number: egusphere-2025-606**
**Manuscript title: Modelling river-sea continuum: the case of the Danube Delta**

The original Reviewer's comments and suggestions are shown in regular typeface, while our responses are shown in italics. The line and figures numbers we use refer to the revised document.

**R2.1** The paper introduces a rather unique SHYFEM configuration that is integrating the Danube delta and the RSLS lagoon system into a coastal model for the western Black Sea. This unified approach to modelling the land-estuary-sea continuum is demanding in terms of model numeric and physics. The authors have carried out a thorough study and achieved good and relevant results that were used to estimate the water transport and hydrographic conditions in the Danube delta system.

The following includes some general comments, followed by a more detailed review of the paper.

*Response: We thank the referee for the in-depth and useful review. We appreciate the comments and we will improve the manuscript in accordance with all suggestions. In particular, we will restructure the manuscript to better clarify the scope and content of this study.*

**R2.2** The authors carried out a thorough research study, but could have done more to analyse the data in a more comprehensive way targeting well-defined research questions. I am missing a consistent storyline leading through the paper. This is the case for both the entire paper and individual chapters. As it is, the authors have basically written a general study of the Danube river and Razelm Sinoie Lagoon System (RSLS), which provides a lot of information, but is not embedding it into a consistent story-line. The manuscript is actually two papers in one, each with its own research question. The first paper is (1.) demonstrating that the river-sea continuum can be modelled successfully using unstructured grid models and (2.) the second paper is studying the impact of openings of the RSLS towards the Black Sea and its impact on local circulation pattern and hydrographic conditions. I would suggest the authors to define one or two research questions and then to develop a story line addressing this questions. With this, they could streamline the whole paper and make it more concise.

In the first paper (1.), I would expect to find an analysis of the impact of coastal high-resolution configurations on river plume modelling: comparison with and without the Danube river model and RSLS lagoon model. The paper could demonstrate the effect of improved river-plume-modelling on the freshwater distribution, pollutant distribution, etc., with impact on the local hydrographic conditions. To be true, this has been done, but the analysis could have been further expanded and could have been presented with a view to answer the research question rather than to present model results. The analysis could also include a comparison of modelled river data with either observations or hydrological model results (E-hype, SWAT, etc.). I think E-hype climatology is freely available.

The subject of the second paper idea (2.) mentioned above, the study of the effects of openings in the RSLS towards the Black Sea on the hydrographic conditions is only shortly covered in the paper. The analysis of impacts could be extended. The model results could be analyzed in the light of the objectives of the openings, which are not very clear to me. Here, too, I would prefer if the model results could be used to answer the research question rather than being presented.

*Response: We thank the referee for the comment, which has helped us to clarify the scope and content of our paper. In the "Introduction" section we will specify that "this paper focuses on the investigation of the hydrodynamic processes, water exchange and connectivity among the different interconnected water compartements (river branches, channels, lagoons, coastal sea) forming the Danube Delta river-sea continuum. To archive this goal, we implemented the SHYFEM (System of HydrodYnamic Finite Element Modules; Umgiesser et al., 2022) model to the entire Danube Delta, covering about 500 km of the river network, the Razelm Sinoie Lagoon System and part of the prodelta coastal sea (Fig. 1). The model results are used to quantify water discharge distribution among the river branches, to evaluate the effects of multiple river plumes on the coastal dynamics, and to investigate the water exchange and the renewal capacity of the Razelm Sinoie Lagoon System. Moreover, the numerical tool is used to assess the potential impacts of different hypothetical lagoon-sea reconnection solutions (what-if scenarios) on the processes regulating the exchanges between the river, lagoon, and sea."*

*The analysis of the numerical model results will be improved to investigate the processes regulating the exchanges among the different water compartments.*

*We thank the reviewer for the suggestion of using hydrological model results for validating the SHYFEM model. Unfortunately, none of the available model datasets (EFAS, E-Hype, HERA) provides the water division in the river network of the Danube Delta, and, therefore, cannot be used in the model validation.*

**R2.3** The quality of the writing varies. I would strongly recommend to improve the orthography and grammar. Sometimes the construction of the sentences is not correct. Furthermore, the style is often rather direct, and focused on presenting facts. This is often done in loosely connected paragraphs, which could be better integrated.

*Response: The manuscript's orthography and grammar will be improved.*

**R2.4** The different measures of water transport and mixing: ROFI, WFT, WRT could be introduced in a combined and more consistent way in the method section. Currently there is only WRT defined, in the part of thepaper that is dealing with the SHYFEM model (Line 77-87). WFT is later on defined when using it. A clear definition for ROFI has not been provided. It would be good to define these quantities and how they are used in a consistent way. It is for example not clear until later, that the ratio of WFT and WRT is used. This should be done in a separate part of the method section, not in the model description.

*Response: We will include in the "Method" section the following paragraph describing the different measures of water transport and mixing (WFT, WRT, mixing capacity): "To investigate how the difference forcing and processes influence the water mixing and renewal in the semiclosed Razelm Sinoie Lagoon System, the numerical model has been used to estimate two transport time scales: the water flushing time (WFT) and the water renewal time (WRT) (Umgiesser et al., 2014). The basin-wide WFT is defined as the theoretical time necessary to replace the complete volume of the water body with new water and assuming a hypothetical fully mixed basin and is computed diving the basin volume by*

*the volumetric water flux flowing out of the system. WRT is computed by simulating the transport and diffusion of a Eulerian conservative tracer released uniformly throughout the entire lagoon system with a concentration corresponding to 1, while a concentration of zero was imposed on the seaward and freshwater boundaries. The local WRT is considered as the time required for each cell of the RSLS to replace the mass of the conservative tracer, originally released, with new water. The ratio between the basin wide WFT and WRT can be interpreted as an index of the mixing behaviour of the basin. The reader may refer to Cucco et al. (2009) and Umgiesser et al. (2022) for a more comprehensive description of the transport time scales."*

*Since these variables are directly estimated from model results, we will place this paragraph at the end of the "The modelling system" section.*

*ROFI is not a variable. This name, which is an acronym for Region Of Freshwater Influence, is commonly used in literature to define a coastal area influenced by the river plume. The definition for ROFI will be provided at the beginning of section 3.3.*

**R2.5** Chapter 0. Abstract. In its current form, the abstract outlines the scope of the study. It could provide more motivation as to why this study has been carried out. The key findings should be listed as well. It is good to think of the abstract as a mini-IMRAD scheme, including all the parts of the paper.

*Response: We thank the referee for the suggestion. The new abstract will read: "Understanding water transport and circulation in coastal seas and transitional environments is among the key topics of oceanographic and climate research, as well as recognizing the role of the land-sea interface. The Danube Delta represents a natural laboratory for river-sea hydrodynamic modelling due to its complex morphology composed by several river branches, channels and lagoons. Moreover, this coastal environment is subjected to several natural and anthropogenic stressors and a numerical model could provide the scientific basis to assess the impact of human activities. In this work, the SHYFEM finite element hydrodynamic model was applied to the whole river-sea continuum of the Danube Delta region to describe the transport and mixing processes in the different interconnected water bodies forming the delta. The model was run for the period 2015-2019 and allowed to characterize: 1) the water discharge distribution among the river branches, 2) the general hydrodynamic characteristics of the coastal region of freshwater influence, 3) the transport time scale of the Razelm Sinoie Lagoon System, and 4) the processes driving the river-lagoon-sea interconnections. Lastly, the Danube Delta modelling tool was used to evaluate the potential effects of hydrological reconnection (restoration) measures in the Razelm Sinoie Lagoon System designed to improve connectivity and water renewal."*

**R2.6** Line 3: The sentence should end after "morphology". Then a new sentence should start.

*Response: We will modify the sentence as "The Danube Delta represents a natural laboratory for river-sea hydrodynamic modelling due to its complex morphology. Moreover, this coastal environment is subjected to several natural and anthropogenic stressors."*

**R2.7** Line 5: "The model was run for several years ..." How many? "Several year" is a bit vague.

*Response: We will modify the sentence as "The model was run for the period 2015-2019 to ..."*

**R2.8** Line 19: Think of a better begin of the sentence than "Modelling these coastal transitional water systems". Are these not "estuaries and coastal seas".

*Response: We will modify the sentence as "Modelling estuaries and deltas is challenging ..."*

**R2.9** Line 31: I would say "... Danube delta, covering ..." Please refer to Figure 1.

*Response: We will modify the sentence as "... to the entire Danube Delta, covering about 500 km of the river network, the Razelm Sinoie Lagoon System and part of the prodelta coastal sea (Fig. 1)".*

**R2.10** Line 32 "The manuscript ...", maybe better: "The paper". You can also use the active voice and write "We focus ...'.

*Response: We will modify the sentence as "The paper focuses ..."*

**R2.11** Note: This part of the introduction gives an overview of the scope of the paper. The different points could be used to identify the research question. The advantage of a good research question is that it describes the problem and the motivation for solving the problem.

*Response: We will modify this part of the introduction to clarify the scope of the paper. See the response to comment R2.2.*

**R2.12** Line 36: When we talk about what-if scenarios here, then they must at least described in general terms. It must also be explained what a what-if scenario is. It is not a commonly used term.

*Response: We will modify the sentence to clarify that the numerical tool is used to assess the potential impacts of different hypothetical lagoon-sea reconnection solutions (what-if scenarios) on the processes regulating exchanges between the river, lagoon, and sea.*

**R2.13** Line 44: Please rewrite the sentence starting with "The Romanian part ..." Line 46: "extends on about", use "extends for about" Line 51: "were finalized ... " not "ended up" Line 51: "As a result, more fresh water is discharged into the lagoon system"

*Response: We will correct the mentioned sentences following the reviewer's suggestions.*

**R2.14** Line 54: Please mark Portit or use another way to show it on the map. It is not good to say: "near reconnection option A", before these have been introduced. The same is true for "reconnection option C" in Line 58.

*Response: Following the reviewer's suggestion, we indicate the two former inlets in Fig. 1 (included in this document as Fig. 1).*

**R2.15** Line 55: It's not intuitive that a coastal defense structure could enhance coastal erosion. This sentence could be reformulated or explained in more detail.

*Response: The mentioned sentence will be removed since it is not relevant to the purpose of this study.*

**R2.16** Line 62: The sentence starting with "Anyway" should be rewritten, like: As part of the master plan for the protection of the Romanian Littoral against erosion, a major hydraulic engineering project is currently implemented, to ensure a permanent water exchange through the Periboina Inlet. This is an example. The structure of many sentences in the document could be improved.

*Response: We will correct the sentence in accordance with the suggestion.*

[Figure]

Figure 1: (a) Unstructured numerical grid and bathymetry of the hydrodynamic model of the Danube Delta and Black Sea shelf with the red dots and the green triangles marking the sea temperature and sea level monitoring stations, respectively; (b) zoom of the grid over the Danube Delta with the blue bars near Ceatal Izmail indicating the river discharge monitoring stations; (c) zoom of the grid over the Razelm Sinoie Lagoon Systems with the red bars illustrating the considered reconnection solutions and the yellow diamonds marking the satellite SST control points. Background: ©OpenStreetMap contributors 2024; distributed under the Open Data Commons Open Database License (ODbL) v1.0.

**R2.17** Line 65: What does "lower part" mean? Please re-write: "... average water discharge of [so much], with values ranging from ..."

*Response: We will modify the sentence as "The Danube River before the delta has an average water discharge of 6500 $m^3$ $s^{-1}$, with values ranging".*

**R2.18** Line 77-88: I would suggest to move this paragraph to a different part of the method section and to enhance it. I don't think it makes sense here in the SHYFEM related part of the document. The WRT parameter should be introduced after the model. The other variables ROFI, WFT should be introduced as well. The motivation for using these variables to study water transport through the lagoon system should be clear from the beginning.

*Response: As mentioned in the response to comment R2.2, the description of WFT and WRT will been enhanced. Such a description will be placed at the end of the The modelling system.*

**R2.19** Line 86: Does WRT really measure the time until the concentration fall to zero or is the time until they fall below a small value enough? I could imagine that it takes a long time until absolute zero is reached.

*Response: We will modify the sentence as "The local WRT is considered as the time required for each cell of the RSLS to replace the mass of the conservative tracer, originally released, with new water."*

**R2.20** Line 86: I assume the water parcels are grid cells inside the investigated water body, i.e. the lagoon.

*Response: See the response to comment R2.22.*

**R2.21** Figure 1a: The sea level and sea temperature stations could be marked a bit more clearly

*Response: We will increase the size of the markers.*

**R2.22** Line 99-129: I would suggest to restructure this part of the document and to combine the information on model configuration: bathymetry data, boundary data, initial conditions, forcing data (atmosphere, river), etc. into one part. The first paragraph of 2.2 Numerical experiments is actually presenting the model configuration. It should be included here. The second paragraph of 2.2 Numerical experiments is actually belonging to the model settings that should be part of the SHYFEM related part of the method section. The information on the numerical experiments: (a.) the model validation and water transport assessment and (b.) for the what-if scenarios should be presented in a separate section. Here you could provide more background information on the choosen simulations.

*Response: Following the reviewer's suggestion, we will restrucure the Numerical experiments by including:*

- *in section 2.1 the description of the SHYFEM model, the numerical grid, the bathymetric dataset, the model setting and the methods for computing WFT, WRT and mixing capacity;*

- *in section 2.2 the description of the simulations duration, the forcing and boundary data, the initial conditions, and the what-if scenarios. In this last part, we will provide more background information on the choosen reconnection solutions.*

**R2.23** Line 101-105: Please add the resolution of the input data. I assume that you used gridded data products. Who provided the data for the Razelm Sinoie Lagoon and the river branches?

*Response: We will integrate the description of the bathymetric datasets as: "The model bathymetry is obtained by a bilinear interpolation on the numerical grid of the following available datasets (all referred to the Marea Neagra Sulina vertical datum):*

- *the 2022 European Marine Observation and Data Network dataset (EMODnet Bathymetry Consortium, 2022) for the shelf sea on a regular grid of 1/16\*1/16 arc minutes, ca. 115 metre grid;*

- *the 2024 dataset for the Razelm Sinoie Lagoon System acquired on (mostly West-East-oriented) transects spaced 450 m apart on average and covering the whole system. The distance between two points within each transect is ∼1 m.*

- *three separate multibeam datasets (provided at a ∼1 m resolution) for the main river branches: the 2023 dataset for Chilia; the 2019 dataset for Sulina; the 2016-2017 dataset for Sf. Gheorghe. Available sparse data was used for some secondary branches and small channels.*

*"*

**R2.24** Line 123-129: Could you provide a bit more information on why these what-if scenarios were chosen? Are these realistic scenarios?

*Response: We will integrate the description of the what-if scenarios as "Additional numerical experiments were conducted to investigate the potential effects on the lagoons' water renewal and salinisation of different reconnection solutions designed in collaboration with local stakeholders to enhance the river-lagoon-sea exchange. The dredging of*

*a new inlet is under consideration by local communities and authorities, as part of the activities developed under the framework of the Horizon Europe Project DANUBE4all (`https://www.danube4allproject.eu/`). The three what-if scenarios considered in this study consisted of opening a 1.5 m depth and 70 m wide channel to connect the either the Razelm Lagoon (solutions A and B in Fig. 1c) or the Sinoie Lagoon (solution C in Fig. 1c) with the Black Sea. These reconnection solutions are located in the vicinity of previous inlets, now either closed by humans (Portiţa) or now clogged (Gura Buhazului inlet, active till the beginning of the 1990s). The period, parametrization, forcing and boundary conditions considered in these what-if numerical experiments are the same as those adopted in the reference run (hereinafter called REF)."*

**R2.25** Line 130-151: In my opinion, it would be better to move the list of validation data sets to the validation chapter. The model validation chapter could be a separate part of the paper, because it is not so much presenting new results, but is demonstrating the quality and usefulness of the model.

*Response: Following the reviewer's suggestion, we will move the description of the validation datasets to the "Model validation" section. This paper presents the first comprehensive modelling study of the Danube Delta, and we believe the validation must be presented in the "Results" section.*

**R2.26** Line 132-135: Please refer to figure 2 here. The validation could also include a comparison with hydrological model results: (E-Hype, SWAT). I think the E-Hype climatology is freely available. Annual mean discharges could be calculated and compared with hydrological model results.

*Response: We thank the reviewer for the suggestion of using hydrological model results for validating the SHYFEM model. Unfortunately, none of the available model datasets (EFAS, E-Hype, HERA) provides the water division in the river network of the delta, and, therefore, cannot be used in the model validation.*

**R2.27** Line 136-137: Here you say that hourly values were available. Then why did you perform the model validation using daily averaged data sets? You say that the model can represent the sea level fluctuations (anomaly) associated with intense meteorological events (Line 168). But these have much shorter time scales. I assume that at least hourly data would be needed. We use 10 minutes data for sea level warnings.

*Response: I understand that using daily averaged sea levels may seem unconventional; however, this approach is justified because tidal effects in the study area are negligible. As a result, sea level variations are primarily driven by open sea conditions and atmospheric disturbances with typical time scales of 1 to 10 days. Additionally, the model is forced at the Black Sea and Danube River boundaries using daily datasets. Given these assumptions and to limit the model's output volume, results are saved at a daily frequency. Finally, the analysis of sub-daily dynamics is beyond the scope of this study.*

**R2.28** Line 141-151: I'm a bit puzzled that you did not use CMEMS satellite SST product for Black Sea. Is the quality of the CMEMS product not good enough? Why did you use the level 2 product and not the gridded data set?

*Response: The capacity of the model in reproducing the sea temperature in the coastal waters of the Black Sea was assessed through the comparison of in-situ timeseries (Constanta, Mangalia, 15360). Satellite SST data were only used to validate the model in the Razelm Sinoie Lagoon System. We used the level 2 product (kindly provided by colleagues from the University of Stirling) because they were specifically processed for a very shallow environment.*

**R2.29** Line 153-183: The chapter jumps right away into model validation statistical methods and results. It would be good if you could provide a bit more background information and motivate the validation exercise and the specific choice of parameters and methods.

*Response: The "Model validation" section will be reworked to include background information, methods, validation data and validation results.*

**R2.30** Line 154-157: Please split this sentence. It is much too long. Line 155: I assume it is "Pearson correlation" rather than "Pearson cross-correlation" Line 156: "slope of the linear regression best-fit line". I would suggest to write the "best fit calculated by linear regression". Line 160: "The model well represents ... ", change to "The model represents ..... well." Line 162-163: I would suggest avoiding this type of reduced writing using brackets. You can form sentences like: While it is underestimating here, it is overestimating there.

*Response: We will revise the manuscript in accordance with the reviewer's suggestions.*

**R2.31** Line 158-165: This is a less comprehensive validation study of the quality of hydrological predictions than I would have expected from a paper focusing on river-to-sea continuum modelling. Only 2 stations close to the Danube source point have been chosen. There is a straight river section running from the source point to the Danube river branching point at Ceatal Izmail where the model is validated. I can only assume that errors accumulate further down the river network. The validation exercise could be extended with a comparison of modelled discharge values using SHYFEM and modelled discharge values using hydrological model (E-Hype, SWAT,...). Maybe a literature study would also provide runoff data that could be used for comparison.

*Response: We acknowledge the reviewer's concerns; however, the area of interest is poorly monitored. Limited spatial and temporal coverage of existing monitoring networks, along with restricted (freely) available data, are critical issues in the Danube Delta. To address this, we reached out to various authorities and research centers to gather data on bathymetry, river discharge, water level, temperature, and salinity for use in the development and validation of our model. We used all available datasets, and while we agree that validation could be further enhanced, we believe the results demonstrate the robustness of our model. We hope this modelling effort will encourage more effective data sharing among institutions and contribute to advancing research in this ecologically significant transitional environment.*

**R2.32** Line 162: Are the situations with underestimation at Chilia coinciding with situations of overestimation at Tulcea?

*Response: Yes, we will mention it in the manuscript.*

**R2.33** Figure2: Fit2 (a): Is this the river runoff time series at Isaccea? Could this location be marked in Fig 1. What do the different colors of the symbols in Fig2 (b) and (c) represent? There are periods of systematic undeprediction (Constanta: July-September 2027). Could these be linked to meteorological conditions?

*Response: Isaccea is indicated with name and arrow in Fig. 1a. The figure's caption will be improved by adding the following sentence "The black diamonds, the red circles and the red lines in panels b and c represent the scatter data, the 5 to 99th percentiles and the line of best fit, respectively."*

*The underestimation and overestimation during flood events in the two branches is likely attributable to the model's inability to correctly reproduce the river overflow into the sur-*

*rounding floodplains. We expect that meteorological conditions do not significantly affect the river flow in the upper delta.*

**R2.34** Line 168: As mentioned before, I doubt that the model quality with regards to predicting storm events can be validated using daily mean sea level data. It is not possible to do a peak error validation. At least hourly data (which is available, Line 136) should be used.

*Response: See the response to comment R2.29.*

**R2.35** Line 169: It is mentioned here that storm events up to 10 days lead time can be predicted well. This would require a forecast validation (assessing the quality of the forecast according to lead time), which has not been presented. The validation exercise uses daily hindcast data sets.

*Response: The numerical model is run in hindcast mode and no forecasts are presented in this study. We will correct the sentence to clarify that, in the study area, the sea level variations are primarily driven by open sea conditions and atmospheric disturbances with typical time scales of 1 to 10 days.*

**R2.36** Line 170-171: Could the validation results be presented in a table, maybe table 1.

*Response: Following the reviewer's suggestion, the results of the statistical analysis of river discharge, sea level and sea temperature will be reported in a table (included in this document as Table 1).*

Table 1: Statistical analysis of simulated river discharge, sea level, sea temperature and sea surface temperature.

| Variable | Station | N data | RMSE | BIAS | CC | SLOPE |
|---|---|---|---|---|---|---|
| River discharge | Chilia | 120 | 158 | -46 | 1.00 | 0.90 |
| ($m^3\ s^{-1}$ | Tulcea | 120 | 158 | 43 | 1.00 | 1.10 |
| Sea level | Constanta | 624 | 6.5 | - | 0.66 | 0.70 |
| (cm) | Mangalia | 722 | 7.8 | - | 0.55 | 0.62 |
| Sea Temperature | 15360 | 972 | 1.7 | 0.2 | 0.98 | 1.08 |
| (°C) | Constanta | 966 | 1.6 | -0.4 | 0.97 | 0.98 |
| | Mangalia | 908 | 1.5 | -0.2 | 0.97 | 0.96 |
| SST (°C) | Razelm Sinoie | 135 | 1.0 | 0.3 | 0.97 | 0.99 |

**R2.37** Line 172: You mention that sea level prediction errors originate from the reanalysis product that you use at the boundaries. You could validate the CMEMS reanalysis product at the two stations Mangalia andConstanta to demonstrate this. The CMEMS Black Sea MYP QUID unfortunately does not use tide-gauge data for validation.

*Response: To better explore this dependence, we reported in Fig. 2 both the timeseries of the sea level simulated by our SHYFEM application and by the Black Sea Physics Reanalysis (hereinafter BLKSEA).*

*The figure clearly shows the strong dependency of the SHYFEM results on the imposed boundary conditions. However, the statistical analysis reported in Table 2 demonstrates that SHYFEM is performing slightly better than BLKSEA. This is due to the higher resolution of the SHYFEM model application at the coast, which allows to represent coastal dynamics better.*

[Figure]

Figure 2: Observed (red line) and simulated (black dashed line) sea levels at Constanta (top panel) and Mangalia (bottom panel) for year 2017.

Table 2: Statistical analysis (in terms of centered RMSE, BIAS and R) of simulated sea temperature at the monitoring stations.

| Station | RMSE (cm) | | CC | |
|---|---|---|---|---|
| | SHYFEM | BLKSEA | SHYFEM | BLKSEA |
| Constanta | 6.5 | 7.6 | 0.66 | 0.58 |
| Mangalia | 7.8 | 8.3 | 0.55 | 0.51 |

*It is not our intention, unless specifically requested by the reviewer, to present in our manuscript a validation (either as timeseries in the figure or as statistical metrics in the text) of the Black Sea Physics Reanalysis.*

**R2.38** Line 178: "varies strongly" rather than "strongly varies.

*Response: We will correct the sentence.*

**R2.39** Line 178: Could you write which year

*Response: We will modify the sentence as "... that sea surface temperature in RSLS varies strongly over time with values ranging..."*

**R2.40** Line 178-183: Why did you use daily mean values for satellite SST validation? Aren't midnight values usually used to reduce the errors related to the impact of diurnal warming on the skin-to-bulk temperature conversion?

*Response: Yes, they are midnight values. We will better describe the SST dataset in the "Model validation" section.*

**R2.41** Line 178-183: could you present the validation results in a figure, maybe even a spatial distribution of model errors.

*Response: Due to the limited number of available values at each control point, we prefer to apply the statistical analysis to a dataset containing all samples. Moreover, spatially SST in the lagoons has a small spatial variability with difference among stations lower than 1.5 °C.*

**R2.42** After Line 183: The model validation chapter provides a lot of data, but only little analysis. Could you write a paragraph evaluating the model performance and in the light of your model application, i.e. the adequate representation of the river-to-sea continuum for detailed model studies of the Danube delta. I think this is needed.

*Response: The following paragraph will be added at the end of the "Model validation" section: "Concluding, the validation analysis demonstrates that the SHYFEM model application correctly reproduces hydrodynamics in the different water compartments of the delta. The variable model resolution is of fundamental importance for reproducing the complex morphology of the Danube Delta realizing a seamless transition between different spatial scales, from river branches to the coastal sea. The validation of the model could be further improved with the availability of new observations in the future, particularly river discharge and salinity data. We hope this modelling effort will promote more effective data monitoring and support ongoing research in this ecologically significant transitional environment."*

**R2.43** Line 185: First 3 sentences: I think you should rewrite these sentences to make them more clear. I think you want to say that you can only use the model system to estimate the water discharge distribution among the "main" river branches.

*Response: We will reformulate these sentences as "The Danube Delta River network comprises a highly complex system of hundreds of natural and artificial channels, streams, and lakes, whose morphological complexity exceeds the resolution capabilities of the numerical model. Consequently, the model was configured to represent only the most hydraulically significant watercourses, enabling the estimation of water discharge distribution among the principal river branches."*

**R2.44** Line 188: "estimate the relative load". I think you mean "relative runoff". The term load refers to substances carried with the river, liked pollutants.

*Response: We will correct the mentioned sentences following the reviewer's suggestion.*

**R2.45** Figure 5 and text: Can you provide a number for the total runoff in the considered period 2015-2019. Then readers can calculate the absolute values of river runoff from the percentages presented in the figure.

*Response: We will provide the value of the average the Danube River discharge imposed at the open boundary of Isaccea (6500 $m^3$ $s^{-1}$)*

**R2.46** Line 194-203: Would it be possible to compare your river runoff data with literature values or values from hydrological models, to get a feeling for the quality of the prediction. This could be done in the validation section.

*Response: See the response to comment R2.28.*

**R2.47** Line 209: Could you briefly introduce the variable ROFI.

*Response: See the response to comment R2.6.*

**R2.48** Line 215: and following: Why do you use the unit g/L and not the more widespread unit psu?

*Response: The numerical model is computing salinity as g/L. It can be converted to Practical Salinity Units (PSU) by understanding that 1 PSU is approximately equal to 1 gram of salt per kilogram of solution.*

**R2.49** Line 218: Is this the 15 th of Jun 2019?

*Response: We will correct the text.*

**R2.50** Line 222: The currents in figure 6 are not very easy to see. Do you know what drives these coastal currents? Are they influenced by steeric effects? In other words, does the amount of discharged freshwater and its distribution affect the coastal currents.

*Response: We increased the arrows' line width. Moreover, following Reviewer#1's suggestion, we added to Fig. 6 (here Fig. 3) the plots of salinity along transect N-S.*

[Figure]

Figure 3: Surface salinity and current velocity maps, and N-S salinity transects: a) and d) average values over the 2015-2019 period; b) and e) instant values on 15 June 2019; c) and f) instant values on 11 November 2019. The arrows in the top right corner of panels b and c indicate the wind direction.

*We will improve the description of the spatial and temporal variability of coastal dynamics.*

**R2.51** Line 222 and following: If you want to use standard unit for currents, then you should use m/s.

*Response: We will correct the unit for currents to m $s^{-1}$.*

**R2.52** Line 226-236: What exactly is a seasonal standard deviation? Is it the standard deviation of the month of all years belonging to the season? Could you motivate, why you are

doing a seasonal analysis? You say that you are using a seasonal analysis to calculate the standard deviation, but that can not be the motivation for the analysis.

*Response: To clarify the motivation and method of the season analysis, we will modify the mentioned sentence as "To analyse the temporal variability of the coastal dynamics, the model results were processed to computed the standard deviation (hereinafter STD) of the month of all years belonging to the four seasons (winter=DJF, spring=MAM, summer=JJA, fall=SON)."*

**R2.53** Line 233: "multiple mouth". I would use another term.

*Response: We will modify the sentence as "The freshwater discharged by the different branches determine a similar coastal salinity pattern in winter (Fig. 7e) and fall (Fig. 7h)."*

**R2.54** Line 205-236: This is just an idea: To show the advantage of using SHYFEM for modelling the Danube delta and RSLS, a comparison of 2 simulations – one including and one excluding the Danube delta and RSLS domain could be presented. In the second type of simulations without Danube delta and RSLS domain, the freshwater discharge could be added to the coarse Black Sea grid. I think the simulation would show that the second type of simulations are less able to produce realistic river plumes and discharge patterns. This is the advantage of resolving the lagoons and estuaries using dynamical models.

*Response: The additional simulation proposed by the reviewer would entail substantial effort and falls outside the scope of the present study. The Black Sea Physics Reanalysis (`https://doi.org/10.25423/CMCC/BLKSEA_MULTIYEAR_PHY_007_004`) serves as an example of a coarse-resolution model that is unable to realistically simulate river plumes and discharge patterns in front of the Danube Delta.*

**R2.55** Line 238-264: Can you provide a motivation for why you want to study River-Lagoon-Sea connectivity? The main purpose seems to be, to calculate the ratio of WFT and WRT. Later, however, only WRT is used for assessing the what-if scenarios.

*Response: We agree with the reviewer that River-lagoon-sea connectivity was not an appropriate title for this section. We will change the title to Lagoons's water exchange and renewal.*

**R2.56** Line 238: What is a "choked water body". Can you please explain this term or use another one.

*Response: According to (Kjerfve and Magill, 1989), coastal lagoons can conveniently be subdivided into choked, restricted, and leaky systems based on the degree of water exchange between lagoon and ocean. Since the classification of the RSLS is beyond the scope of the present study, we will remove the "chocked" term in the revised manuscript.*

**R2.57** Line 242-249: Could you rewrite this part and make the different contributions more clear? The following questions may help you. Of the 62 m3/s water discharge from the Dunavat and Dranov channel, are 42 m3/s discharged into the Black Sea? Should the sum of the inflow from the Black Sea (+16 m3/s), the outflow to the Black Sea (42 m3/s), the river/channel runoff (+62 m3/s) and the amount of water lost by evaporation (-20 m3/s) cancel out? When I add all the contributions, I get 16 m3/s, which is equal to the inflow from the Black Sea.

*Response: We will rewrite this part to clarify the different contributions to the RSLS's water budget. The new text will read as "The RSLS has a water volume of about 1300 millions $m^3$ and receives 40 and 22 $m^3 \ s^{-1}$ of freshwater from the Dunavăţ and Dranov*

*canals, respectively. This excess water entering the lagoons is primarily discharged into the Black Sea via the Edighiol and Periboina inlets, resulting in a average seaward flow of 58 $m^3$ $s^{-1}$. The average inflow of marine water into the RSLS amounts to 16 $m^3$ $s^{-1}$. Evaporation over the lagoon system overpasses precipitation resulting in a net loss of 20 $m^3$ $s^{-1}$. The lagoons receive a total water flux of 78 $m^3$ $s^{-1}$ from the sea and the river. Therefore, the basin-wide water flushing time is 193 days."*

**R2.58** Line 266-272: Could you provide a few more words on the different proposals for reconnecting RSLS with the Black Sea? Could you motivate the choice of the proposals.

*Response: A more detailed description of the different reconnecting solutions will be included in "Numerical experiments" section. See the response to comment R2.26.*

**R2.59** Line 266 and following: What is the time scale of the assessment? How long are the model simulations? Which years do they cover?

*Response: The simulations for the what-if scenarios span the period 2014–2019 (year 2014 is considered the model spin-up time and not included in the analysis), consistent with the reference simulation. We will add a sentence to specify that the period, parametrization, forcing and boundary conditions considered in the what-if numerical experiments are the same adopted in the reference run.*

**R2.60** Table 2: It took me a minute to understand the table. I guess, I expected it also to cover the case of multiple openings: single opening, two openings together, three openings together. Maybe this could be explained somewhere.

*Response: No multiple openings scenarios were simulated. The three what-if scenarios considered in this study consisted of opening one 1.5 m depth and 70 m wide channel to connect the either the Razelm Lagoon (solutions A and B in Fig. 1c) or the Sinoie Lagoon (solution C in Fig. 1c) with the Black Sea. The above sentence will be included in the manuscript to clarify the setting of the different scenarios.*

**R2.61** Line 277-297: It should be explained somewhere that changing the net-flow through the lagoon, changes also the sea level in the lagoon. This is visible in figure 13.

*Response: This hydrodynamic feedback is reported in the discussion (lines 357-360): "Connecting the Razelm Lagoon with the Black Sea with solutions A and B do not only allow the inflow of marine waters but also changes the water level of the two basins (lines red and magenta in Fig. 13) decreasing the water exchange between the two lagoons and the outflow via the existing inlets (Table 2)."*

**R2.62** Line 283-288: I think it would be easier to analyse difference plots (figure 9).

*Response: Following Reviewer#1's suggestion, we created a figure (included in this document as Fig. 4) presenting the results of the reference simulation as absolute values and of the reconnection scenarios as differences with respect to the reference run. In the revised version of the manuscript, we will discuss the presented findings in details.*

**R2.63** Figure 9: Here in the figure you use the salinity unit psu, whereas in the text you are using g/L.

*Response: The figure will be corrected to ensure that g/L is used consistently throughout the manuscript.*

**R2.64** Line 299: I'm not sure that I understand the structure of the paper. After analyzing the time scales of water transport in the Danube delta system and introducing the what-if scenarios for water transport, now you take one step back and discuss the hydrographic

[Figure]

Figure 4: Average water renewal time and salinity over the Razelm Sinoie Lagoon System for the reference simulation (as absolute values) and the reconnection scenarios A, B and C (as difference with respect to the reference run). The red dots in panel *e* indicate the location of the two control points where the salinity timeseries were extracted (Fig. 10).

conditions predicted by your model. I would rather suggest move this part to the model study and validation exercise as it is related to the general dynamic in the area. You could focus on the what-if scenarios here.

You could also split the analysis into a Danube and the RSLS part. This way you could have one chapter dealing with the what-if scenarios.

This chapter presents many new results that properly analyzed could form even another paper. I would suggest to take a step back, to define a research question and to restructure the paper accordingly.

*Response: All results will be presented in the "Results" section, while the interpretation of our findings within a broader context will be included in the "Discussion and Conclusions" section.*

**R2.65** Line 299: The first 2-3 sentences are very general. You could just say that you want to investigate the water exchange between the different parts of the Danube delta to study their impact on the local hydrographic conditions. I guess you want to discuss as part of the what-if scenario studies, but this is not clear her.

*Response: The "Discussion" section will be restructured. See the response to comment R2.64.*

**R2.66** Figure 10: Could you please choose some different colors. I can not see the difference between A and B. Solution C is also rather difficult to see.

*Response: Following the reviewer's suggestion, we changed the lines' colours in the mentioned figure (included in this document as Fig. 5).*

[Figure]

Figure 5: Timeseries of modelled salinity extracted in the control points in the Razelm (top panel) and Sinoie (bottom panel).

**R2.67** Line 311: "Water bulges". I can only see the temperature distribution, not the sea level. Is this because of the limited transport capacity from the near-shore to the off-shore?

*Response: We will use "patterns" instead of "water bulges".*

**R2.68** Line 313: I would suggest to avoid constructions like "warmer (colder)".

*Response: We will remove them.*

**R2.69** Line 314: Upwelling is usually the result of water mass transport, not mixing.

*Response: We will reformulate the sentence as "The vertical alongshore sea temperature transect presented in Figs. 6c and Figs. 6d indicate an upwelling-driven transport of marine waters from deeper layers to the coastal zone, enhancing mixing between open sea and riverine waters."*

**R2.70** Line 313-317 and figure 11: thanks for the nice plot, but I don't know exactly which point you want to make here. The upwelling event is likely wind driven and would

have happened with and without rivers. It is of course clear that the horizontal surface temperature distribution would look different without the implementation of the rivers. But I'm not sure that this is what you want to say, that running a model with an extended Danube estuary results in a better representation of the river plumes.

*Response: Figure 11 illustrates distinctive coastal circulation patterns that emerge off the delta during southerly wind conditions. The interaction between wind-driven coastal upwelling and river outflow gives rise to small-scale nearshore features, situated between river mouths, which exhibit thermo-haline properties distinct from those of the surrounding waters. The model's variable resolution is crucial for capturing the seamless transition from river branches to the coastal sea, enabling the accurate reproduction of such complex coastal dynamics.*

*These results will be moved to the "Results" section.*

**R2.71** Line 318-329: further studies could investigate if the salt water intrusions happen gradually with time or if they are related to certain meteorological conditions and events.

*Response: We will remove the section describing saltwater intrusion since it does not add significant advancements for this specific study area.*

**R2.72** Line 323-324: The locations of maximum salt water intrusion could be shown on the map.

*Response: See the response to comment R2.73.*

**R2.73** Line 330: Here you present the part related to the what-if scenario. This could be combined with the assessment in chapter 3.4.

*Response: Following the reviewer's suggestion, we will move the part related to the what-if scenario to section 3.5.*

**R2.74** Line 331: Can you rewrite the sentence starting with "Due to the input …". Results should not be put in brackets. Time periods should be put to the end of the sentence.

*Response: We will rewrite the sentence as "The freshwater inflow from the Dunavăţ and Dranov canals creates a persistent water level gradient from Razelm Lagoon to Sinoie Lagoon and the adjacent coastal sea (green line in Fig. 10)."*

**R2.75** Line 333: The flow is not necessarily barotropic because of a sea level gradient. You can still have a stratified flow. But your model results should show if at least a seasonal halo-or thermocline in the RSLS exist.

*Response: We thank the reviewer for highlighting this point. We will clarify that due to the wind energy and relatively shallow water (the average depth is 1.8 m), the water masses are generally vertically well mixed.*

**R2.76** Line 338: I assume that meteorological conventions are used and the winds are from north-easterly directions.

*Response: We will correct the sentence as "… by the dominant winds from north-easterly direction".*

**R2.77** Figure 13: The results in this figure could be analyzed for the different what-if scenarios. The differences in sea level could be related to the differences in water transport (table 2, chapter 3.5).

*Response: We will move these results to section 3.5 and discuss them in relation to the water fluxes presented in Table 2.*

**R2.78** Line 349: There could be an easier formulation for the sentence starting with: "The inflow of marine waters."

*Response: We will modify the sentence as "Marine waters flow into the lagoon when there is a positive water level gradient from the sea to the lagoon."*

**R2.79** Figure 14: It should be made clear that this figure is using results from the reference simulation.

*Response: We will indicate in the figure's caption that the results are from the reference simulation.*

**R2.80** Line 348-363: The discussion here is rather qualitative. Only time series results from the reference simulation are presented. The different what-if scenarios are only discussed in general terms. I would suggest to extend the analysis and to present quantitative results for the different what-if scenarios. Otherwise, the analysis remains a bit unsatisfying. As mentioned before, this study could be combined with chapter 3.5.

*Response: Following the reviewer's suggestion, we will improve the analysis and combine the part related to the what-if scenario with section 3.5.*

**R2.81** Chapter 5: Concluding remarks and perspectives. The concluding remarks focus on the comprehensive modelling tools that have been developed, but they leave the results of the modelling study out. As mentioned before, I would restructure the paper, defining a research question and a story line. The conclusions should summarize the key findings.

*Response: We will restructure the manuscript following the reviewer's suggestions. All results will be presented in the "Results" section, while the interpretation of our findings within a broader context will be included in the "Discussion and Conclusions" section.*

**References**

Cucco, A., Umgiesser, G., Ferrarin, C., Perilli, A., Melaku Canu, D., and Solidoro, C.: Eulerian and lagrangian transport time scales of a tidal active coastal basin, Ecol. Model., 220, 913–922, https://doi.org/10.1016/j.ecolmodel.2009.01.008, 2009.

EMODnet Bathymetry Consortium: EMODnet Digital Bathymetry (DTM 2022), https://doi.org/10.12770/ff3aff8a-cff1-44a3-a2c8-1910bf109f85, 2022.

Kjerfve, B. and Magill, K. E.: Geographic and hydrodynamic characteristics of shallow coastal lagoons., Marine Geology, 88, 187–199, 1989.

Umgiesser, G., Ferrarin, C., Cucco, A., De Pascalis, F., Bellafiore, D., Ghezzo, M., and Bajo, M.: Comparative hydrodynamics of 10 Mediterranean lagoons by means of numerical modeling, J. Geophys. Res. Oceans, 119, 2212–2226, https://doi.org/10.1002/2013JC009512, 2014.

Umgiesser, G., Ferrarin, C., Bajo, M., Bellafiore, D., Cucco, A., De Pascalis, F., Ghezzo, M., Mc Kiver, W., and Arpaia, L.: Hydrodynamic modelling in marginal and coastal seas - The case of the Adriatic Sea as a permanent laboratory for numerical approach, Ocean Model., 179, 102123, https://doi.org/10.1016/j.ocemod.2022.102123, 2022.

---

## Author Response (AR1)

**Responses to the Referee#1 Comments and Suggestions**

**Journal: Ocean Sciences (OS)**
**Manuscript number: egusphere-2025-606**
**Manuscript title: Modelling river-sea continuum: the case of the Danube Delta**

The original Reviewer's comments and suggestions are shown in regular typeface, while our responses are shown in italics. The line and figures numbers we use refer to the revised document.

**R1.1** The manuscript "Modelling river-sea continuum: the case of the Danube Delta", by Ferrarin et al., describes a new model configuration for the river-lagoon-sea interconnections in the Danube Delta Region. The model is first validated under current conditions and then used to explore the temporal and spatial variability of oceanographic parameters and three alternative reconnection options.

Evaluation: overall the manuscript covers and interesting topic in a region and type of system that could be extrapolated to other similar systems. The manuscript also does a good job (through visuals) at showing the spatial and temporal variability of oceanographic parameters. However, there are questions about the datasets used, the validation of the model results and the interpretation of results could be strengthened. The manuscript would benefit from some reorganisation since the findings and the implications are difficult to follow beyond the focus on the local issue. The scientific questions and relevance are not clearly articulated in the Introduction and some of them appear further down in the Methods. New results are presented in the Discussion and on the other hand the results of the study (both those in the result section and those in the discussion section) are not put into the broader context and remain descriptive. Perhaps the validation should go in supplementary materials and the result figures currently in the discussion (11-14) could be presented in the Results section highlighting how they relate to current understanding of this type of systems. The Conclusion restates the local findings and should be strengthen by highlighting why or how this piece of work is of general interest.

*Response. We thank the referee for the in-depth and useful review. We appreciate the comments and we improved the manuscript in accordance with all suggestions. In particular, we restructured the manuscript presenting all results in the "Results" section, while the interpretation of our findings within a broader context have been included in the "Discussion" section. The conclusions have been improved by including the key findings.*

**R1.2** The modelling system and Numerical experiments. Significant details on the datasets and the modelling system need to be included without these details it would be difficult to reproduce the results and to evaluate the validity of the numerical modelling: What is the bathymetric resolution of the different datasets? In coastal environments the bathymetry has a non-negligible impact on the obtained results. What is the temporal resolution of the Black Sea lateral boundary condition? Some of the datasets are only available upon request so it will be challenging to reproduce these results. What is the temporal resolution of the model outputs?

*Response: We thank the reviewer for highlighting this issue. In the revised version of the manuscript (lines 90-98), we included more details about the modelling application setup. In particular, for the bathymetric dataset we specified that:*

- *the 2022 European Marine Observation and Data Network dataset (EMODnet Bathymetry Consortium, 2022) for the shelf sea on a regular grid of 1/16\*1/16 arc minutes, ca. 115 metre grid;*

- *the 2024 dataset for the Razelm Sinoie Lagoon System acquired on (mostly West-East-oriented) transects spaced 450 m apart on average and covering the whole system. The distance between two points within each transect is ~1 m.*

- *three separate multibeam datasets (provided at a ~1 m resolution) for the main river branches: the 2023 dataset for Chilia; the 2019 dataset for Sulina; the 2016-2017 dataset for Sf. Gheorghe. Available sparse data was used for some secondary branches and small channels.*

*The choice of saving daily model results was made to limit the volume of model output and is justified by the fact that the boundary conditions for the Black Sea and the Danube River river have a daily frequency.*

**R1.3** The numerical experiments: it is not clear for which period are the what-if scenarios run (L126). Please clarify whether it is 2015-2019 or 2018.

*Response. We added at lines 140-143 a sentence to specify that the period, parametrization, forcing and boundary conditions considered in the what-if numerical experiments are the same adopted in the reference run.*

**R1.4** L123 highlights why this study matters and yet it only appears here. This should we moved to the Introduction section.

*Response. According to the reviewer's suggestion, the mentioned reference, and the related sentence, have been moved to section 1.1.*

**R1.5** L131 you refer to four datasets not three.

*Response. We corrected the text accordingly.*

**R1.6** The model validation: my main concern is the lack of validation data in terms of the water division in the lower parts of the delta river network and salinity on coastal areas.

*Response. We acknowledge the reviewer's concerns; however, the area of interest is poorly monitored. Limited spatial and temporal coverage of existing monitoring networks, along with restricted (freely) available data, are critical issues in the Danube Delta. To address this, we reached out to various authorities and research centers to gather data on bathymetry, river discharge, water level, temperature, and salinity for use in the development and validation of our model. We utilized all available datasets, and while we agree that validation could be further enhanced, we believe the results demonstrate the robustness of our model. We hope this modelling effort will encourage more effective data sharing among institutions and contribute to advancing research in this ecologically significant transitional environment.*

*By analyzing the literature, we found that the simulated distribution of the Danube's mean discharge between the Chilia, Sulina and Sf. Gheorghe branches (46.3 %, 29.7 % and 25.7 %, respectively) is similar to the results reported by Nichersu et al. (2025) (45 %, 34 % and 21 %, respectively). To further elaborate on the model's characteristics, requirements and limitations, a new sub-section (4.1) has been added to the manuscript.*

**R1.7** Fig 2. What are the red circles and the black diamonds in Fig2b and 2c? There is no legend for those. There might be a typo on the x and y axis of Fig 2b and Fig 2c, shouldn't it be "m³/s" instead of "m"?

*Response. The figure's caption have been improved by adding the following sentence "The gray diamonds and the red lines in panels b and c represent the scatter data and the line of best fit, respectively." We removed the red cicles representing the percentiles.*

**R1.8** The validation data for sea level and sea temperature is relatively far from the zone of interest where the manuscript focuses. Why is the validation for sea level done for mean sea level and not at the time the measurements are collected? Does the validation improve when done for mean daily values or is it that the model output are daily values? This is not clear. Validation of sea level is quite poor at times (e.g. differences of over 0.1m for 0.7m variation in sea level Fig.3). The reasons (L171) for the poor fitting should be better explained/explored, does the bathymetry play a role as well in this? Why are results not presented as scatter plots?

*Response. See the response to comment R1.6 regarding the model validation. The sea level and sea temperature validation has been performed on daily values because this is the model output frequency (see the response to comment R1.1). It is worth noting that the tide in the area of interest is negligible and therefore the sea level oscillations are mostly influenced by the open sea conditions and wind and pressure associated with atmospheric perturbations having a typical time scale of 1-10 days.*

*We decided to present the sea level validation as a timeseries instead of a scatter plot to visualise the amplitude and duration of the sea level variability in the area of interest. The results of the statistical analysis are reported in the text (please note that some values have changed by re-performing the validation).*

*As mentioned in the manuscript, the SHYFEM model's performance is strongly related to the capacity of the Black Sea Physics Reanalysis (hereinafter BLKSEA) in reproducing sea level oscillations. To better explore this dependence, we reported in Fig. 1 both the timeseries of the sea level simulated by our SHYFEM application and by the Black Sea Physics Reanalysis.*

*The figure clearly shows the strong dependency of the SHYFEM sea level results on the imposed boundary conditions. However, the statistical analysis reported in Table 1 demonstrates that SHYFEM is performing slightly better than BLKSEA. This is due to the higher resolution of the SHYFEM model application at the coast, which allows to represent coastal dynamics better.*

Table 1: Statistical analysis (in terms of RMSE and CC) of simulated sea levels at the monitoring stations.

| Station | RMSE (cm) | | CC | |
|---|---|---|---|---|
| | SHYFEM | BLKSEA | SHYFEM | BLKSEA |
| Constanta | 6.5 | 7.6 | 0.66 | 0.58 |
| Mangalia | 7.8 | 8.3 | 0.55 | 0.51 |

*Unless specifically requested by the reviewer, we do not intend to include a validation of the Black Sea Physics Reanalysis in our manuscript, either in the form of time series figures or statistical metrics in the text.*

**R1.9** Similarly for the sea surface temperature, why is the validation for Constanta not shown in Fig. 4 while data is available? Temperature is usually a minor part in estuarine dynamics

[Figure]

Figure 1: Observed (red line) and simulated (black dashed line) sea levels at Constanta (top panel) and Mangalia (bottom panel) for year 2017.

with most of the density gradient being driven by salinity differences. However, here the validation is presented in terms of temperature and yet you focus later on salinity results, are there no data available in terms of salinity? There are 6 satellite control SST points, however a single set of statistical parameters are given in L182. What are the implications of comparing averaged daily values with specific timings in the satellite data?

*Response: The validation for Constanta was not presented to limit the number of panels in Fig. 4. The results of the statistical analysis reported in Table 1 demonstrate the good performance of the model at all three sea temperature monitoring stations. We can include the timeseries of the sea temperature in Constanza, if required by the reviewer.*

*We are aware that salinity plays a major role in driving estuarine dynamics but, unfortunately, the area of interest is poorly monitored and, to our knowledge, no salinity observing stations exist (see the response to comment R1.6).*

*Satellite sea surface temperature values are provided at midnight to reduce the errors related to the impact of diurnal warming on the skin-to-bulk temperature conversion. Water temperature values were extracted from the simulation results in the surface layer at the location and time corresponding to the satellite SST data. Due to the limited number of available values at each control point, we prefer to apply the statistical analysis to a dataset containing all samples.*

**R1.10** I think that if this validation section could go into supplementary materials to streamline the paper.

*Response: This paper presents the first comprehensive modelling study of the Danube Delta, and we believe the validation must be presented in the main text.*

**R1.11** Table 1 could include the statistical analysis for all the validation datasets (i.e. also include sea level and satellite SST) and not just temperature for monitoring stations.

*Response: Following the reviewer's suggestion, we included a table reporting the statistical analysis for all the validation datasets (included here as Table 2) at the beginning of the "Model validation" section.*

Table 2: Statistical analysis of simulated river discharge, sea level, sea temperature and sea surface temperature.

| Variable | Station | N data | RMSE | BIAS | CC | SLOPE |
|---|---|---|---|---|---|---|
| River discharge $(\mathrm{m^3\ s^{-1}})$ | Chilia | 120 | 158 | -46 | 1.00 | 0.90 |
| | Tulcea | 120 | 158 | 43 | 1.00 | 1.10 |
| Sea level (cm) | Constanta | 624 | 6.5 | - | 0.66 | 0.70 |
| | Mangalia | 722 | 7.8 | - | 0.55 | 0.62 |
| Sea Temperature (°C) | 15360 | 972 | 1.7 | 0.2 | 0.98 | 1.08 |
| | Constanta | 966 | 1.6 | -0.4 | 0.97 | 0.98 |
| | Mangalia | 908 | 1.5 | -0.2 | 0.97 | 0.96 |
| SST (°C) | Razelm Sinoie | 135 | 1.0 | 0.3 | 0.97 | 0.99 |

**R1.12** Water division: water division is only validated in the upper part of the delta, and only temperature values are shown for a point in the vicinity of the outlets. Without discharge measurements in the network or temperature or salinity in the outlets it is difficult to assess how well or bad the model is performing. The water division is likely to influence the coastal dynamics and the plumes observed. These results also don't seem to be further explored in the paper.

*Response: The numerical model application has been designed to resolve the more relevant water courses and can therefore be used to estimate the water discharge distribution in the delta. We believe that, even if only partially validated, the results presented in section 3.2 and Fig. 5 provide significant insights on the river-sea dynamics which are worth publishing. See also the response to comment R1.6 regarding the model validation.*

*We included at lines 235-244 the following paragraph to highlight the influence of the water division on the plume dynamics: "In front of the Danube Delta, the general coastal circulation (determined averaging the values over the whole simulated period) reflects these processes with the several branches of the multiple-mouth delta forming separated freshwater plumes having shape and dimension defined by amount of water carried out by the different river branches and the coastline characteristics (Fig. 5a). Indeed, the largest plume is found south of the Sulina mouth, where the 5 km long artificial jetty enhances the offshore spread of riverine waters and creates a well-defined recirculation structure. It has to be noted that this plume is reinforced by the freshwater discharged by the nearby Chilia mouth. Well-defined plumes can be also recognized out of the Sf. Gheorghe, Novo Stambul and Potapov mouths. On average, the ROFI associated with the Danube River extends for about 15 km offshore the river mouths. As illustrated in Fig. 5d, the freshwater inputs determine a stratified water column along the coast with Black Sea waters (defined here as having salinity higher than 16 g $L^{-1}$) located on average below 5 to 10 m from the surface."*

**R1.13** Spatial and temporal variability of coastal dynamics. L216 please define the scales of variability you refer to in the text. Here it is implicit you are considering seasonal scales as per your figure 6.

*Response: We modified the sentence to clarify that we are here presenting the results for two specific events having different hydro-meteo-marine conditions: (1) a summer event with peak river discharge (13,000 $m^3$ $s^{-1}$) and calm weather (16 June 2019), and (2) an autumn event with low river discharge (2,400 $m^3$ $s^{-1}$) and windy (northerly) conditions (11 November 2019). The seasonal scale analysis is reported in Fig. 7.*

**R1.14** L234 there is no correlation calculated here, suggest change "are highly correlated" to "can be explained by".

*Response: We corrected the text following the reviewer's suggestion.*

**R1.15** River lagoon connectivity: Over which period are calculated the averages and estimates in L241 to L249?

*Response: We mentioned (lines 210-211) that unless otherwise specified, the values reported in the manuscript refer to averages over the whole 2015-2019 simulation period.*

**R1.16** what causes the difference in WRT in the different years in L253 to L256? Are the weather regimes very different? Is it the different river discharge? It would be good to further explore the reason for these differences.

*Response: We found an error in the WRT values reported in the original manuscript: the minimum (181 days) and maximum (333 days) basin-wide WRT values are found in 2018 and 2017, respectively. To further explore the role of forcing on the WRT computation, we analysed the characteristics of river discharge and wind in different years. The main findings are reported in the following sentences, which have be included in the revised manuscript at lines 308-314: "The spatial and temporal variability of river, ocean and meteorological conditions affects the river-lagoon-sea fluxes as well as the internal mixing in the lagoons, and consequently the WRT computation. Indeed, the difference in WRT across the different years primarily reflects the freshwater input into the lagoons that mainly drives the river-lagoon-sea fluxes and therefore the flushing of the lagoon waters. Indeed, the minimum (181 days) and maximum (333 days) basin-wide WRT values are found in the flood (2018) and drought (2017) years, respectively (Fig. 2). A secondary, but not negligible, role is played by the wind which, in 2018 was characterized by frequent and intense Northerlies that enhanced internal mixing and favored the outflow from the lagoon towards the sea."*

**R1.17** L259 The sense of the gradient and the number of days seem reversed.

*Response: We corrected the sentence as "... a marked east-to-west WRT gradient (from 50 to more than 300 days) is evident ...".*

**R1.18** Assessment of lagoon-sea reconnection solutions L270 please indicate for which period where the what if scenarios run.

*Response: We indicated the 2015-2019 period. See the response to comment R1.3.*

**R1.19** You could combine Fig. 8 and Fig. 9 and present current Fig. 9 as differences with respect to the reference run instead of absolute values. That way the differences would be more easily perceived.

*Response: Following the reviewer's suggestion, we created a figure (included in this document as Fig. 2) presenting the results of the reference simulation as absolute values and of*

*the reconnection scenarios as differences with respect to the reference run. In the revised version of the manuscript, we replaced Fig. 8 and removed Fig. 9.*

[Figure]

Figure 2: Average water renewal time and salinity over the Razelm Sinoie Lagoon System for the reference simulation (as absolute values) and the reconnection scenarios A, B and C (as difference with respect to the reference run). The red dots in panel *e* indicate the location of the two control points where the salinity timeseries were extracted (Fig. 11).

**R1.20** L277-282: please point to table 2

*Response: We added the reference to Table 2.*

**R1.21** Fig 10, please change the colours they're not very colour blind friendly particularly sol.A and sol. B.

*Response: Following the reviewer's suggestion, we changed the lines' colours in the mentioned figure (included in this document as Fig. 3).*

**R1.22** Discussion. The discussion further presents new results (there are 4 results figures in this section) and while they are interesting, they're not put in context or related to similar systems. I believe the discussion would be strengthened if these new 4 figures were moved into the results section and the results were further discussed and contextualised in the discussion. It is difficult to see what is of interest beyond the regional area and what learnings could be taken to other regions.

[Figure]

Figure 3: Timeseries of modelled salinity extracted in the control points in the Razelm (top panel) and Sinoie (bottom panel).

*Response: Following the reviewer suggestion, all results have been presented in the "Results" section, while the interpretation of our findings within a broader context have been included in the "Discussion" section.*

**R1.23** L305-309: I believed this is the first time that stratification is mentioned in the manuscript. So far, the analysis has been limited to the surface. Beside there is no reference to the different wind regimes, could you please elaborate on its influence on the circulation and vertical mixing patterns? Fig 6b and 6c correspond to surface salinity and currents.

*Response: We moved these results and the related figure in section 3.2. Moreover, to investigate the influence of river discharge and wind on the coastal dynamics, we added to Fig. 5 (here Fig. 4) the plots of salinity along transect N-S. Such a figure, as well as the following text have been included in the revised version of the manuscript (lines 242-244 and 255-260).*

*"As illustrated in Fig. 5d, the freshwater inputs determine a stratified water column along the coast, with Black Sea waters (defined here as having salinity higher than 16 g $L^{-1}$) located on average below 5 to 10 m from the surface.*

*During peak river flow and northerly conditions, vertical mixing processes near the coast occupy the whole water column (Fig. 5e). On the contrary, during low river discharge, the surface coastal dynamic is mainly driven by the wind. The autumn event presented in Fig. 5c is characterized by a general northward surface transport of saline waters with the ROFI limited to river plumes extending north-eastward for a few km from the river mouths. During such an event, the water column in front of the delta is well mixed except*

*for a surficial 2 m thick layer in front of the main river branches (Fig. 5f)."*

[Figure]

Figure 4: Surface salinity and current velocity maps, and N-S salinity transects: a) and d) average values over the 2015-2019 period; b) and e) instant values on 15 June 2019; c) and f) instant values on 11 November 2019. The arrows in the top right corner of panels b and c indicate the wind direction.

**R1.24** L306 This is the first reference to the stratification of the water column and the stratification has not been shown anywhere in the paper, please point the reader to the figures as needed or rephrase.

*Response: We rephrased this sentence. See also the response to comment R1.23.*

**R1.25** L309 Fig6b and 6c only show surface, not vertical processes as indicated, please rephrase. The wind regimes are not shown anywhere in the paper so it is difficult to follow the reasoning.

*Response: We rephrased this sentence. The wind arrows have been included in the figures Fig. 4b and c. See also the response to comment R1.23.*

**R1.26** L315 Upwelling is wind driven although it may interact with river plumes. You may wish to clarify the reasoning in this paragraph.

*Response: We changed the mentioned sentence as (lines 265-266) "The presented analysis indicates that these peculiar features are generated by upwelling processes induced by the action of southerly winds blowing along the coastline and interacting with the river outflow."*

**R1.27** L325 is the first reference to salt intrusion, there is no previous information or profile to assess the type of salt intrusion. You could include a profile along the main branches in Fig., 12.

*Response: We have removed the section describing saltwater intrusion because, although this phenomenon poses a serious threat to several coastal areas compromising freshwater*

*supplies for agriculture and human use (Li et al., 2025), it has not yet been reported as a major issue in the Danube Delta. A mention of the saltwater intrusion findings has been included in the discussion (lines 428-436).*

**R1.28** Fig 13, please change the colours. Sol. A and Sol. B are difficult to distinguish. It is not clear how or where the differences are calculated. Is this over the whole lagoon? Is it the differences between two points one in the lagoon and one at sea?

*Response: Following the reviewer's suggestion, we changed the lines' colours in Fig. 9 (included in this document as Fig. 5). We corrected the figure's caption to clarify that the image represents the average (over 2015-2019 period) water levels along the AB transect crossing the Razelm Sinoie Lagoon Systems and indicated with a red line in the right panel.*

[Figure]

Figure 5: Average (over 2015-2019 period) water levels along the AB transect crossing the Razelm Sinoie Lagoon Systems indicated with a red line in the right panel. Background: ©OpenStreetMap contributors 2024; distributed under the Open Data Commons Open Database License (ODbL) v1.0.

**R1.29** L337 please clarify what you mean by hydraulically limited here. You again refer to the wind here, but you do not present any information about the wind. You could include wind information in the figure in the same way that you include the river discharge.

*Response: We modified the sentence as (lines 290-293) "The water level jumps between the two lagoons and between the Sinoie Lagoon and the open sea indicate that the water exchange between the different water bodies is limited by the transport capacity of the narrow and shallow connecting canals (Canal 2, Canal 5, Edighiol and Periboina inlets)."*

*As mentioned in section 1.1, the two major wind regimes characterizing the study area are from north-east, being the most intense, and south-south-west, that can drive alongshore water and sediment transport (Dan et al., 2009). To provide more information about the wind variability, we included the wind speed and direction in Fig. 8 (see the response to comment R1.30).*

**R1.30** L340. Fig14 would benefit from including wind regime.

*Response: Following the reviewer's suggestion, we included the wind speed and direction in Fig. 8 (included in this document as Fig. 6).*

**R1.31** L345-347 please clarify what you mean by with hydraulically controlled.

*Response: The mentioned sentence have been modified as (lines 285-287) "It must be noted that the water flux is not linearly dependent on the water level gradients confirming*

[Figure]

Figure 6: (a) Daily values for the year 2018 of Danube River discharge, (b) wind speed and direction (0°means a northerly wind and 180°indicates a southerly wind) in the RSLS, (c) simulated sea-lagoon (Sinoie) water level difference, (d) simulated sea-lagoon water (in $m^3\ s^{-1}$) and salt (in $10^{-1}\ kg\ s^{-1}$) fluxes. Positive values of water and salt fluxes indicate inflow into the lagoons, while negative values indicate outflow from the lagoon to the sea. Model results are from the reference simulation.

*that the flow between the lagoon and the sea is limited by the transport capacity of the Edighiol and Periboina inlets (ad example at the beginning of March)."*

**R1.32** Conclusions. The conclusion could be strengthened by highlighting why or how this work is relevant beyond the study area at present it just restates what the paper does, and it is difficult to see why it would be of interest of the broader community beyond the study area. Could this be further explored?

*Response: As mentioned in the response to comment R1.22, the interpretation of our findings have been included in the "Discussions" section. Following Referee#2's suggestion, the conclusions have been improved by including the key findings of this study.*

**R1.33** L382 "four lagoon-sea reconnections" I believe you explore 3 different options.

*Response: We corrected the text.*

**References**

Dan, S., Stive, M., Walstra, D.-J. R., and Panin, N.: Wave climate, coastal sediment budget and shoreline changes for the Danube Delta, Mar. Geol., 262, 39–49, https://doi.org/10.1016/j.margeo.2009.03.003, 2009.

EMODnet Bathymetry Consortium: EMODnet Digital Bathymetry (DTM 2022), https://doi.org/10.12770/ff3aff8a-cff1-44a3-a2c8-1910bf109f85, 2022.

Li, M., Najjar, R. G., Kaushal, S., Mejia, A., Chant, R. J., Ralston, D. K., Burchard, H., Hadjimichael, A., Lassiter, A., and Wang, X.: The emerging global threat of salt contamination of water supplies in tidal rivers, Environ. Sci. Technol. Lett., 12, 881–892, https://doi.org/10.1021/acs.estlett.5c00505, 2025.

Nichersu, I., Livanov, O., Mierlă, M., Trifanov, C., Simionov, M., Lupu, G., Ibram, O., Burada, A., Despina, C., Covaliov, S., Doroftei, M., Doroşencu, A., Bolboacă, L., Năstase, A., Ene, L., and Balaican, D.: Chapter 6 - The Danube Delta - The link between the Danube River and the Black Sea, in: The Danube River and The Western Black Sea Coast, edited by Bloesch, J., Cyffka, B., Hein, T., Sandu, C., and Sommerwerk, N., Ecohydrology from Catchment to Coast, pp. 107–122, Elsevier, https://doi.org/10.1016/B978-0-443-18686-8.00002-0, 2025.

**Responses to the Referee#2 Comments and Suggestions**

Journal: Ocean Sciences (OS)
Manuscript number: egusphere-2025-606
Manuscript title: Modelling river-sea continuum: the case of the Danube Delta

The original Reviewer's comments and suggestions are shown in regular typeface, while our responses are shown in italics. The line and figures numbers we use refer to the revised document.

**R2.1** The paper introduces a rather unique SHYFEM configuration that is integrating the Danube delta and the RSLS lagoon system into a coastal model for the western Black Sea. This unified approach to modelling the land-estuary-sea continuum is demanding in terms of model numeric and physics. The authors have carried out a thorough study and achieved good and relevant results that were used to estimate the water transport and hydrographic conditions in the Danube delta system.

The following includes some general comments, followed by a more detailed review of the paper.

*Response: We thank the referee for the in-depth and useful review. We appreciate the comments and we improved the manuscript in accordance with all suggestions. In particular, we restructured the manuscript to better clarify the scope and content of this study.*

**R2.2** The authors carried out a thorough research study, but could have done more to analyse the data in a more comprehensive way targeting well-defined research questions. I am missing a consistent storyline leading through the paper. This is the case for both the entire paper and individual chapters. As it is, the authors have basically written a general study of the Danube river and Razelm Sinoie Lagoon System (RSLS), which provides a lot of information, but is not embedding it into a consistent story-line. The manuscript is actually two papers in one, each with its own research question. The first paper is (1.) demonstrating that the river-sea continuum can be modelled successfully using unstructured grid models and (2.) the second paper is studying the impact of openings of the RSLS towards the Black Sea and its impact on local circulation pattern and hydrographic conditions. I would suggest the authors to define one or two research questions and then to develop a story line addressing this questions. With this, they could streamline the whole paper and make it more concise.

In the first paper (1.), I would expect to find an analysis of the impact of coastal high-resolution configurations on river plume modelling: comparison with and without the Danube river model and RSLS lagoon model. The paper could demonstrate the effect of improved river-plume-modelling on the freshwater distribution, pollutant distribution, etc., with impact on the local hydrographic conditions. To be true, this has been done, but the analysis could have been further expanded and could have been presented with a view to answer the research question rather than to present model results. The analysis could also include a comparison of modelled river data with either observations or hydrological model results (E-hype, SWAT, etc.). I think E-hype climatology is freely available.

The subject of the second paper idea (2.) mentioned above, the study of the effects of openings in the RSLS towards the Black Sea on the hydrographic conditions is only shortly covered in the paper. The analysis of impacts could be extended. The model results could be analyzed in the light of the objectives of the openings, which are not very clear to me. Here, too, I would prefer if the model results could be used to answer the research question rather than being presented.

*Response: We thank the referee for the comment, which has helped us to clarify the scope and content of our paper. In the "Introduction" section we specify that "The paper focuses on the investigation of the hydrodynamic processes, water exchange and connectivity among the different interconnected water compartments (river branches, channels, lagoons, coastal sea) forming the Danube Delta river-sea continuum. To achieve this goal, we implemented the SHYFEM (System of HydrodYnamic Finite Element Modules; Umgiesser et al., 2022) model to the entire Danube Delta, covering about 500 km of the river network, the Razelm Sinoie Lagoon System and part of the prodelta coastal sea (Fig. 1). The model results are used to quantify water discharge distribution among the river branches, to evaluate the effects of multiple river plumes on the coastal dynamics, and to investigate the water exchange and the renewal capacity of the Razelm Sinoie Lagoon System. Moreover, the numerical tool is used to assess the potential impacts of different hypothetical lagoon-sea reconnection solutions (what-if scenarios) on the processes regulating the exchanges between the river, lagoon, and sea."*

*The analysis of the numerical model results have been improved to investigate the processes regulating the exchanges among the different water compartments.*

*We thank the reviewer for the suggestion of using hydrological model results for validating the SHYFEM model. Unfortunately, none of the available model datasets (EFAS, E-Hype, HERA) provides the water division in the river network of the Danube Delta, and, therefore, cannot be used in the model validation (see also the response to comment R2.31).*

**R2.3** The quality of the writing varies. I would strongly recommend to improve the orthography and grammar. Sometimes the construction of the sentences is not correct. Furthermore, the style is often rather direct, and focused on presenting facts. This is often done in loosely connected paragraphs, which could be better integrated.

*Response: The manuscript's orthography and grammar have been improved.*

**R2.4** The different measures of water transport and mixing: ROFI, WFT, WRT could be introduced in a combined and more consistent way in the method section. Currently there is only WRT defined, in the part of thepaper that is dealing with the SHYFEM model (Line 77-87). WFT is later on defined when using it. A clear definition for ROFI has not been provided. It would be good to define these quantities and how they are used in a consistent way. It is for example not clear until later, that the ratio of WFT and WRT is used. This should be done in a separate part of the method section, not in the model description.

*Response: We included in "The modelling system" section (lines 105-115) the following paragraph describing the different measures of water transport and mixing (WFT, WRT, mixing capacity): "To investigate how the difference forcing and processes influence the water mixing and renewal in the semiclosed Razelm Sinoie Lagoon System, the numerical model has been used to estimate two transport time scales: the water flushing time (WFT) and the water renewal time (WRT) (Umgiesser et al., 2014). The basin-wide WFT is defined as the theoretical time necessary to replace the complete volume of the water body*

*with new water and assuming a hypothetical fully mixed basin and is computed diving the basin volume by the volumetric water flux flowing out of the system. WRT is computed by simulating the transport and diffusion of a Eulerian conservative tracer released uniformly throughout the entire lagoon system with a concentration corresponding to 1, while a concentration of zero was imposed on the seaward and freshwater boundaries. The local WRT is considered as the time required for each cell of the RSLS to replace the mass of the conservative tracer, originally released, with new water. The ratio between the basin wide WFT and WRT can be interpreted as an index of the mixing behaviour of the basin. The reader may refer to Cucco et al. (2009) and Umgiesser et al. (2022) for a more comprehensive description of the transport time scales."*

*ROFI is not a variable. This name, which is an acronym for Region Of Freshwater Influence, is commonly used in literature to define a coastal area influenced by the river plume (lines 233-234).*

**R2.5** Chapter 0. Abstract. In its current form, the abstract outlines the scope of the study. It could provide more motivation as to why this study has been carried out. The key findings should be listed as well. It is good to think of the abstract as a mini-IMRAD scheme, including all the parts of the paper.

*Response: We thank the referee for the suggestion. The new abstract now reads: "Understanding water transport and circulation in coastal seas and transitional environments is a key focus of oceanographic and climate research, particularly in recognizing the role of the land-sea interface. The Danube Delta serves as a natural laboratory for river-sea hydrodynamic modeling due to its complex morphology, composed of multiple river branches, channels, and lagoons. Moreover, this coastal environment is subjected to various natural and anthropogenic stressors, and numerical modelling can provide a scientific basis for assessing the impact of human activities. In this work, the SHYFEM finite element hydrodynamic model was applied to the entire river-sea continuum of the Danube Delta region to describe the transport and mixing processes within and between the interconnected water bodies forming the delta. The model was run for the period 2015-2019 and enabled the characterization of: (1) water discharge distribution among the river branches; (2) general hydrodynamic characteristics of the coastal region of freshwater influence; (3) transport time scale of the Razelm Sinoie Lagoon System. Finally, the Danube Delta modeling tool was used to evaluate the potential effects of hydrological reconnection (restoration) measures in the Razelm Sinoie Lagoon System aimed at improving connectivity and water renewal."*

**R2.6** Line 3: The sentence should end after "morphology". Then a new sentence should start.

*Response: See the response to comment R2.5.*

**R2.7** Line 5: "The model was run for several years . . . " How many? "Several year" is a bit vague.

*Response: We modified the sentence as "The model was run for the period 2015-2019 to ..."*

**R2.8** Line 19: Think of a better begin of the sentence than "Modelling these coastal transitional water systems". Are these not "estuaries and coastal seas".

*Response: We modified the sentence as "Modelling estuaries and deltas is challenging ..."*

**R2.9** Line 31: I would say ". . . Danube delta, covering . . . " Please refer to Figure 1.

*Response: We modified the sentence as "... to the entire Danube Delta, covering about 500 km of the river network, the Razelm Sinoie Lagoon System and part of the prodelta coastal sea (Fig. 1)".*

**R2.10** Line 32 "The manuscript ...", maybe better: "The paper". You can also use the active voice and write "We focus ...'.

*Response: We modified the sentence as "The paper focuses ..."*

**R2.11** Note: This part of the introduction gives an overview of the scope of the paper. The different points could be used to identify the research question. The advantage of a good research question is that it describes the problem and the motivation for solving the problem.

*Response: We modified this part of the introduction to clarify the scope of the paper. See the response to comment R2.2.*

**R2.12** Line 36: When we talk about what-if scenarios here, then they must at least described in general terms. It must also be explained what a what-if scenario is. It is not a commonly used term.

*Response: We modified the sentence (lines 36-38) to clarify that the numerical tool is used to assess the potential impacts of different hypothetical lagoon-sea reconnection solutions (what-if scenarios) on the processes regulating exchanges between the river, lagoon, and sea.*

**R2.13** Line 44: Please rewrite the sentence starting with "The Romanian part ..." Line 46: "extends on about", use "extends for about" Line 51: "were finalized ... " not "ended up" Line 51: "As a result, more fresh water is discharged into the lagoon system"

*Response: We corrected the mentioned sentences following the reviewer's suggestions.*

**R2.14** Line 54: Please mark Portit or use another way to show it on the map. It is not good to say: "near reconnection option A", before these have been introduced. The same is true for "reconnection option C" in Line 58.

*Response: Following the reviewer's suggestion, we modified Figure 1 to indicate the two former inlets (included in this document as Fig. 1).*

**R2.15** Line 55: It's not intuitive that a coastal defense structure could enhance coastal erosion. This sentence could be reformulated or explained in more detail.

*Response: The mentioned sentence have been removed since it is not relevant to the purpose of this study.*

**R2.16** Line 62: The sentence starting with "Anyway" should be rewritten, like: As part of the master plan for the protection of the Romanian Littoral against erosion, a major hydraulic engineering project is currently implemented, to ensure a permanent water exchange through the Periboina Inlet. This is an example. The structure of many sentences in the document could be improved.

*Response: We corrected the sentence in accordance with the suggestion.*

**R2.17** Line 65: What does "lower part" mean? Please re-write: "... average water discharge of [so much], with values ranging from ..."

*Response: We modified the sentence as "The Danube River before the delta has an average water discharge of 6500 $m^3$ $s^{-1}$, with values ranging from 1300 to 16000 $m^3$ $s^{-1}$ (Pekárová et al., 2021)".*

[Figure]

Figure 1: (a) Unstructured numerical grid and bathymetry of the hydrodynamic model of the Danube Delta and Black Sea shelf with the red dots and the green triangles marking the sea temperature and sea level monitoring stations, respectively; (b) zoom of the grid over the Danube Delta with the blue bars near Ceatal Izmail indicating the river discharge monitoring stations; (c) zoom of the grid over the Razelm Sinoie Lagoon Systems with the red bars illustrating the considered reconnection solutions and the yellow diamonds marking the satellite SST control points. Background: ©OpenStreetMap contributors 2024; distributed under the Open Data Commons Open Database License (ODbL) v1.0.

**R2.18** Line 77-88: I would suggest to move this paragraph to a different part of the method section and to enhance it. I don't think it makes sense here in the SHYFEM related part of the document. The WRT parameter should be introduced after the model. The other variables ROFI, WFT should be introduced as well. The motivation for using these variables to study water transport through the lagoon system should be clear from the beginning.

*Response: As mentioned in the response to comment R2.4, the description of WFT, WRT and ROFI have been enhanced.*

**R2.19** Line 86: Does WRT really measure the time until the concentration fall to zero or is the time until they fall below a small value enough? I could imagine that it takes a long time until absolute zero is reached.

*Response: We modified the sentence as "The local WRT is considered as the time required for each cell of the RSLS to replace the mass of the conservative tracer, originally released, with new water."*

**R2.20** Line 86: I assume the water parcels are grid cells inside the investigated water body, i.e. the lagoon.

*Response: Yes, you are correct. See the response to comment R2.19.*

**R2.21** Figure 1a: The sea level and sea temperature stations could be marked a bit more clearly

*Response: We increased the size of the markers.*

**R2.22** Line 99-129: I would suggest to restructure this part of the document and to combine the information on model configuration: bathymetry data, boundary data, initial conditions,

forcing data (atmosphere, river), etc. into one part. The first paragraph of 2.2 Numerical experiments is actually presenting the model configuration. It should be included here. The second paragraph of 2.2 Numerical experiments is actually belonging to the model settings that should be part of the SHYFEM related part of the method section. The information on the numerical experiments: (a.) the model validation and water transport assessment and (b.) for the what-if scenarios should be presented in a separate section. Here you could provide more background information on the choosen simulations.

*Response: Following the reviewer's suggestion, we restrucured the "Numerical experiments" section by including:*

- *in section 2.1 the description of the SHYFEM model, the numerical grid, the bathymetric dataset, the model setting and the methods for computing WFT, WRT and mixing capacity;*

- *in section 2.2 the description of the simulations duration, the forcing and boundary data, the initial conditions, and the what-if scenarios. In this last part, we provided more background information on the choosen reconnection solutions.*

**R2.23** Line 101-105: Please add the resolution of the input data. I assume that you used gridded data products. Who provided the data for the Razelm Sinoie Lagoon and the river branches?

*Response: We integrated the description of the bathymetric datasets as (lines 90-98): "The model bathymetry is obtained by a bilinear interpolation on the numerical grid of the following available datasets (all referred to the Marea Neagra Sulina vertical datum):*

- *the 2022 European Marine Observation and Data Network dataset (EMODnet Bathymetry Consortium, 2022) for the shelf sea on a regular grid of 1/16\*1/16 arc minutes, ca. 115 metre grid;*

- *the 2024 dataset for the Razelm Sinoie Lagoon System acquired on (mostly West-East-oriented) transects spaced 450 m apart on average and covering the whole system. The distance between two points within each transect is ∼1 m.*

- *three separate multibeam datasets (provided at a ∼1 m resolution) for the main river branches: the 2023 dataset for Chilia; the 2019 dataset for Sulina; the 2016-2017 dataset for Sf. Gheorghe. Available sparse data was used for some secondary branches and small channels."*

**R2.24** Line 123-129: Could you provide a bit more information on why these what-if scenarios were chosen? Are these realistic scenarios?

*Response: We integrated the description of the what-if scenarios as (lines 132-142) "In the past, the lagoons were connected to the sea via several inlets, while nowadays only the Periboina and Edighiol connections remain open. Additional numerical experiments were conducted to investigate the potential effects on the lagoons' water renewal and salinisation of different reconnection solutions designed in collaboration with local stakeholders to enhance the river-lagoon-sea exchange. The dredging of a new inlet is under consideration by local communities and authorities, as part of the activities developed under the framework of the Horizon Europe Project DANUBE4all (`https://www.danube4allproject.eu/`). The three what-if scenarios considered in this study consisted of opening one 1.5 m depth and 70 m wide channel to connect the either the Razelm Lagoon (solutions A and B in Fig. 1c) or the Sinoie Lagoon (solution C in Fig. 1c) with the Black Sea. These reconnection solutions are located in the vicinity of previous inlets, now either closed by humans (Portiţa) or clogged (Gura Buhazului inlet, active till*

*the beginning of the 1990s). The period, parametrization, forcing and boundary conditions considered in these what-if numerical experiments are the same as those adopted in the reference run (hereinafter called REF)."*

**R2.25** Line 130-151: In my opinion, it would be better to move the list of validation data sets to the validation chapter. The model validation chapter could be a separate part of the paper, because it is not so much presenting new results, but is demonstrating the quality and usefulness of the model.

*Response: Following the reviewer's suggestion, we moved the description of the validation datasets to the "Model validation" section. This paper presents the first comprehensive modelling study of the Danube Delta, and we believe the validation must be presented in the "Methods" section.*

**R2.26** Line 132-135: Please refer to figure 2 here. The validation could also include a comparison with hydrological model results: (E-Hype, SWAT). I think the E-Hype climatology is freely available. Annual mean discharges could be calculated and compared with hydrological model results.

*Response: We thank the reviewer for the suggestion of using hydrological model results for validating the SHYFEM model. Unfortunately, none of the available model datasets (EFAS, E-Hype, HERA) provides the water division in the river network of the delta, and, therefore, cannot be used in the model validation. See also the response to comment R2.31.*

**R2.27** Line 136-137: Here you say that hourly values were available. Then why did you perform the model validation using daily averaged data sets? You say that the model can represent the sea level fluctuations (anomaly) associated with intense meteorological events (Line 168). But these have much shorter time scales. I assume that at least hourly data would be needed. We use 10 minutes data for sea level warnings.

*Response: I understand that using daily averaged sea levels may seem unconventional; however, this approach is justified because tidal effects in the study area are negligible. As a result, sea level variations are primarily driven by open sea conditions and atmospheric disturbances with typical time scales of 1 to 10 days. Additionally, the model is forced at the Black Sea and Danube River boundaries using daily datasets. Given these assumptions and to limit the model's output volume, results are saved at a daily frequency. Finally, the analysis of sub-daily dynamics is beyond the scope of this study.*

**R2.28** Line 141-151: I'm a bit puzzled that you did not use CMEMS satellite SST product for Black Sea. Is the quality of the CMEMS product not good enough? Why did you use the level 2 product and not the gridded data set?

*Response: The capacity of the model in reproducing the sea temperature in the coastal waters of the Black Sea was assessed through the comparison of in-situ timeseries (Constanta, Mangalia, 15360). Satellite SST data were only used to validate the model in the Razelm Sinoie Lagoon System. We used the level 2 product (kindly provided by colleagues from the University of Stirling) because they were specifically processed for a very shallow environment.*

**R2.29** Line 153-183: The chapter jumps right away into model validation statistical methods and results. It would be good if you could provide a bit more background information and motivate the validation exercise and the specific choice of parameters and methods.

*Response: The "Model validation" section have been reworked to include background information, methods, validation data and validation results.*

**R2.30** Line 154-157: Please split this sentence. It is much too long. Line 155: I assume it is "Pearson correlation" rather than "Pearson cross-correlation" Line 156: "slope of the linear regression best-fit line". I would suggest to write the "best fit calculated by linear regression". Line 160: "The model well represents ... ", change to "The model represents ..... well." Line 162-163: I would suggest avoiding this type of reduced writing using brackets. You can form sentences like: While it is underestimating here, it is overestimating there.

*Response: We revised the manuscript in accordance with the reviewer's suggestions.*

**R2.31** Line 158-165: This is a less comprehensive validation study of the quality of hydrological predictions than I would have expected from a paper focusing on river-to-sea continuum modelling. Only 2 stations close to the Danube source point have been chosen. There is a straight river section running from the source point to the Danube river branching point at Ceatal Izmail where the model is validated. I can only assume that errors accumulate further down the river network. The validation exercise could be extended with a comparison of modelled discharge values using SHYFEM and modelled discharge values using hydrological model (E-Hype, SWAT,...). Maybe a literature study would also provide runoff data that could be used for comparison.

*Response: We acknowledge the reviewer's concerns; however, the area of interest is poorly monitored. Limited spatial and temporal coverage of existing monitoring networks, along with restricted (freely) available data, are critical issues in the Danube Delta. To address this, we reached out to various authorities and research centers to gather data on bathymetry, river discharge, water level, temperature, and salinity for use in the development and validation of our model. We used all available datasets, and while we agree that validation could be further enhanced, we believe the results demonstrate the robustness of our model. We hope this modelling effort will encourage more effective data sharing among institutions and contribute to advancing research in this ecologically significant transitional environment.*

*By analyzing the literature, we found that the simulated distribution of the Danube's mean discharge between the Chilia, Sulina and Sf. Gheorghe branches (46.3 %, 29.7 % and 25.7 %, respectively) is similar to the results reported by Nichersu et al. (2025) (45 %, 34 % and 21 %, respectively). To further elaborate on the model's characteristics and performance, a new sub-section (4.1) has been added to the manuscript.*

**R2.32** Line 162: Are the situations with underestimation at Chilia coinciding with situations of overestimation at Tulcea?

*Response: Yes, we mentioned it in the manuscript at lines 178-179.*

**R2.33** Figure2: Fit2 (a): Is this the river runoff time series at Isaccea? Could this location be marked in Fig 1. What do the different colors of the symbols in Fig2 (b) and (c) represent? There are periods of systematic undeprediction (Constanta: July-September 2027). Could these be linked to meteorological conditions?

*Response: Isaccea is indicated with name and arrow in Fig. 1a. The figure's caption have been improved by adding the following sentence "The gray diamonds and the red lines in panels b and c represent the scatter data and the line of best fit, respectively." We removed the red cicles representing the percentiles.*

*The underestimation and overestimation during flood events in the two branches is likely attributable to the model's inability to correctly reproduce the river overflow into the surrounding floodplains. We expect that meteorological conditions do not significantly affect the river flow in the upper delta.*

**R2.34** Line 168: As mentioned before, I doubt that the model quality with regards to predicting storm events can be validated using daily mean sea level data. It is not possible to do a peak error validation. At least hourly data (which is available, Line 136) should be used.

*Response: See the response to comment R2.27.*

**R2.35** Line 169: It is mentioned here that storm events up to 10 days lead time can be predicted well. This would require a forecast validation (assessing the quality of the forecast according to lead time), which has not been presented. The validation exercise uses daily hindcast data sets.

*Response: The numerical model is run in hindcast mode and no forecasts are presented in this study. We corrected the sentence to clarify that, in the study area, the sea level variations are primarily driven by open sea conditions and atmospheric disturbances with typical time scales of 1 to 10 days.*

**R2.36** Line 170-171: Could the validation results be presented in a table, maybe table 1.

*Response: Following the reviewer's suggestion, the results of the statistical analysis of river discharge, sea level and sea temperature have been reported in a table (included in this document as Table 1).*

Table 1: Statistical analysis of simulated river discharge, sea level, sea temperature and sea surface temperature.

| Variable | Station | N data | RMSE | BIAS | CC | SLOPE |
|---|---|---|---|---|---|---|
| River discharge ($m^3\ s^{-1}$) | Chilia | 120 | 158 | -46 | 1.00 | 0.90 |
| | Tulcea | 120 | 158 | 43 | 1.00 | 1.10 |
| Sea level (cm) | Constanta | 624 | 6.5 | - | 0.66 | 0.70 |
| | Mangalia | 722 | 7.8 | - | 0.55 | 0.62 |
| Sea Temperature (°C) | 15360 | 972 | 1.7 | 0.2 | 0.98 | 1.08 |
| | Constanta | 966 | 1.6 | -0.4 | 0.97 | 0.98 |
| | Mangalia | 908 | 1.5 | -0.2 | 0.97 | 0.96 |
| SST (°C) | Razelm Sinoie | 135 | 1.0 | 0.3 | 0.97 | 0.99 |

**R2.37** Line 172: You mention that sea level prediction errors originate from the reanalysis product that you use at the boundaries. You could validate the CMEMS reanalysis product at the two stations Mangalia andConstanta to demonstrate this. The CMEMS Black Sea MYP QUID unfortunately does not use tide-gauge data for validation.

*Response: To better explore this dependence, we reported in Fig. 2 both the timeseries of the sea level simulated by our SHYFEM application and by the Black Sea Physics Reanalysis (hereinafter BLKSEA).*

*The figure clearly shows the strong dependency of the SHYFEM sea level results on the imposed boundary conditions. However, the statistical analysis reported in Table 2 demonstrates that SHYFEM is performing slightly better than BLKSEA. This is due to the higher resolution of the SHYFEM model application at the coast, which allows to represent coastal dynamics better.*

*It is not our intention, unless specifically requested by the reviewer, to present in our manuscript a validation (either as timeseries in the figure or as statistical metrics in the text) of the Black Sea Physics Reanalysis.*

[Figure]

Figure 2: Observed (red line) and simulated (black dashed line) sea levels at Constanta (top panel) and Mangalia (bottom panel) for year 2017.

Table 2: Statistical analysis (in terms of RMSE and CC) of simulated daily sea levels at the monitoring stations.

| Station | RMSE (cm) | | CC | |
|---|---|---|---|---|
| | SHYFEM | BLKSEA | SHYFEM | BLKSEA |
| Constanta | 6.5 | 7.6 | 0.66 | 0.58 |
| Mangalia | 7.8 | 8.3 | 0.55 | 0.51 |

**R2.38** Line 178: "varies strongly" rather than "strongly varies".

*Response: We corrected the sentence.*

**R2.39** Line 178: Could you write which year.

*Response: We modified the sentence as "… that sea surface temperature in RSLS varies strongly over the 2015-2019 period with values ranging …"*

**R2.40** Line 178-183: Why did you use daily mean values for satellite SST validation? Aren't midnight values usually used to reduce the errors related to the impact of diurnal warming on the skin-to-bulk temperature conversion?

*Response: Yes, they are midnight values. We better described the SST dataset in the "Model validation" section (lines 158-168).*

**R2.41** Line 178-183: could you present the validation results in a figure, maybe even a spatial

distribution of model errors.

*Response: Due to the limited number of available values at each control point, we prefer to apply the statistical analysis to a dataset containing all samples. Moreover, spatially SST in the lagoons has a small spatial variability with difference among stations lower than 1.5 °C.*

**R2.42** After Line 183: The model validation chapter provides a lot of data, but only little analysis. Could you write a paragraph evaluating the model performance and in the light of your model application, i.e. the adequate representation of the river-to-sea continuum for detailed model studies of the Danube delta. I think this is needed.

*Response: A new sub-section (4.1) dedicated to the discussion of the model's characteristics, requirements and limitations has has been added in the manuscript.*

**R2.43** Line 185: First 3 sentences: I think you should rewrite these sentences to make them more clear. I think you want to say that you can only use the model system to estimate the water discharge distribution among the "main" river branches.

*Response: We reformulated these sentences as "The Danube Delta's river network comprises a highly complex system of hundreds of natural and artificial channels, streams, marshes, and lakes, whose morphological complexity exceeds the resolution capabilities of the current model implementation. The model was configured to represent only the most hydraulically significant watercourses, enabling the estimation of water discharge distribution among the principal river branches. "*

**R2.44** Line 188: "estimate the relative load". I think you mean "relative runoff". The term load refers to substances carried with the river, liked pollutants.

*Response: We corrected the mentioned sentences following the reviewer's suggestion.*

**R2.45** Figure 5 and text: Can you provide a number for the total runoff in the considered period 2015-2019. Then readers can calculate the absolute values of river runoff from the percentages presented in the figure.

*Response: We provided at line 174 the value of the average the Danube River discharge imposed at the open boundary of Isaccea (6000 $m^3$ $s^{-1}$).*

**R2.46** Line 194-203: Would it be possible to compare your river runoff data with literature values or values from hydrological models, to get a feeling for the quality of the prediction. This could be done in the validation section.

*Response: See the response to comment R2.26.*

**R2.47** Line 209: Could you briefly introduce the variable ROFI.

*Response: See the response to comment R2.4.*

**R2.48** Line 215: and following: Why do you use the unit g/L and not the more widespread unit psu?

*Response: The numerical model is computing salinity as g/L. It can be converted to Practical Salinity Units (PSU) by understanding that 1 PSU is approximately equal to 1 gram of salt per kilogram of solution.*

**R2.49** Line 218: Is this the 15 th of Jun 2019?

*Response: We corrected the text.*

**R2.50** Line 222: The currents in figure 6 are not very easy to see. Do you know what drives these coastal currents? Are they influenced by steeric effects? In other words, does the amount of discharged freshwater and its distribution affect the coastal currents.

*Response: We increased the arrows' line width. Moreover, following Reviewer#1's suggestion, we added to Fig. 6 (here Fig. 3) the plots of salinity along transect N-S.*

[Figure]

Figure 3: Surface salinity and current velocity maps, and N-S salinity transects: a) and d) average values over the 2015-2019 period; b) and e) instant values on 15 June 2019; c) and f) instant values on 11 November 2019. The arrows in the top right corner of panels b and c indicate the wind direction.

*We improved the description of the spatial and temporal variability of coastal dynamics.*

**R2.51** Line 222 and following: If you want to use standard unit for currents, then you should use m/s.

*Response: We corrected the unit for currents to m $s^{-1}$.*

**R2.52** Line 226-236: What exactly is a seasonal standard deviation? Is it the standard deviation of the month of all years belonging to the season? Could you motivate, why you are doing a seasonal analysis? You say that you are using a seasonal analysis to calculate the standard deviation, but that can not be the motivation for the analysis.

*Response: To clarify the motivation and method of the season analysis, we modified the mentioned sentence as (lines 267-269) "To analyse the temporal variability of the coastal dynamics, the model results were processed to computed the standard deviation (hereinafter STD) of the month of all years belonging to the four seasons (winter=DJF, spring=MAM, summer=JJA, fall=SON)."*

**R2.53** Line 233: "multiple mouth". I would use another term.

*Response: We modified the sentence as (lines 273-274) "The freshwater discharged by the different branches determine a similar coastal salinity pattern in winter (Fig. 7e) and fall (Fig. 7h)."*

**R2.54** Line 205-236: This is just an idea: To show the advantage of using SHYFEM for modelling the Danube delta and RSLS, a comparison of 2 simulations – one including and one excluding the Danube delta and RSLS domain could be presented. In the second type of simulations without Danube delta and RSLS domain, the freshwater discharge could be added to the coarse Black Sea grid. I think the simulation would show that the second type of simulations are less able to produce realistic river plumes and discharge patterns. This is the advantage of resolving the lagoons and estuaries using dynamical models.

*Response: The additional simulation proposed by the reviewer would entail substantial effort and falls outside the scope of the present study. The Black Sea Physics Reanalysis (`https://doi.org/10.25423/CMCC/BLKSEA_MULTIYEAR_PHY_007_004`) serves as an example of a coarse-resolution model that is unable to realistically simulate river plumes and discharge patterns in front of the Danube Delta.*

**R2.55** Line 238-264: Can you provide a motivation for why you want to study River-Lagoon-Sea connectivity? The main purpose seems to be, to calculate the ratio of WFT and WRT. Later, however, only WRT is used for assessing the what-if scenarios.

*Response: We agree with the reviewer that "River-lagoon-sea connectivity" was not an appropriate title for this section. We will change the title to "Lagoons' water exchange, mixing and renewal capacity".*

**R2.56** Line 238: What is a "choked water body". Can you please explain this term or use another one.

*Response: According to (Kjerfve, 1986), coastal lagoons can conveniently be subdivided into choked, restricted, and leaky systems based on the degree of water exchange between lagoon and ocean. Additional text has been added at lines 447-449.*

**R2.57** Line 242-249: Could you rewrite this part and make the different contributions more clear? The following questions may help you. Of the 62 m3/s water discharge from the Dunavat and Dranov channel, are 42 m3/s discharged into the Black Sea? Should the sum of the inflow from the Black Sea (+16 m3/s), the outflow to the Black Sea (42 m3/s), the river/channel runoff (+62 m3/s) and the amount of water lost by evaporation (-20 m3/s) cancel out? When I add all the contributions, I get 16 m3/s, which is equal to the inflow from the Black Sea.

*Response: We reformulated this part of the manuscript to clarify the different contributions to the RSLS's water budget. The text now reads as (lines 295-300) "The RSLS has an average water volume of about 1,300 millions $m^3$ and receives 40 and 22 $m^3\ s^{-1}$ of freshwater from the Dunavăţ and Dranov canals, respectively. This excess water entering the lagoons is primarily discharged into the Black Sea via the Edighiol and Periboina inlets, resulting in a average seaward flow of 58 $m^3\ s^{-1}$. The average inflow of marine water into the RSLS amounts to 16 $m^3\ s^{-1}$. Evaporation over the lagoon system overpasses precipitation resulting in a net loss of 20 $m^3\ s^{-1}$. The lagoons receive a total water flux of 78 $m^3\ s^{-1}$ from the sea and the river. The average fluxes are reported in Table 2."*

**R2.58** Line 266-272: Could you provide a few more words on the different proposals for reconnecting RSLS with the Black Sea? Could you motivate the choice of the proposals.

*Response: A more detailed description of the different reconnecting solutions have been included in "Numerical experiments" section. See also the response to comment R2.24.*

**R2.59** Line 266 and following: What is the time scale of the assessment? How long are the model simulations? Which years do they cover?

*Response: The simulations for the what-if scenarios span the period 2014-2019 (year 2014 is considered as the model's spin-up time and not included in the analysis), consistent with the reference simulation. We added a sentence at lines 140-142 to specify that the period, parametrization, forcing and boundary conditions considered in the what-if numerical experiments are the same adopted in the reference run.*

**R2.60** Table 2: It took me a minute to understand the table. I guess, I expected it also to cover the case of multiple openings: single opening, two openings together, three openings together. Maybe this could be explained somewhere.

*Response: No multiple openings scenarios were simulated. The three what-if scenarios considered in this study consisted of opening one 1.5 m depth and 70 m wide channel to connect the either the Razelm Lagoon (solutions A and B in Fig. 1c) or the Sinoie Lagoon (solution C in Fig. 1c) with the Black Sea. The above sentence have been included in the manuscript at lines 132-140 to clarify the settings of the different scenarios.*

**R2.61** Line 277-297: It should be explained somewhere that changing the net-flow through the lagoon, changes also the sea level in the lagoon. This is visible in figure 13.

*Response: This hydrodynamic feedback is reported in the discussion (lines 357-360): "Connecting the Razelm Lagoon with the Black Sea with solutions A and B do not only allow the inflow of marine waters but also changes the water level of the two basins (red and blue lines in Fig. 9) decreasing the water exchange between the two lagoons and the outflow via the existing inlets (Table 2)."*

**R2.62** Line 283-288: I think it would be easier to analyse difference plots (figure 9).

*Response: Following Reviewer#1's indication, we created a figure (included in this document as Fig. 4) presenting the results of the reference simulation as absolute values and of the reconnection scenarios as differences with respect to the reference run. In the revised version of the manuscript, we discussed the presented findings in details.*

**R2.63** Figure 9: Here in the figure you use the salinity unit psu, whereas in the text you are using g/L.

*Response: The figure have been corrected to ensure that g/L is used consistently throughout the manuscript.*

**R2.64** Line 299: I'm not sure that I understand the structure of the paper. After analyzing the time scales of water transport in the Danube delta system and introducing the what-if scenarios for water transport, now you take one step back and discuss the hydrographic conditions predicted by your model. I would rather suggest move this part to the model study and validation exercise as it is related to the general dynamic in the area. You could focus on the what-if scenarios here.

You could also split the analysis into a Danube and the RSLS part. This way you could have one chapter dealing with the what-if scenarios.

This chapter presents many new results that properly analyzed could form even another paper. I would suggest to take a step back, to define a research question and to restructure the paper accordingly.

*Response: All results will be presented in the "Results" section, while the interpretation of our findings within a broader context will be included in the "Discussion" section. The results of the what-if scenarios have been included as a subsection (3.3.1) of the "Lagoons's water exchange, mixing and renewal capacity" section. The conclusions have been improved by including the key findings.*

[Figure]

Figure 4: Average water renewal time and salinity over the Razelm Sinoie Lagoon System for the reference simulation (as absolute values) and the reconnection scenarios A, B and C (as difference with respect to the reference run). The red dots in panel *e* indicate the location of the two control points where the salinity timeseries were extracted (Fig. 5).

**R2.65** Line 299: The first 2-3 sentences are very general. You could just say that you want to investigate the water exchange between the different parts of the Danube delta to study their impact on the local hydrographic conditions. I guess you want to discuss as part of the what-if scenario studies, but this is not clear her.

*Response: The "Discussion" section have been restructured. See the response to comment R2.64.*

**R2.66** Figure 10: Could you please choose some different colors. I can not see the difference between A and B. Solution C is also rather difficult to see.

*Response: Following the reviewer's suggestion, we changed the lines' colours in the mentioned figure (included in this document as Fig. 5).*

**R2.67** Line 311: "Water bulges". I can only see the temperature distribution, not the sea level. Is this because of the limited transport capacity from the near-shore to the off-shore?

*Response: We will use "patterns" or "features" instead of "water bulges".*

**R2.68** Line 313: I would suggest to avoid constructions like "warmer (colder)".

[Figure]

Figure 5: Timeseries of modelled salinity extracted in the control points in the Razelm (top panel) and Sinoie (bottom panel).

*Response: We removed them.*

**R2.69** Line 314: Upwelling is usually the result of water mass transport, not mixing.

*Response: We reformulated the sentence as (lines 263-264) "The vertical alongshore sea temperature transect presented in Figs. 6c and Figs. 6d indicate an upwelling-driven transport of marine waters from deeper layers to the coastal zone, enhancing mixing between open sea and riverine waters."*

**R2.70** Line 313-317 and figure 11: thanks for the nice plot, but I don't know exactly which point you want to make here. The upwelling event is likely wind driven and would have happened with and without rivers. It is of course clear that the horizontal surface temperature distribution would look different without the implementation of the rivers. But I'm not sure that this is what you want to say, that running a model with an extended Danube estuary results in a better representation of the river plumes.

*Response: Figure 6 illustrates distinctive coastal circulation patterns that emerge off the delta during southerly wind conditions. The interaction between wind-driven coastal upwelling and river outflow gives rise to small-scale nearshore features, situated between river mouths, which exhibit thermo-haline properties distinct from those of the surrounding waters. The model's variable resolution is crucial for capturing the seamless transition from river branches to the coastal sea, enabling the accurate reproduction of such complex coastal dynamics.*

*These results have been moved to the "Results" section.*

**R2.71** Line 318-329: further studies could investigate if the salt water intrusions happen gradually with time or if they are related to certain meteorological conditions and events.

*Response: We have removed the section describing saltwater intrusion because, although this phenomenon poses a serious threat to several coastal areas compromising freshwater supplies for agriculture and human use (Li et al., 2025), it has not yet been reported as a major issue in the Danube Delta. A mention of the saltwater intrusion findings has been included in the discussion (lines 428-436).*

**R2.72** Line 323-324: The locations of maximum salt water intrusion could be shown on the map.

*Response: See the response to comment R2.71.*

**R2.73** Line 330: Here you present the part related to the what-if scenario. This could be combined with the assessment in chapter 3.4.

*Response: Following the reviewer's suggestion, we moved the part related to the what-if scenario to section 3.3.1.*

**R2.74** Line 331: Can you rewrite the sentence starting with "Due to the input . . . ". Results should not be put in brackets. Time periods should be put to the end of the sentence.

*Response: We rewrited the sentence as (lines 289-290) "The freshwater inflow from the Dunavăţ and Dranov canals creates a persistent water level gradient from Razelm Lagoon to Sinoie Lagoon and the adjacent coastal sea (green line in Fig. 9)."*

**R2.75** Line 333: The flow is not necessarily barotropic because of a sea level gradient. You can still have a stratified flow. But your model results should show if at least a seasonal halo-or thermocline in the RSLS exist.

*Response: We thank the reviewer for highlighting this point. We clarified at lines 437-438 that due to the wind energy and relatively shallow water (the average depth is 1.8 m), the water masses are generally vertically well mixed.*

**R2.76** Line 338: I assume that meteorological conventions are used and the winds are from north-easterly directions.

*Response: We corrected the sentence as "... by the dominant winds from north-easterly direction".*

**R2.77** Figure 13: The results in this figure could be analyzed for the different what-if scenarios. The differences in sea level could be related to the differences in water transport (table 2, chapter 3.5).

*Response: We moved these results to section 3.3.1 and discuss them in relation to the water fluxes presented in Table 2.*

**R2.78** Line 349: There could be an easier formulation for the sentence starting with: "The inflow of marine waters."

*Response: We modified the sentence as "Marine waters flow into the lagoon when sea level is higher than the lagoon water level."*

**R2.79** Figure 14: It should be made clear that this figure is using results from the reference simulation.

*Response: We indicated in the figure's caption that the results are from the reference simulation.*

**R2.80** Line 348-363: The discussion here is rather qualitative. Only time series results from the reference simulation are presented. The different what-if scenarios are only discussed in

general terms. I would suggest to extend the analysis and to present quantitative results for the different what-if scenarios. Otherwise, the analysis remains a bit unsatisfying. As mentioned before, this study could be combined with chapter 3.5.

*Response: Following the reviewer's suggestion, we improved the analysis of the what-if scenarios (section 3.3.1).*

**R2.81** Chapter 5: Concluding remarks and perspectives. The concluding remarks focus on the comprehensive modelling tools that have been developed, but they leave the results of the modelling study out. As mentioned before, I would restructure the paper, defining a research question and a story line. The conclusions should summarize the key findings.

*Response: We restructured the manuscript following the reviewer's suggestions. All results have been included in the "Results" section, while the interpretation of our findings within a broader context have been reported in the "Discussion" section. Following the reviewer's suggestion, the conclusions have been improved by including the key findings of this study.*

**References**

Cucco, A., Umgiesser, G., Ferrarin, C., Perilli, A., Melaku Canu, D., and Solidoro, C.: Eulerian and lagrangian transport time scales of a tidal active coastal basin, Ecol. Model., 220, 913–922, https://doi.org/10.1016/j.ecolmodel.2009.01.008, 2009.

EMODnet Bathymetry Consortium: EMODnet Digital Bathymetry (DTM 2022), https://doi.org/10.12770/ff3aff8a-cff1-44a3-a2c8-1910bf109f85, 2022.

Kjerfve, B.: Comparative oceanography of coastal lagoons, in: Estuarine Variability, edited by D. A. Wolfe, pp. 63–81, Academic Press, New York, USA, https://doi.org/10.1016/B978-0-12-761890-6.50009-5, 1986.

Li, M., Najjar, R. G., Kaushal, S., Mejia, A., Chant, R. J., Ralston, D. K., Burchard, H., Hadjimichael, A., Lassiter, A., and Wang, X.: The emerging global threat of salt contamination of water supplies in tidal rivers, Environ. Sci. Technol. Lett., 12, 881–892, https://doi.org/10.1021/acs.estlett.5c00505, 2025.

Nichersu, I., Livanov, O., Mierlă, M., Trifanov, C., Simionov, M., Lupu, G., Ibram, O., Burada, A., Despina, C., Covaliov, S., Doroftei, M., Doroşencu, A., Bolboacă, L., Năstase, A., Ene, L., and Balaican, D.: Chapter 6 - The Danube Delta - The link between the Danube River and the Black Sea, in: The Danube River and The Western Black Sea Coast, edited by Bloesch, J., Cyffka, B., Hein, T., Sandu, C., and Sommerwerk, N., Ecohydrology from Catchment to Coast, pp. 107–122, Elsevier, https://doi.org/10.1016/B978-0-443-18686-8.00002-0, 2025.

Pekárová, P., Mèszáros, J., Miklánek, P., Pekár, J., Prohaska, S., and Ilić, A.: Long-term runoff variability analysis of rivers in the Danube basin, Acta Horticulturae et Regiotecturae, 24, 37–44, https://doi.org/10.2478/ahr-2021-0008, 2021.

Umgiesser, G., Ferrarin, C., Cucco, A., De Pascalis, F., Bellafiore, D., Ghezzo, M., and Bajo, M.: Comparative hydrodynamics of 10 Mediterranean lagoons by means of numerical modeling, J. Geophys. Res. Oceans, 119, 2212–2226, https://doi.org/10.1002/2013JC009512, 2014.

Umgiesser, G., Ferrarin, C., Bajo, M., Bellafiore, D., Cucco, A., De Pascalis, F., Ghezzo, M., Mc Kiver, W., and Arpaia, L.: Hydrodynamic modelling in marginal and coastal seas - The case of the Adriatic Sea as a permanent laboratory for numerical approach, Ocean Model., 179, 102123, https://doi.org/10.1016/j.ocemod.2022.102123, 2022.

---

## Referee Report (RR1)

**Review of "Modelling river-sea continuum: the case of the Danube Delta", by Ferrarin et al.**

This is the second review of the manuscript "Modelling river-sea continuum: the case of the Danube Delta", by Ferrarin et al. Structure and quality of the text was much improved, but several of the more fundamental points have not been fully addressed. The manuscript would still benefit from the development of a more consistent story line and the formulation of a research question. The authors added a research statement in the introduction focusing on process-oriented model analysis, without providing a definite goal for the analysis. As a consequence, the manuscript demonstrates qualitatively the SHYFEM model's ability to produce realistic results, but it does not necessarily provide sufficient information, for all processes and parameters, to verify the accuracy of the model results quantitatively as well. The scope of the model study is rather broad and covers not just river-sea continuum, but also coastal processes, like upwelling. The model validation study is limited by the amount of data used. This is not a reason to reject the findings. I find them reasonable and qualitatively appropriate. But the limitations of the paper should be stated clearly, preferably in the introduction or method section, rather than, as the authors have chosen, in the result-discussion section (4.1).

The structure of the review is as follow: The first part of this review addresses the manuscript chapter by chapter. The second part responds to the authors' reply to the initial review.

**Chapter 1: Line 13-38: Introduction**

The novelty of the paper, to resolve the river-sea continuum within a single unstructured grid model, could be more clearly emphasized. It should be more clearly stated how the presented modelling approach differs from previous studies. Some of this could be achieved by moving parts of section 4.1 (line 360-404) to the introduction section.

**Chapter 2.1: Line 70-115: The modelling system**

The first paragraph (line 71-77) should be modified to remove some errors and make it more readable.

Line 74: Suggestion: The model solves the primitive equations using an unstructured finite element grid in the horizontal and a z-coordinate system in the vertical.

Line 75: The model has been applied ....

Line 76: ... and in other coastal systems.

Line 78: SHYFEM was applied to model the river network ...

Line 90: The model bathymetry is derived by bilinearly interpolating the following datasets onto the model's numerical grid.

Line 99-104: This paragraph contains on the model components, the configuration and the implementation of the model processes all in one. I would suggest to improve the structure of the text and to use references wherever possible. Maybe you can highlight different choices made for dynamical parameter in the open sea and the river network.

Line 103-104: The time frequency of the model output is defined by the dynamic time scales of the processes you intend to resolve and you are able to resolve. If you are able to resolve the processes on the required time scales within the model (wind driven sea level dynamic for example), then you should use a higher output frequency. The point to make is that the dynamic of your model is mainly restricted by the boundary conditions, because the spatial coverage of your model is relatively small. Therefore, the analysis is based on daily mean output files. This might be justified, because you mainly study the longer-term dynamic of the coastal system for the what-if scenarios.

Line 108-109: please correct:  …. and is computed by dividing …. with …. .

**Chapter 2.2: line 117-144: Numerical experiments**

It would be beneficial to include more context regarding the purpose and intended outcomes of the simulations. Reading the text, it is not clear (up to the end) that the main simulation (2015-2019) is the reference simulation for the analysis of the what-if scenarios. Later on you use the term "Reference simulation", but without explaining exactly what it refers to. You could begin with a short overview of the purpose of the simulations followed by a detailed description of the simulation.

The first two sentences (line 117-120) are very general. It's not so clear what these sentences are building towards.

Line 132-135: These lines motivate the numerical studies that have been carried out. They could be presented at the beginning of the chapter.

**Chapter 2.3: Line 146-198: Model validation**

Line 146-148: I think you can write here using the term "Reference simulation" which you have introduced. You don't compare parameters, which are usually fixed, but you use a statistical metric for validation.

Line 146-148: Your validation study is too limited to assess the model quality with regards to reproducing the hydrodynamic in the entire water body of the Danube Delta and Black Sea continuum. I think you should be open about the limitations of your validation study. You can say that you want to give the reader an impression of the quality of the model. It would be good if you could link the validation exercise to specific applications: sea level validation to the prediction of blocking events, or temperature validation to seasonal model forecasts, etc.

Line 149-168: There might be more data available on EmodNet physics, SeaDataNet or ICES.

Table 1: River discharge: I think there is some rounding involved to get to a correlation coefficient of 1.0. I would suggest to show the first decimal which is not zero.

Table 1: Sea level: Why is the sea level bias zero. Have the time series be de-biased before analysis?

Line 173-180: Suggestion: I think it would be good to know how well the model can predict the total runoff entering the Black Sea. The predicted value can be compared with the total runoff of hydrological models or with the results from the study from Nichersu et al 2025.

Line 181: It should be mentioned that daily averaged time series of modelled sea levels were used for validation.

Line 183: Please see also the reply to R2.27. The sentence should be re-formulated: I don't think it's appropriate to draw conclusions about long-term dynamics, which typically span several years, as well as storm surge-related short-term dynamics, which occur over just a few hours. A suggestion would be: The model can reproduce the variability of the daily mean sea level in the year 2017 including mayor sea level events associated with stormy periods of typically 1-10 days.

Line 188: Please see the reply to R2.37. It would be beneficial to demonstrate the dependency of the model quality with regards to sea level prediction on the quality Black Sea physical reanalysis. The impacts of the model quality on the finding of the paper could be discussed.

Line 197: The comparison with satellite SST data could be provided as supplementary material.

**Chapter 3.1: line 204-228: Water division in the river network of the delta**

This is an interesting study, demonstrating the capacity of the model. However, I find it quite separated from the rest of the paper. There are no conclusions drawn from the results of this chapter. As I see this chapter, it shall provide evidence for the adequate modelling of the freshwater input into the Razelm lagoon and the Black Sea, which have implications for the what-if scenario study and the coastal dynamic study.

**Chapter 3.2: line 230-276: Spatial and temporal variability of coastal dynamics**

Line 232: I would rewrite the sentence starting with "Along the coast". I would not call river plumes a peculiar (meaning: strange, unusual) pattern.

Line 235: Please explain the method of deriving coastal pattern in a number of sentences, with references to figures. It should not be done in a bracket. I think you mean averaging over certain periods: (1.) the entire period of 5 years and (2.) daily periods for in 2019.

Line 235-238: Please reformulate this sentence. It is much too long.

Line 240: …. discharged from the nearby Chilia mouth.

Line 254: …. Danube river inflow extends …. . Or use the word "discharge". I would associate the river flow with the velocity of the water in the river.

Line 255: During peak river inflow ……

Line 260: "2m thick layer near the surface"

Line 261-262: It should be made clear that both set of plots in figure 6 show configurations with southerly wind, and that the main difference between them is the season.

I'm not sure that you mention somewhere that you use meteorological definition of wind directions, with winds in the direction they are "coming from", not the oceanographic convention with winds in the direction they are "going to".

Line 262: When you say "between river mouths", then you discuss the surface pattern of the combined upwelling and river plume configuration. Vertically, the configuration looks different, as river plumes do not affect deeper layers.

Line 265: I would not use the word "peculiar".

Line 267-268: Please rewrite "were processed to computed the standard deviation". I also think that you mean "of all the month of the years 2015-2019".

Figure 7: I think it would be better to use "Autumn" instead of "Fall". In the caption it could be mentioned that the plot refers to the model data set covering the period 2015-2019.

**Chapter 3.3: line 278-320: Lagoon water exchange, mixing and renewal capacity**

Line 283: Could you explain the term "flood river conditions"? Are these situations of high river discharge?

Line 284: Better to use "driven by the wind action"

**Chapter 3.3.1: line 322-359: Assessment of the impact of lagoon-sea reconnection solutions**

Line 321: Maybe you can use the term "what-if scenario" in the title, to make it more visible.

Figure 11: Maybe you can plot the reference configuration and the difference to the reference configuration, with different y-axis on the left and right side of the plot to make the differences between the 3 solutions clearer.

Line 322-359: I'm missing a bit a discussion of the different solutions (A, B, C), which one is more in line with the expected outcome of the reconnection of the lagoon with the Black Sea.

Recommendation: I think the structure of the manuscript with assessing three different topics: (1.) Danube river and estuary modelling, (2.) Coastal dynamic modelling and (3.) what-if scenario assessment makes it rather difficult to discuss the different topics in one

single chapter. Even more so, as the 3 different topics are presented separately before the discussion. I would present and discuss each topic separately and write a discussion on the integrated river-to-estuary-and-coastal scale in the end.

**Chapter 4: line 360-464, River-sea modelling, characteristic, requirements and limitations**

There should be a chapter 4: Discussions, before sub-chapter 4.1

Line 361-404: I would recommend to move chapter 4.1 to the introduction or method part of the paper. Lines 364–394 address the requirements for adequate river-to-estuary-to-coastal-sea modelling, effectively justifying the chosen modelling approach and highlighting the novelty of the paper. Line 395-404 deal with the limitation of the validation study and could be moved to chapter 2.3

Line 399: Note: A lot has been done in terms of observation data collection. There are no transnational monitoring systems, because most of the funds come from the national institutions, but there are transnational data collection and cataloguing initiatives, like ICES, EmodNet and SeaDataNet. There are also near real-time ship data exchange initiatives among some of the operational centres. EmodNet is providing river runoff measurements for the Danube from 2022 onwards and is also providing SST observations for several coastal stations in the Black Sea (I didn't check the metadata).

Line 463-464: Could you please explain what you mean with alternative lagoon-sea reconnection solutions. Is this not solution A, B, C. Are you thinking about the timing (design) of the reconnection process, the monitoring of the reconnection process or other activities?

**Chapter 5: Line 465-499: Conclusions**

Note: The authors could provide their perspective on the strengths of their integrated modelling approach compared to separate assessments using standalone hydrological and ocean models.

**Reply to the authors response:**

R2.26: Reply: It was suggested that the total runoff into the sea, as predicted by the hydrological model, could be used as a criterion for assessing the quality of the water transport predictions in the Danube river network. The value predicted by hydrological model could be compared to the predictions of the SHYFEM model. This would make it possible to estimate the quality of the SHYFEM forecast not just near the source, but also near the estuary of the river system. However, this was just a suggestion for the case that no observed runoff data was available.

Danube river observations from EmodNet exist, but unfortunately only after 2022.

R2.27: Reply: If I understand the argumentation of the authors correctly, they say that the model's geographical coverage is too small to generate wind driven sea level extremes caused by storm surges within the model domain. Therefore, they rely on the boundary conditions to provide the sea level forcing. But as daily mean sea levels from the Black Sea reanalysis are used as boundary conditions, they can not expect to model wind-driven sea level anomalies characterized by sharp peaks lasting several hours. Using daily mean values will remove most of these peaks from the data set. The highest peak that you are predicting is therefore also about 20cm. I think it would be better to write "stormy periods" rather than storm surges. This reply also applies to R2.34.

R2.31: Reply: thank you for adding a link to Nichersu et al. (2025). Would it be possible to compare the actual values of the runoff calculations?

R2.36: Reply: Thanks for adding a table of statistical parameters. Is the correlation coefficient of the river discharge really equal to 1.0? This would only be possible if the modelled and observed values would coincide. I guess it's rounded up.

R2.37: Reply: Thanks for adding the Black Sea reanalysis time series to the sea level plot. But it's really hard for me to distinguish between the dashed SHYFEM time series and the dotted BLKSEA time series. Maybe you can calculate the correlation coefficient (CC) and the root-mean-square-deviation (RMSD) of the two timeseries (SHYFEM AND BLKSEA) to demonstrate how the daily mean sea level variation at Constanta and Mangalia depends on the boundary condition.

R2.42: Here, I thought about linking the purpose of the validation closer to the applications.

R2.48: Reply: There is a temperature dependency in the conversion, which makes it that 1 PSU is not exactly equal to 1 g/L. I just wanted to make the point that in the oceanographic community usually PSU is used as a unit. But g/L can be used, if it is applied consistently.

R2.58: Reply: Thank you for motivating the choice of the what-if scenario. In the light of this application, the "Numerical experiment" section could be restructured a bit, mentioning the purpose of the simulations, to study different the river-open-sea continuum and different proposals for connecting the Danube delta with the Black Sea in the beginning. Then it would be consistent to present the reference configuration and the what-if scenarios. Right now, the term "reference simulations" is introduced later, with only little connection to the "Numerical Experiment" section.

R2.66: Reply: Thank you for changing the figure 11 (in the current document). It has much improved, but is still rather difficult to see. Maybe it would be possible to present the results of the reference simulation and difference plots, choosing maybe different axis for the different plots. This is just a suggestion.

R2.70: Reply: I'm not sure about the formulation "small scale nearshore feature, situated between river mouths". Maybe you should refer to a figure here. Is this not just the surface

picture that you are describing. It's more the overlaying configuration of the upwelling pattern and the river plume. The vertical plots show that the rivers only affect the surface, whereas the upwelling pattern extends to deeper layers.

---

## Editor Decision (ED1)

Dear Authors

I now have the referees' comments on your revised manuscript; I assume you have seen them (please tell me if you do not have them). Below I have some *italicised comments* on how you might respond to them. This looks like a lot but I should tell you that both referees rated this manuscript as "good" and asked for (only) "minor revision"; please do what is practicable to address these comments.

Yours sincerely

John Huthnance (editor)

Referee 1

First paragraph of the review "The manuscript would still benefit from the development of a more consistent story line and the formulation of a research question." *In your line 30 "investigates" is rather vague and would be clarified by immediate connection to lines 34-38 where – implicitly – you state the aims by stating what you quantified.* "But the limitations of the paper should be stated clearly, preferably in the introduction or method section, rather than, as the authors have chosen, in the result-discussion section (4.1)." *I think probably in the method section, but only as limitations known before doing the work. It is quite conventional to discuss limitations as part of a Discussion section, perhaps especially the limitations that appear by doing the work.*

"Chapter 1: Line 13-38: Introduction . . ." *I mostly agree. Your present introduction is quite short and could include more about "the state of the art" and how you intend to advance it. However, in the introduction you should avoid referring to what the manuscript achieves or to the results obtained.*

"Chapter 2.1: Line 70-115: . . ." *The specific points on lines, 74, . . 90 are all improvements.* "Line 99-104: . . ." *I am not sure about this. Implementation values are most naturally kept with the formulation they relate to. But perhaps aspects of formulation (wind drag, bottom friction, vertical diffusivity – and boundary conditions?) should come before time step and output frequency. More references would help.*
"Line 103-104: . . ." *Maybe the point is that (for example) stress resulting from a daily-average flow is not equal to (usually greater than) daily-average stress. So it is necessary to use variables at the frequency of the time step for all (especially non-linear) calculations before "final" output.*
"Line 108-109: please correct: …. and is computed by dividing …. with …. ." *I think "…. and is computed by dividing …. by …. ."*

"Chapter 2.2: line 117-144: Numerical experiments . . ." *I generally agree with these comments.*

"Chapter 2.3: Line 146-198: Model validation . . ."  *I generally agree with these comments so please consider them.  Of course for validation you can only use data that you can find.  You should put enough of your previous responses in the manuscript for other readers not to ask the same questions that you responded to.*

"Chapter 3.1: line 204-228: Water division . . ."  *This is probably fair comment but the last paragraph of this section 3.1 is important regarding some form of validation and its first and last sentences.  Respond as best you can but do keep this section in some form.*

"Chapter 3.2: line 230-276: Spatial and . . ."  *I agree with a majority of the comments.  In lines 232, 265 maybe "peculiar" –> "particular".  However, I think you should choose your wording for lines 240, 254, 255, not necessarily follow the referee.  In line 260, best to avoid "surficial" but you might simply want "2 m surface layer".*

"Chapter 3.3: line 278-320: Lagoon . . ."  *Line 283 I agree with the referee but Line 284 either "induced" or "driven" is OK.*

"Chapter 3.3.1: line 322-359: Assessment . ."  *Line 321 -  I agree.  Figure 11 – I am not sure; there is a strong point that "Sol. A" and "Sol. B" are very close and "Sol. C" and "REF" are very close.  Difference plots (A-B or C-REF) would need a different scale.  Line 322-359 – is there an "expected outcome of the reconnection"?*

"Recommendation . ."  *I think this is a suggestion that you might move the discussion relating to (1), (2), (3) to the respective sub-sections of section 3.  I do not insist on this but you should think about it.*

"Chapter 4: line 360-464, River-sea modelling . ."  *What you do about "There should be a chapter 4: Discussions, before sub-chapter 4.1" depends on what you do about "Recommendation . .".*

"Line 361-404:" *See above about "First paragraph of the review"*

"Line 399" and "Line 463-464".  *Please respond to these comments in the revised manuscript.*

"Chapter 5: Line 465-499: Conclusions . ."  *I agree*

"Reply to the authors response:"  *Not all of this asks for changes, but do consider what requested changes would be easy to do.*

Referee 2.  *Please take account of these comments as best you can.  The references to L105 and L121 might be looking at a different version of the manuscript*

Editor comments previously sent to you.

Line 276.  This part of the sentence does not relate to the previous part.  Maybe ". . and of the winds . ." but the sentence is too long.

Line 318.  "which resulted mostly influenced" needs correction.

---

## Author Response (AR2)

**Responses to the Editor Comments and Suggestions**

Journal: Ocean Sciences (OS)
Manuscript number: egusphere-2025-606
Manuscript title: Modelling river-sea continuum: the case of the Danube Delta

The original Editor's comments and suggestions are shown in regular typeface, while our responses are shown in italics. The line and figures numbers we use refer to the revised document.

**ED.1** I now have the referees' comments on your revised manuscript; I assume you have seen them (please tell me if you do not have them). Below I have some italicised comments on how you might respond to them. This looks like a lot but I should tell you that both referees rated this manuscript as "good" and asked for (only) "minor revision"; please do what is practicable to address these comments.

*Response. We thank the editor for the careful reading of the manuscript. We improved the manuscript in accordance with the suggestions.*

**ED.2** First paragraph of the review "The manuscript would still benefit from the development of a more consistent story line and the formulation of a research question." In your line 30 "investigates" is rather vague and would be clarified by immediate connection to lines 34-38 where - implicitly - you state the aims by stating what you quantified. "But the limitations of the paper should be stated clearly, preferably in the introduction or method section, rather than, as the authors have chosen, in the result-discussion section (4.1)." I think probably in the method section, but only as limitations known before doing the work. It is quite conventional to discuss limitations as part of a Discussion section, perhaps especially the limitations that appear by doing the work.

*Response: We appreciate the comments and we improved the manuscript in accordance with all suggestions. In particular, we modified the introduction at lines 45-54 to better clarify the scope and content of this study.*

*The limitations of this modelling study are primarily presented in the Methods section (e.g., the exclusion of the delta floodplain system from the computational domain).*

**ED.3** "Chapter 1: Line 13-38: Introduction . . ." I mostly agree. Your present introduction is quite short and could include more about "the state of the art" and how you intend to advance it. However, in the introduction you should avoid referring to what the manuscript achieves or to the results obtained.

*Response: In response to the referee's comment, we have moved the general statements from Section 4.1 to the Introduction. See also the response to comment R2.31.*

**ED.4** "Chapter 2.1: Line 70-115: . . ." The specific points on lines, 74, . . 90 are all improvements.

*Response: The text has been corrected following the referee's recommendations.*

**ED.5** Line 99-104: . . ." I am not sure about this. Implementation values are most naturally kept with the formulation they relate to. But perhaps aspects of formulation (wind drag, bottom friction, vertical diffusivity - and boundary conditions?) should come before time step and output frequency. More references would help.

*Response: We improved the description of the parameters and settings by restructuring the original paragraph into the following sentences.*

- *Lines 91-92: "Vertical viscosity and diffusivity are calculated by the $k - e$ turbulence closure module of the General Ocean Turbulence Model (Burchard and Petersen, 1999)."*

- *Lines 118-121: "The drag coefficient for the momentum transfer of wind has been set to a constant value of $2.5 \ 10^{-3}$. The friction in the model is parameterized by a quadratic bottom friction expression following the Strickler formulation (Umgiesser et al., 2004, 2022). Due to the lack of data on bottom sediment characteristics, no spatial variation in bottom friction was applied, and the Strickler coefficient was uniformly set to $32 \ m^{1/3} \ s^{-1}$."*

- *Lines 138-140: "The main simulation (hereinafter referred to as the reference simulation, or REF) covers the period from 2014/01/01 to 2019/12/31, with the year 2014 considered as model spin-up time. The results are analyzed over the period 2015-2019."*

- *Lines 151-156: "The maximum allowable time step in the simulation was set to 60 s, and the model adopts automatic sub-stepping over time to enforce numerical stability with respect to advection and diffusion. Model outputs are saved at a daily frequency. This choice was made to limit the volume of model outputs and is justified by the fact that shorter time-scale processes are not relevant in the study area. Indeed, tides along the northwestern Black Sea coast are negligible; therefore, coastal dynamics are primarily influenced by open sea conditions, river discharge, and atmospheric disturbances with typical time scales of 1 to 10 days. Additionally, the model is forced at the boundaries of the Black Sea and the Danube River using daily datasets."*

**ED.6** "Line 103-104: . . ." Maybe the point is that (for example) stress resulting from a daily- average flow is not equal to (usually greater than) daily-average stress. So it is necessary to use variables at the frequency of the time step for all (especially non-linear) calculations before "final" output.

*Response: To clarify this aspect, we have revised the text as follows (lines 152-156) "Model outputs are saved at a daily frequency. This choice was made to limit the volume of model outputs and is justified by the fact that shorter time-scale processes are not relevant in the study area. Indeed, tides along the northwestern Black Sea coast are negligible; therefore, coastal dynamics are primarily influenced by open sea conditions, river discharge, and atmospheric disturbances with typical time scales of 1 to 10 days. Additionally, the model is forced at the boundaries of the Black Sea and the Danube River using daily datasets."*

**ED.7** "Line 108-109: please correct: . . . . and is computed by dividing . . . . with . . . . ." I think ". . . . and is computed by dividing . . . . by . . . . ."

*Response: The text has been corrected following the referee's recommendations.*

**ED.8** "Chapter 2.2: line 117-144: Numerical experiments . . . ." I generally agree with these comments.

*Response: Section 2.2 has been restructured following the referee's recommendations. The first paragraph of this section now reads "The main purpose of the model application is*

*to reproduce the seasonal and interannual variability of the Danube Delta hydrodynamics under the influence of river discharge, heat and momentum fluxes at the water surface, salinity and sea temperature gradients and open sea forcing (sea level oscillations and currents). Moreover, the concurrence of intense atmospheric forcing, direct morphological interventions within the delta territory and freshwater inflows results in the Danube Delta being characterized by a wide range of transport phenomena. The main simulation (hereinafter referred to as the reference simulation, or REF) covers the period from 2014/01/01 to 2019/12/31, with the year 2014 considered as model spin-up time. The results are analyzed over the period 2015-2019."*

*Section 2.2. is now structured as follows: first, the purpose and characteristics of the main simulation are introduced; second, the simulation setting (boundary and forcing conditions, simulation and output time steps) are described; finally, the additional what-if numerical experiments are presented.*

**ED.9** "Chapter 2.3: Line 146-198: Model validation . . ." I generally agree with these comments so please consider them. Of course for validation you can only use data that you can find. You should put enough of your previous responses in the manuscript for other readers not to ask the same questions that you responded to.

*Response: We are aware that the model validation is limited and constrained by the availability of observations during the 2015-2019 period availability on the study site. To clarify this aspect, we inserted the following sentence in lines 169-171 "Limited spatial and temporal coverage of existing monitoring networks, along with restricted availability of freely accessible data, are critical issues in the Danube Delta. Consequently, model validation was constrained by the availability of observations during the 2015-2019 period." Moreover, the issue of limited model validation is also addressed in at the end of the "Model validation" section.*

**ED.10** "Chapter 3.1: line 204-228: Water division . . ." This is probably fair comment but the last paragraph of this section 3.1 is important regarding some form of validation and its first and last sentences. Respond as best you can but do keep this section in some form.

*Response: To stress the importance of accurately capturing water distribution in the delta's river network, we added the following two sentences to section 4.2.:*

- *lines 400-401: "An accurate representation of water distribution within the delta's river network, along with its temporal variability, is crucial for correctly reconstructing the freshwater input into the Black Sea, and consequently, the river plumes and coastal dynamics."*
- *lines 421-422: "Capturing the spatial and temporal variability of water distribution in the delta's river network is essential for accurately modelling the amount of freshwater entering the Razelm Lagoon via the Dunavăţ and Dranov canals."*

**ED.11** "Chapter 3.2: line 230-276: Spatial and . . ." I agree with a majority of the comments. In lines 232, 265 maybe "peculiar" -¿ "particular". However, I think you should choose your wording for lines 240, 254, 255, not necessarily follow the referee. In line 260, best to avoid "surficial" but you might simply want "2 m surface layer".

*Response: The text has been corrected following the referee's recommendations. See the responses to comments R2.20, R2.21 and R2.22.*

**ED.12** "Chapter 3.3: line 278-320: Lagoon . . ." Line 283 I agree with the referee but Line 284 either "induced" or "driven" is OK.

*Response: We replaced flood river with high discharge.*

**ED.13** "Chapter 3.3.1: line 322-359: Assessment . ." Line 321 - I agree. Figure 11 - I am not sure; there is a strong point that "Sol. A" and "Sol. B" are very close and "Sol. C" and "REF" are very close. Difference plots (A-B or C-REF) would need a different scale. Line 322-359 - is there an "expected outcome of the reconnection"?

*Response: The 3.3.1 sections's title has been modified to "Assessment of the potential impact of what-if lagoon-sea reconnection scenarios".*

*We prefer to retain Figure 11 as it is, as it effectively illustratea the impact of the considered reconnection measure on the absolute salinity values.*

*We improved the discussion of the potential impact of the different reconnection measures section 4.2. The text at lines 441-451 now reads "In the RSLS, efforts to enhance ecological status and improve water circulation have prompted exploration into the potential impacts of creating a new inlet to strengthen the lagoon's connection with the sea. The findings presented in section 3.1.1 suggest that even a localized morphological modification can significantly influence the overall hydrodynamics of the lagoon system. Introducing a new inlet in the Razelm Lagoon (scenarios A and B) leads to a 20 % reduction in the RSLS renewal time, which helps mitigate stagnation and enhances ecological conditions. However, it also results in elevated salinity levels up to 9 and 16 g $L^{-1}$ in the Razelm and Sinoie lagoons, respectively. While fisheries and tourist activities would benefit from this intervention, the increased salinization of the lagoon's waters poses a considerable risk to agricultural freshwater resources. In contrast, the impact of solution C on WRT and salinity is limited to the southern part of the Sinoie Lagoon. To help local authorities and communities manage these issues, the model will be used to explore lagoon-sea reconnection solutions with flow regulation based on seasonal and meteo-marine conditions."*

**ED.14** "Recommendation . ." I think this is a suggestion that you might move the discussion relating to (1), (2), (3) to the respective sub-sections of section 3. I do not insist on this but you should think about it.

*Response: Following the referre's suggestion, we restructured the Discussion section to focus on the processes driving the exchange between water bodies. A brief discussion on the implications of the considered reconnection solutions is included at the end of this section.*

**ED.15** "Chapter 4: line 360-464, River-sea modelling . ." What you do about "There should be a chapter 4: Discussions, before sub-chapter 4.1" depends on what you do about "Recommendation . .".

*Response: See the response to comment ED.14.*

**ED.16** "Line 361-404:" See above about "First paragraph of the review"

*Response: In response to the referee's comment, we have moved the general statements from Section 4.1 to the Introduction. See also the response to section ED.14.*

**ED.17** "Line 399" and "Line 463-464". Please respond to these comments in the revised manuscript.

*Response: We concur with the reviewer that significant progress has been made in observational data collection, and that CMEMS, ICES, EMODnet, and SeaDataNet are highly valuable data portals. However, the Danube Delta remains a poorly monitored environment, and not all data collected by local and national authorities is publicly accessible.*

*We modified the mentioned sentence as "To help local authorities and communities manage these issues, the model will be used to explore lagoon-sea reconnection solutions with flow regulation based on seasonal and meteo-marine conditions."*

**ED.18** "Chapter 5: Line 465-499: Conclusions . ." I agree

*Response: We modified the first paragraph of the Conclusions section as "This work presents the first cross-scale hydrodynamic model implementation covering the entire Danube Delta to investigate the river-sea continuum. To study the hydrodynamic processes driving water exchange and connectivity among the various interconnected water compartments of the delta, the 3D unstructured hydrodynamic SHYFEM model was applied to a domain representing the delta river network, the Razelm Sinoie Lagoon System coastal, and part of the western Black Sea shelf. The variable model resolution is of fundamental importance for reproducing the complex morphology of the Danube Delta and achieving a seamless transition across spatial scales, from river branches to the coastal sea. Compared to standalone hydrological and ocean models, the river-sea continuum approach is essential to accurately represent the non-linear and bidirectional interactions among the various water compartments of the delta. In particular, cross-scale modelling is essential in the coastal sea near the river mouths to accurately capture plume dynamics. By contrast, most regional models of the Black Sea (e.g., Lima et al., 2020; Miladinova et al., 2020) employ coarse resolutions (greater than 2 km) and simplified representations of river inputs, which are inadequate for describing the complex coastal circulation patterns revealed in this study."*

**ED.19** "Reply to the authors response:" Not all of this asks for changes, but do consider what requested changes would be easy to do.

*Response: We have addressed all the concerns raised by Referee#2.*

**ED.20** Referee 1. Please take account of these comments as best you can. The references to L105 and L121 might be looking at a different version of the manuscript

*Response: We have addressed all the concerns raised by Referee#1.*

**ED.21** Line 276. This part of the sentence does not relate to the previous part. Maybe ". . and of the winds . ." but the sentence is too long.

*Response: The mentioned sentence has been modified as "The freshwater discharged by the different branches determine a similar salinity standard deviation pattern in winter (Fig. 7e) and fall (Fig. 7h). These findings reflect the seasonal variability in the strength of the main drivers: (i) the Danube River discharge, which usually peaks in spring or early summer, while drought conditions are generally observed in autumn (Fig. 2a); and (ii) wind forcing (both northerly and southerly), which tends to be stronger in winter and autumn (Bajo et al., 2014)."*

**ED.22** Line 318. "which resulted mostly influenced" needs correction.

*Response. The sentence has been shortened. The revised version reads as follows: "The input of marine waters through the Edighiol and Periboina inlets has a limited effect on the local WRT."*

**References**

Bajo, M., Ferrarin, C., Dinu, I., Stanica, A., and Umgiesser, G.: The circulation near the Romanian coast and the Danube Delta modelled with finite elements, Cont. Shelf Res., 78, 62–74, https://doi.org/10.1016/j.csr.2014.02.006, 2014.

Burchard, H. and Petersen, O.: Models of turbulence in the marine environment - a comparative study of two equation turbulence models, J. Mar. Syst., 21, 29–53, https://doi.org/10.1016/S0924-7963(99)00004-4, 1999.

Lima, L., Aydogdu, A., Escudier, R., Masina, S., Ciliberti, S. A., Azevedo, D., Peneva, E. L., Causio, S., Cipollone, A., Clementi, E., Cretì, S., Stefanizzi, L., Lecci, R., Palermo, F., Coppini, G., Pinardi, N., and Palazov, A.: Black Sea Physical Reanalysis (CMEMS BS-Currents) (Version 1)[Data set], https://doi.org/10.25423/CMCC/BLKSEA_MULTIYEAR_PHY_007_004, 2020.

Miladinova, S., Stips, A., Macias Moy, D., and Garcia-Gorriz, E.: Pathways and mixing of the north western river waters in the Black Sea, Estuarine Coastal Shelf Sci., 236, 106 630, https://doi.org/10.1016/j.ecss.2020.106630, 2020.

Umgiesser, G., Melaku Canu, D., Cucco, A., and Solidoro, C.: A finite element model for the Venice Lagoon. Development, set up, calibration and validation, J. Mar. Syst., 51, 123–145, https://doi.org/10.1016/j.jmarsys.2004.05.009, 2004.

Umgiesser, G., Ferrarin, C., Bajo, M., Bellafiore, D., Cucco, A., De Pascalis, F., Ghezzo, M., Mc Kiver, W., and Arpaia, L.: Hydrodynamic modelling in marginal and coastal seas - The case of the Adriatic Sea as a permanent laboratory for numerical approach, Ocean Model., 179, 102123, https://doi.org/10.1016/j.ocemod.2022.102123, 2022.

**Responses to the Referee#1 Comments and Suggestions**

**Journal: Ocean Sciences (OS)**
**Manuscript number: egusphere-2025-606**
**Manuscript title: Modelling river-sea continuum: the case of the Danube Delta**

The original Reviewer's comments and suggestions are shown in regular typeface, while our responses are shown in italics. The line and figures numbers we use refer to the revised document.

**R1.1** The authors have made clear efforts to address the comments; the revised manuscript shows significant improvement in structure and clarity. The messages are more explicit and effectively conveyed and the discussion is substantially strengthened. The responses are detailed and most of the requested information has been incorporated. I am generally satisfied but I have some minor remaining concerns that would be good to address before acceptance.

*Response. We thank the referee for the in-depth and useful review. We appreciate the comments and we improved the manuscript in accordance with all suggestions.*

**R1.2** L105 and R1.2 is the reanalysis only available at daily frequency or do you chose to select daily input for the boundary. If that is the case, you're reasoning is circular. If you chose to use save daily outputs and higher resolution data is available for open boundary conditions, your choice is because smaller scales are not that relevant (i.e. no tide and meteorological events larger scale). If the dataset is only available at daily scales is also likely that it is because shorter scales are not relevant. You explain this better in the reply to reviewer. Could you add this in the manuscript L105?

*Response: Following Reviewer#1's indication, we modified the text as "Model outputs are saved at a daily frequency. This choice was made to limit the volume of model outputs and is justified by the fact that shorter time-scale processes are not relevant in the study area. Indeed, tides along the northwestern Black Sea coast are negligible; therefore, coastal dynamics are primarily influenced by open sea conditions, river discharge, and atmospheric disturbances with typical time scales of 1 to 10 days. Additionally, the model is forced at the boundaries of the Black Sea and the Danube River using daily datasets."*

**R1.3** In R1.8 the authors state that SHYFEM performs slightly better than BLKSEA, however in Table 1 RMSE is larger and CC is lower for BLKSEA than it is for SHYFEM which contradicts their statement. Looking at Fig 1 it would seem that BLKSEA presents slightly elevated values than SHYFEM across the time seriessith BLKSEA overestimaring peaks and SHYFEM reproducing better the low values. Please better indicate the reason for discrepancies between BLKSEA and observations.

*Response: We do not really understand this reviewer's comment. As reported in Table 1 of R1.8, SHYFEM has a lower RMSE and higher CC than BLKSEA. Therefore, SHYFEM is performing better than BLKSEA.*

**R1.4** R1.9 and Fig3 I am curious as to why include station 15360 instead of Constanta when you already have sea levels for Constanta. I understand that you the figure may look unbalanced with 3 sea levels and 2 salinities but I would like to see Constanta time series in the figure. Please do include the time series temperature for Constanta.

*Response: We decided to include station 15360 instead of Constanta because it is located closer to the Danube Delta study area. We preferred to present a balanced Figure 3, with two panels for sea level and two for sea temperature. However, to meet the referee's request, the temperature time series for Constanta is now included in panel (d) instead of Mangalia. This modification does not significantly affect the presentation of the results, as Constanta and Mangalia have similar sea temperature validation statistics, as shown in Table 1.*

**R1.5** R1.31 please add the example at the beginning of March (L287).

*Response: We modified the sentence as (line 322) "(as occurring at the end of February 2018; Figure 8c and d)".*

**R1.6** L121 I think it should be "leads"

*Response: We modified the sentence as (lines 136-138) "Moreover, the concurrence of intense atmospheric forcing, direct morphological interventions within the delta territory and freshwater inflows results in the Danube Delta being characterized by a wide range of transport phenomena."*

**R1.8** L178 please avoid constructions such as "underestimate (overestimate the peak discharge values in the Chilia (Tulcea)" Could you reformulate? This was highlighted as well by R2d.

*Response: We modified the sentence as (lines 201-203) "It is worth noting that the model tends to underestimate peak discharge values in Chilia and overestimate them in the Tulcea, as indicated by the slopes of the linear regression best-fit lines: 0.90 for Chilia and 1.10 for Tulcea."*

**R1.9** L301 you could use WFT instead of flushing time since already defined and you use the acronym afterwards. You have also described what it is, so no need to define it again here.

*Response: Corrected.*

**R1.10** L319 I think you can remove "where".

*Response: We modified the sentence as (lines 352-353) "Salinity shows limited variability across the RSLS, with values ranging from 1 to 8 g $L^{-1}$. The highest values are observed in the area of Sinoie Lagoon, near the Edighiol and Periboina inlets (Fig. 10e)."*

**R1.11** L334 I think it should be "does not".

*Response: Corrected.*

**R1.12** ·L367 I don't think Zhang and Yu 2025 is relevant here, their model is 2D barotropic so it does not resolve the type of processes you are looking at in your paper. Please remove it. If you wanted more Pearl River Delta examples, the following would be more relevant since they do resolve baroclinic processes, you could cite https://doi.org/10.1007/s44218-022-00008-0, or more recent https://doi.org/10.1029/2021JC017523 or https://doi.org/10.1016/j.ocemod.20

*Response: We thank the referee for bringing these references to our attention. We replaced the Zhang and Yu (2025) with Payo-Payo et al. (2022).*

**R1.13** L412 It is the first time you talk about the standard deviation of the relative discharge among branches and temporal variability, in Section 3.1 you're giving averages for the whole period. How accurately these averages are is also relevant for the plumes you obtain later. Perhaps you could develop more the discussion on that aspect.

*Response: To stress the importance of accurately capturing water distribution in the delta's river network, we added the following two sentences to section 4.2.:*

- *lines 400-401: "An accurate representation of water distribution within the delta's river network, along with its temporal variability, is crucial for correctly reconstructing the freshwater input into the Black Sea, and consequently, the river plumes and coastal dynamics."*
- *lines 421-422: "Capturing the spatial and temporal variability of water distribution in the delta's river network is essential for accurately modelling the amount of freshwater entering the Razelm Lagoon via the Dunavăţ and Dranov canals."*

**R1.14** L419 "configuration" is confusing here.

*Response: We modified the sentence as (lines 404-406) "This type of coastal dynamics is common among many of the world's major river deltas - such as the Mississippi and the Nile (Horner-Devine et al., 2015) - as well as in coastal regions where multiple river mouths are located in close proximity (Warrick and Farnsworth, 2017)."*

**R1.15** L426 presumably the temperature of the smaller scale patterns between river mouths either warmer or colder has to do with the background temperature of the ocean and that of the river discharge. Is you river temperature constant? Or does it change throughout the year?

*Response: As mentioned at lines 149-150, water temperature at the Danube River boundary was taken from the daily results of the wflow catchment model implemented over the Danube River basin (van Gils et al., 2025). The vertical alongshore sea temperature transect presented in Figs. 6c and and 6d indicate an upwelling-driven transport of offshore marine waters from deeper layers to the coastal zone.*

**R1.16** L432 the salt intrusion length values are introduced here for the first time and it is difficult to see where you show those values, perhaps point to the figure or introduce those values in the result section so the discussion here makes more sense.

*Response: Compared to the first version of the manuscript, we have removed the section describing saltwater intrusion. Although this phenomenon poses a serious threat to several coastal areas - compromising freshwater supplies for agriculture and human use (Li et al., 2025) - it has not yet been reported as a major issue in the Danube Delta. Therefore, we decided to limit the presentation of the saltwater-related findings to a paragraph in the discussion section.*

**References**

Horner-Devine, A. R., Hetland, R. D., and MacDonald, D. G.: Mixing and Transport in Coastal River Plumes, Annu. Rev. Fluid Mech., 47, 569–594, https://doi.org/10.1146/annurev-fluid-010313-141408, 2015.

Li, M., Najjar, R. G., Kaushal, S., Mejia, A., Chant, R. J., Ralston, D. K., Burchard, H., Hadjimichael, A., Lassiter, A., and Wang, X.: The emerging global threat of salt

contamination of water supplies in tidal rivers, Environ. Sci. Technol. Lett., 12, 881–892, https://doi.org/10.1021/acs.estlett.5c00505, 2025.

Payo-Payo, M., Bricheno, L. M., Dijkstra, Y. M., Cheng, W., Gong, W., and Amoudry, L. O.: Multiscale temporal response of salt intrusion to transient river and ocean forcing, J. Geophys. Res. Oceans, 127, e2021JC017523, https://doi.org/10.1029/2021JC017523, 2022.

van Gils, J., Loos, S., and Boisgontier, H.: Simulated fluxes of water, heat, nutrients, fine sediment for 13 large rivers to the Black Sea 2011-2020, https://doi.org/10.5281/zenodo.15675190, 2025.

Warrick, J. A. and Farnsworth, K. L.: Coastal river plumes: Collisions and coalescence, Prog. Oceanogr., 151, 245–260, https://doi.org/10.1016/j.pocean.2016.11.008, 2017.

Zhang, A. and Yu, X.: Development of a land-river-ocean coupled model for compound floods jointly caused by heavy rainfalls and storm surges in large river delta regions, Hydrol. Earth Syst. Sci., 29, 2505–2520, https://doi.org/10.5194/hess-29-2505-2025, 2025.

**Responses to the Referee#2 Comments and Suggestions**

**Journal: Ocean Sciences (OS)**
**Manuscript number: egusphere-2025-606**
**Manuscript title: Modelling river-sea continuum: the case of the Danube Delta**

The original Reviewer's comments and suggestions are shown in regular typeface, while our responses are shown in italics. The line and figures numbers we use refer to the revised document.

**R2.1** This is the second review of the manuscript "Modelling river-sea continuum: the case of the Danube Delta", by Ferrarin et al. Structure and quality of the text was much improved, but several of the more fundamental points have not been fully addressed. The manuscript would still benefit from the development of a more consistent story line and the formulation of a research question. The authors added a research statement in the introduction focusing on process-oriented model analysis, without providing a definite goal for the analysis. As a consequence, the manuscript demonstrates qualitatively the SHYFEM model's ability to produce realistic results, but it does not necessarily provide sufficient information, for all processes and parameters, to verify the accuracy of the model results quantitatively as well. The scope of the model study is rather broad and covers not just river-sea continuum, but also coastal processes, like upwelling. The model validation study is limited by the amount of data used. This is not a reason to reject the findings. I find them reasonable and qualitatively appropriate. But the limitations of the paper should be stated clearly, preferably in the introduction or method section, rather than, as the authors have chosen, in the result- discussion section (4.1).

*Response: We thank the referee for the in-depth and useful review. We appreciate the comments and we improved the manuscript in accordance with all suggestions. In particular, we modified the introduction at lines 45-54 to better clarify the scope and content of this study.*

*The limitations of this modelling study are primarily presented in the Methods section (e.g., the exclusion of the delta floodplain system from the computational domain).*

**R2.2** Chapter 1: Line 13-38: Introduction The novelty of the paper, to resolve the river-sea continuum within a single unstructured grid model, could be more clearly emphasized. It should be more clearly stated how the presented modelling approach differs from previous studies. Some of this could be achieved by moving parts of section 4.1 (line 360-404) to the introduction section.

*Response: In response to the referee's comment, we have moved the general statements from Section 4.1 to the Introduction. See also the response to comment R2.31.*

**R2.3** The first paragraph (line 71-77) should be modified to remove some errors and make it more readable. Line 74: Suggestion: The model solves the primitive equations using an unstructured finite element grid in the horizontal and a z-coordinate system in the vertical. Line 75: The model has been applied .... Line 76: ... and in other coastal systems. Line 78: SHYFEM was applied to model the river network ... Line 90: The

model bathymetry is derived by bilinearly interpolating the following datasets onto the model's numerical grid. Line 108-109: please correct: .... and is computed by dividing .... with .... .

*Response: The text has been corrected following the referee's recommendations.*

**R2.4** Line 99-104: This paragraph contains on the model components, the configuration and the implementation of the model processes all in one. I would suggest to improve the structure of the text and to use references wherever possible. Maybe you can highlight different choices made for dynamical parameter in the open sea and the river network.

*Response: We improved the description of the parameters and settings by restructuring the original paragraph into the following sentences.*

- *Lines 91-92: "Vertical viscosity and diffusivity are calculated by the $k - e$ turbulence closure module of the General Ocean Turbulence Model (Burchard and Petersen, 1999)."*

- *Lines 118-121: "The drag coefficient for the momentum transfer of wind has been set to a constant value of $2.5 \ 10^{-3}$. The friction in the model is parameterized by a quadratic bottom friction expression following the Strickler formulation (Umgiesser et al., 2004, 2022). Due to the lack of data on bottom sediment characteristics, no spatial variation in bottom friction was applied, and the Strickler coefficient was uniformly set to $32 \ m^{1/3} \ s^{-1}$."*

- *Lines 138-140: "The main simulation (hereinafter referred to as the reference simulation, or REF) covers the period from 2014/01/01 to 2019/12/31, with the year 2014 considered as model spin-up time. The results are analyzed over the period 2015-2019."*

- *Lines 151-156: "The maximum allowable time step in the simulation was set to 60 s, and the model adopts automatic sub-stepping over time to enforce numerical stability with respect to advection and diffusion. Model outputs are saved at a daily frequency. This choice was made to limit the volume of model outputs and is justified by the fact that shorter time-scale processes are not relevant in the study area. Indeed, tides along the northwestern Black Sea coast are negligible; therefore, coastal dynamics are primarily influenced by open sea conditions, river discharge, and atmospheric disturbances with typical time scales of 1 to 10 days. Additionally, the model is forced at the boundaries of the Black Sea and the Danube River using daily datasets."*

**R2.5** Line 103-104: The time frequency of the model output is defined by the dynamic time scales of the processes you intend to resolve and you are able to resolve. If you are able to resolve the processes on the required time scales within the model (wind driven sea level dynamic for example), then you should use a higher output frequency. The point to make is that the dynamic of your model is mainly restricted by the boundary conditions, because the spatial coverage of your model is relatively small. Therefore, the analysis is based on daily mean output files. This might be justified, because you mainly study the longer-term dynamic of the coastal system for the what-if scenarios.

*Response: To clarify this aspect, we have revised the text as follows (lines 152-156) "Model outputs are saved at a daily frequency. This choice was made to limit the volume of model outputs and is justified by the fact that shorter time-scale processes are not relevant in the study area. Indeed, tides along the northwestern Black Sea coast are negligible; therefore, coastal dynamics are primarily influenced by open sea conditions, river discharge, and atmospheric disturbances with typical time scales of 1 to 10 days. Additionally, the model is forced at the boundaries of the Black Sea and the Danube River using daily datasets."*

**R2.6** It would be beneficial to include more context regarding the purpose and intended outcomes of the simulations. Reading the text, it is not clear (up to the end) that the main simulation (2015-2019) is the reference simulation for the analysis of the what-if scenarios. Later on you use the term "Reference simulation", but without explaining exactly what it refers to. You could begin with a short overview of the purpose of the simulations followed by a detailed description of the simulation.

*Response: Section 2.2 has been restructured following the referee's recommendations. The first paragraph of this section now reads "The main purpose of the model application is to reproduce the seasonal and interannual variability of the Danube Delta hydrodynamics under the influence of river discharge, heat and momentum fluxes at the water surface, salinity and sea temperature gradients and open sea forcing (sea level oscillations and currents). Moreover, the concurrence of intense atmospheric forcing, direct morphological interventions within the delta territory and freshwater inflows results in the Danube Delta being characterized by a wide range of transport phenomena. The main simulation (hereinafter referred to as the reference simulation, or REF) covers the period from 2014/01/01 to 2019/12/31, with the year 2014 considered as model spin-up time. The results are analyzed over the period 2015-2019.".*

**R2.7** The first two sentences (line 117-120) are very general. It's not so clear what these sentences are building towards.

*Response: Se the response to comment R2.6.*

**R2.8** Line 132-135: These lines motivate the numerical studies that have been carried out. They could be presented at the beginning of the chapter.

*Response: Section 2.2. is now structured as follows: first, the purpose and characteristics of the main simulation are introduced; second, the simulation setting (boundary and forcing conditions, simulation and output time steps) are described; finally, the additional what-if numerical experiments are presented.*

**R2.9** Line 146-148: I think you can write here using the term "Reference simulation" which you have introduced. You don't compare parameters, which are usually fixed, but you use a statistical metric for validation.

*Response: We modified the mentioned sentence as "The reference simulation was validated by comparing various variables to assess the skill of the SHYFEM model in reproducing the hydrodynamics of the different water compartments within the delta."*

**R2.10** Line 146-148: Your validation study is too limited to assess the model quality with regards to reproducing the hydrodynamic in the entire water body of the Danube Delta and Black Sea continuum. I think you should be open about the limitations of your validation study. You can say that you want to give the reader an impression of the quality of the model. It would be good if you could link the validation exercise to specific applications: sea level validation to the prediction of blocking events, or temperature validation to seasonal model forecasts, etc.

*Response: We are aware that the model validation is limited and constrained by the availability of observations during the 2015-2019 period availability on the study site. To clarify this aspect, we inserted the following sentence in lines 169-171 "Limited spatial and temporal coverage of existing monitoring networks, along with restricted availability of freely accessible data, are critical issues in the Danube Delta. Consequently, model validation was constrained by the availability of observations during the 2015-2019 period." Moreover, the issue of limited model validation is also addressed in at the end of the "Model validation" section.*

**R2.11** Line 149-168: There might be more data available on EmodNet physics, SeaDataNet or ICES.

*Response: We checked the EmodNet and SeaDataNet and they contains the same sea level and sea temperature time series we obtained from the in-situ ocean thematic centre of the Copernicus Marine Service.*

**R2.12** Table 1: River discharge: I think there is some rounding involved to get to a correlation coefficient of 1.0. I would suggest to show the first decimal which is not zero.

*Response: The correlation coefficients were rounded to the second decimal, as for the other variables.*

**R2.13** Table 1: Sea level: Why is the sea level bias zero. Have the time series be de-biased before analysis?

*Response: Yes, due to unknown reference datum, the sea level time series were de-biased prior the analysis. This aspect is now mentioned in lines 207-208.*

**R2.14** Line 173-180: Suggestion: I think it would be good to know how well the model can predict the total runoff entering the Black Sea. The predicted value can be compared with the total runoff of hydrological models or with the results from the study from Nichersu et al 2025.

*Response: The SHYFEM numerical grid does not include the delta floodplain system - comprising channels, wetlands, lakes, and marshes - as simulating flood dynamics over these areas is beyond the scope of this study. Therefore, un our model application, the total runoff entering the Black Sea is equal to the total water discharge imposed at the upstream river boundary of Isaccea, excluding the evaporation-precipitation fluxes over the river network. We would like to point out that there is no exact estimate of the total runoff, as no monitoring network or hydrological model currently has the capacity to resolve the hydrological complexity of the delta.*

**R2.15** Line 181: It should be mentioned that daily averaged time series of modelled sea levels were used for validation.

*Response: Corrected.*

**R2.16** Line 183: Please see also the reply to R2.27. The sentence should be re-formulated: I don't think it's appropriate to draw conclusions about long-term dynamics, which typically span several years, as well as storm surge-related short-term dynamics, which occur over just a few hours. A suggestion would be: The model can reproduce the variability of the daily mean sea level in the year 2017 including mayor sea level events associated with stormy periods of typically 1-10 days.

*Response: Corrected.*

**R2.17** Line 188: Please see the reply to R2.37. It would be beneficial to demonstrate the dependency of the model quality with regards to sea level prediction on the quality Black Sea physical reanalysis. The impacts of the model quality on the finding of the paper could be discussed.

*Response: As already specified in out previous reply, it is not our intention to present a validation of the Black Sea Physics Reanalysis in this manuscript. A proper model comparison was provided in our previous report, which also include, in Table 2, a statistical analysis (in terms of RMSE and CC) of daily sea levels from both SHYFEM and BLKSEA at the monitoring stations.*

**R2.18** Line 197: The comparison with satellite SST data could be provided as supplementary material.

*Response: We respectfully disagree with the reviewer's statements. The comparison with satellite SST data is the only model validation we were able to perform for the Razelm Sinoie Lagoon System; therefore, we prefer to retain it in the main manuscript.*

**R2.19** Chapter 3.1: line 204-228: Water division in the river network of the delta This is an interesting study, demonstrating the capacity of the model. However, I find it quite separated from the rest of the paper. There are no conclusions drawn from the results of this chapter. As I see this chapter, it shall provide evidence for the adequate modelling of the freshwater input into the Razelm lagoon and the Black Sea, which have implications for the what-if scenario study and the coastal dynamic study.

*Response: To stress the importance of accurately capturing water distribution in the delta's river network, we added the following two sentences to section 4.2.:*

- *lines 400-401: "An accurate representation of water distribution within the delta's river network, along with its temporal variability, is crucial for correctly reconstructing the freshwater input into the Black Sea, and consequently, the river plumes and coastal dynamics."*
- *lines 421-422: "Capturing the spatial and temporal variability of water distribution in the delta's river network is essential for accurately modelling the amount of freshwater entering the Razelm Lagoon via the Dunavăţ and Dranov canals."*

**R2.20** Line 232: I would rewrite the sentence starting with "Along the coast". I would not call river plumes a peculiar (meaning: strange, unusual) pattern.

*Response: The mentioned sentence has been modified as "Along the coast, these processes create distinct hydrodynamic patterns, the so-called river plumes, having thermohaline characteristics and buoyancy that allow to distinguish them from seawater".*

**R2.21** Line 235: Please explain the method of deriving coastal pattern in a number of sentences, with references to figures. It should not be done in a bracket. I think you mean averaging over certain periods: (1.) the entire period of 5 years and (2.) daily periods for in 2019.

*Response: The mentioned sentence has been modified as "Off the coast of the Danube Delta, the overall coastal circulation - averaged over the entire period of 5 years - reveals distinct freshwater plumes formed by the various branches of the multi-mouth delta. The shape and size of these plumes are influenced by both the volume of water discharged from each river branch and the specific features of the coastline (Fig. 5a)."*

**R2.22** Line 235-238: Please reformulate this sentence. It is much too long. Line 240: .... discharged from the nearby Chilia mouth. Line 254: .... Danube river inflow extends .... . Or use the word "discharge". I would associate the river flow with the velocity of the water in the river. Line 255: During peak river inflow ...... Line 260: "2m thick layer near the surface"

*Response: Corrected.*

**R2.23** Line 261-262: It should be made clear that both set of plots in figure 6 show configurations with southerly wind, and that the main difference between them is the season.

*Response: We modified Figure 6 caption as "Maps of sea surface temperature and north-to-south alongshore transects of sea temperature under southerly wind conditions during winter (10 January 2015; panels a and c) and spring (15 May 2015; panels b and d). The transect location is indicated with a red line in panel a."*

**R2.24** I'm not sure that you mention somewhere that you use meteorological definition of wind directions, with winds in the direction they are "coming from", not the oceanographic convention with winds in the direction they are "going to".

*Response: For wind direction, we applied the standard meteorological convention. In contrast, the oceanographic convention is typically used for waves and currents. We believe it is unnecessary to explicitly state this in the manuscript.*

**R2.25** Line 262: When you say "between river mouths", then you discuss the surface pattern of the combined upwelling and river plume configuration. Vertically, the configuration looks different, as river plumes do not affect deeper layers.

*Response: We didn't understand the referee's comment. We never state that river plumes affect the deeper part of the water column.*

**R2.26** Line 265: I would not use the word "peculiar". Line 267-268: Please rewrite "were processed to computed the standard deviation". I also think that you mean "of all the month of the years 2015-2019". Figure 7: I think it would be better to use "Autumn" instead of "Fall". In the caption it could be mentioned that the plot refers to the model data set covering the period 2015-2019.

*Response: Corrected.*

**R2.27** Line 283: Could you explain the term "flood river conditions"? Are these situations of high river discharge? Line 284: Better to use "driven by the wind action"

*Response: We replaced "flood river" with "high discharge".*

**R2.28** Line 321: Maybe you can use the term "what-if scenario" in the title, to make it more visible.

*Response: The 3.3.1 sections's title has been modified to "Assessment of the potential impact of what-if lagoon-sea reconnection scenarios".*

**R2.29** Figure 11: Maybe you can plot the reference configuration and the difference to the reference configuration, with different y-axis on the left and right side of the plot to make the differences between the 3 solutions clearer.

*Response: We prefer to retain Figure 11 as it is, as it effectively illustratea the impact of the considered reconnection measure on the absolute salinity values.*

**R2.30** Line 322-359: I'm missing a bit a discussion of the different solutions (A, B, C), which one is more in line with the expected outcome of the reconnection of the lagoon with the Black Sea.

*Response: We improved the discussion of the potential impact of the different reconnection measures section 4.2. The text at lines 441-451 now reads "In the RSLS, efforts to enhance ecological status and improve water circulation have prompted exploration into the potential impacts of creating a new inlet to strengthen the lagoon's connection with the sea. The findings presented in section 3.1.1 suggest that even a localized morphological modification can significantly influence the overall hydrodynamics of the lagoon system. Introducing a new inlet in the Razelm Lagoon (scenarios A and B) leads to a 20 % reduction in the RSLS renewal time, which helps mitigate stagnation and enhances ecological conditions. However, it also results in elevated salinity levels up to 9 and 16 $g\ L^{-1}$ in the Razelm and Sinoie lagoons, respectively. While fisheries and tourist activities would benefit from this intervention, the increased salinization of the lagoon's waters poses a considerable risk to agricultural freshwater resources. In contrast, the impact of*

*solution C on WRT and salinity is limited to the southern part of the Sinoie Lagoon. To help local authorities and communities manage these issues, the model will be used to explore lagoon-sea reconnection solutions with flow regulation based on seasonal and meteo-marine conditions."*

**R2.31** Recommendation: I think the structure of the manuscript with assessing three different topics: (1.) Danube river and estuary modelling, (2.) Coastal dynamic modelling and (3.) what-if scenario assessment makes it rather difficult to discuss the different topics in onesingle chapter. Even more so, as the 3 different topics are presented separately before the discussion. I would present and discuss each topic separately and write a discussion on the integrated river-to-estuary-and-coastal scale in the end.

*Response: Following the referre's suggestion, we restructured the Discussion section to focus on the processes driving the exchange between water bodies. A brief discussion on the implications of the considered reconnection solutions is included at the end of this section.*

**R2.32** There should be a chapter 4: Discussions, before sub-chapter 4.1.

*Response: See the response to comment R2.31.*

**R2.33** Line 361-404: I would recommend to move chapter 4.1 to the introduction or method part of the paper. Lines 364-394 address the requirements for adequate river-to-estuary-to-coastal-sea modelling, effectively justifying the chosen modelling approach and highlighting the novelty of the paper. Line 395-404 deal with the limitation of the validation study and could be moved to chapter 2.3.

*Response: In response to the referee's comment, we have moved the general statements from Section 4.1 to the Introduction. See also the response to section R2.31.*

**R2.34** Line 399: Note: A lot has been done in terms of observation data collection. There are no transnational monitoring systems, because most of the funds come from the national institutions, but there are transnational data collection and cataloguing initiatives, like ICES, EmodNet and SeaDataNet. There are also near real-time ship data exchange initiatives among some of the operational centres. EmodNet is providing river runoff measurements for the Danube from 2022 onwards and is also providing SST observations for several coastal stations in the Black Sea (I didn't check the metadata).

*Response: We concur with the reviewer that significant progress has been made in observational data collection, and that CMEMS, ICES, EMODnet, and SeaDataNet are highly valuable data portals. However, the Danube Delta remains a poorly monitored environment, and not all data collected by local and national authorities is publicly accessible.*

**R2.35** Line 463-464: Could you please explain what you mean with alternative lagoon-sea reconnection solutions. Is this not solution A, B, C. Are you thinking about the timing (design) of the reconnection process, the monitoring of the reconnection process or other activities?

*Response: We modified the sentence as "To help local authorities and communities manage these issues, the model will be used to explore lagoon-sea reconnection solutions with flow regulation based on seasonal and meteo-marine conditions."*

**R2.36** Line 465-499: Conclusions. Note: The authors could provide their perspective on the strengths of their integrated modelling approach compared to separate assessments using standalone hydrological and ocean models.

*Response: We modified the first paragraph of the Conclusions section as "This work presents the first cross-scale hydrodynamic model implementation covering the entire Danube Delta to investigate the river-sea continuum. To study the hydrodynamic processes driving water exchange and connectivity among the various interconnected water compartments of the delta, the 3D unstructured hydrodynamic SHYFEM model was applied to a domain representing the delta river network, the Razelm Sinoie Lagoon System coastal, and part of the western Black Sea shelf. The variable model resolution is of fundamental importance for reproducing the complex morphology of the Danube Delta and achieving a seamless transition across spatial scales, from river branches to the coastal sea. Compared to standalone hydrological and ocean models, the river-sea continuum approach is essential to accurately represent the non-linear and bidirectional interactions among the various water compartments of the delta. In particular, cross-scale modelling is essential in the coastal sea near the river mouths to accurately capture plume dynamics. By contrast, most regional models of the Black Sea (e.g., Lima et al., 2020; Miladinova et al., 2020) employ coarse resolutions (greater than 2 km) and simplified representations of river inputs, which are inadequate for describing the complex coastal circulation patterns revealed in this study."*

**R2.37** R2.26: Reply: It was suggested that the total runoff into the sea, as predicted by the hydrological model, could be used as a criterion for assessing the quality of the water transport predictions in the Danube river network. The value predicted by hydrological model could be compared to the predictions of the SHYFEM model. This would make it possible to estimate the quality of the SHYFEM forecast not just near the source, but also near the estuary of the river system. However, this was just a suggestion for the case that no observed runoff data was available. Danube river observations from EmodNet exist, but unfortunately only after 2022.

*Response: See the response to comment R2.14.*

**R2.38** R2.27: Reply: If I understand the argumentation of the authors correctly, they say that the model's geographical coverage is too small to generate wind driven sea level extremes caused by storm surges within the model domain. Therefore, they rely on the boundary conditions to provide the sea level forcing. But as daily mean sea levels from the Black Sea reanalysis are used as boundary conditions, they can not expect to model wind-driven sea level anomalies characterized by sharp peaks lasting several hours. Using daily mean values will remove most of these peaks from the data set. The highest peak that you are predicting is therefore also about 20cm. I think it would be better to write "stormy periods" rather than storm surges. This reply also applies to R2.34.

*Response: We did not state that the model's geographical coverage is too limited to generate wind-driven sea level extremes caused by storm surges within the domain. Our coastal model requires sea level forcing at the open sea boundary, which, in the presented simulations, is derived from the daily Black Sea Physics Reanalysis. As demonstrated in our previous report, boundary conditions have a significant influence on sea level fluctuations within the model domain. However, the model is forced by 3-hourly meteorological fields, which produce sub-daily variability. Nevertheless, we chose to use daily averaged sea levels for model validation, as atmospheric disturbances in the study area typically occur over timescales ranging from 1 to 10 days. This is clearly illustrated in Figure 1, which shows time series of hourly and daily sea level measurements at Constanța.*

**R2.39** R2.31: Reply: thank you for adding a link to Nichersu et al. (2025). Would it be possible to compare the actual values of the runoff calculations?

*Response: See the response to comment R2.14.*

[Figure]

Figure 1: Time series of hourly and daily sea levels measured at Constanta.

**R2.40** R2.36: Reply: Thanks for adding a table of statistical parameters. Is the correlation coefficient of the river discharge really equal to 1.0? This would only be possible if the modelled and observed values would coincide. I guess it's rounded up.

*Response: See the response to comment R2.12.*

**R2.41** R2.37: Reply: Thanks for adding the Black Sea reanalysis time series to the sea level plot. But it's really hard for me to distinguish between the dashed SHYFEM time series and the dotted BLKSEA time series. Maybe you can calculate the correlation coefficient (CC) and the root- mean-square-deviation (RMSD) of the two timeseries (SHYFEM AND BLKSEA) to demonstrate how the daily mean sea level variation at Constanta and Mangalia depends on the boundary condition.

*Response: See the response to comment R2.17.*

**R2.42** R2.42: Here, I thought about linking the purpose of the validation closer to the applications.

*Response: We believe that the revised text addresses the referee's concerns.*

**R2.43** R2.48: Reply: There is a temperature dependency in the conversion, which makes it that 1 PSU is not exactly equal to 1 g/L. I just wanted to make the point that in the oceanographic community usually PSU is used as a unit. But g/L can be used, if it is applied consistently.

*Response: Yes, we are aware of the temperature dependency in the conversion.*

**R2.44** R2.58: Reply: Thank you for motivating the choice of the what-if scenario. In the light of this application, the "Numerical experiment" section could be restructured a bit, mentioning the purpose of the simulations, to study different the river-open-sea continuum and different proposals for connecting the Danube delta with the Black Sea in the beginning. Then it would be consistent to present the reference configuration and the what-if scenarios. Right now, the term "reference simulations" is introduced later, with only little connection to the "Numerical Experiment" section.

*Response: See the response to comment R2.6.*

**R2.45** R2.66: Reply: Thank you for changing the figure 11 (in the current document). It has much improved, but is still rather difficult to see. Maybe it would be possible to present the results of the reference simulation and difference plots, choosing maybe different axis for the different plots. This is just a suggestion.

*Response: See the response to comment R2.29.*

**R2.46** R2.70: Reply: I'm not sure about the formulation "small scale nearshore feature, situated between river mouths". Maybe you should refer to a figure here. Is this not just the surface picture that you are describing. It's more the overlaying configuration of the upwelling pattern and the river plume. The vertical plots show that the rivers only affect the surface, whereas the upwelling pattern extends to deeper layers.

*Response: Following the referee's recommendation, we included the a reference to Figure 6.*

**References**

Burchard, H. and Petersen, O.: Models of turbulence in the marine environment - a comparative study of two equation turbulence models, J. Mar. Syst., 21, 29–53, https://doi.org/10.1016/S0924-7963(99)00004-4, 1999.

Lima, L., Aydogdu, A., Escudier, R., Masina, S., Ciliberti, S. A., Azevedo, D., Peneva, E. L., Causio, S., Cipollone, A., Clementi, E., Cretì, S., Stefanizzi, L., Lecci, R., Palermo, F., Coppini, G., Pinardi, N., and Palazov, A.: Black Sea Physical Reanalysis (CMEMS BS-Currents) (Version 1)[Data set], https://doi.org/10.25423/CMCC/BLKSEA_MULTIYEAR_PHY_007_004, 2020.

Miladinova, S., Stips, A., Macias Moy, D., and Garcia-Gorriz, E.: Pathways and mixing of the north western river waters in the Black Sea, Estuarine Coastal Shelf Sci., 236, 106 630, https://doi.org/10.1016/j.ecss.2020.106630, 2020.

Umgiesser, G., Melaku Canu, D., Cucco, A., and Solidoro, C.: A finite element model for the Venice Lagoon. Development, set up, calibration and validation, J. Mar. Syst., 51, 123–145, https://doi.org/10.1016/j.jmarsys.2004.05.009, 2004.

Umgiesser, G., Ferrarin, C., Bajo, M., Bellafiore, D., Cucco, A., De Pascalis, F., Ghezzo, M., Mc Kiver, W., and Arpaia, L.: Hydrodynamic modelling in marginal and coastal seas - The case of the Adriatic Sea as a permanent laboratory for numerical approach, Ocean Model., 179, 102123, https://doi.org/10.1016/j.ocemod.2022.102123, 2022.